# Hippocampal sharp-wave ripples correlate with periods of naturally occurring self-generated thoughts in humans

Takamitsu Iwata [1], Takufumi Yanagisawa [1,2] ✉, Yuji Ikegaya [3,4,5], Jonathan Smallwood [6], Ryohei Fukuma [1,2], Satoru Oshino [1], Naoki Tani [1], Hui Ming Khoo [1] & Haruhiko Kishima [1]

Core features of human cognition highlight the importance of the capacity to focus on information distinct from events in the here and now, such as mind wandering. However, the brain mechanisms that underpin these self-generated states remain unclear. An emerging hypothesis is that self-generated states depend on the process of memory replay, which is linked to sharp-wave ripples (SWRs), which are transient high-frequency oscillations originating in the hippocampus. Local field potentials were recorded from the hippocampus of 10 patients with epilepsy for up to 15 days, and experience sampling was used to describe their association with ongoing thought patterns. The SWR rates were higher during extended periods of time when participants' ongoing thoughts were more vivid, less desirable, had more imaginable properties, and exhibited fewer correlations with an external task. These data suggest a role for SWR in the patterns of ongoing thoughts that humans experience in daily life.

Both in laboratory settings and in daily life, human cognition often escapes the here and now to focus on information unrelated to the immediate environment[1,2]. Estimates suggest that in daily life, the human mind can wander from the immediate environment for up to 30% of the time[2], with patterns of self-focused future thought characteristically occurring during activities such as exercising or commuting, while patterns of distracting intrusive thoughts occur frequently at rest[3]. An emerging body of literature has begun to uncover the complex associations these states have with intelligence[4–6], autism[7], attention-deficit disorder[8] and happiness and well-being[2,9]. Neuroimaging evidence suggests that these processes are linked to activity in regions such as the posterior cingulate and the medial prefrontal cortex, both of which are key nodes in what is known as the default mode network[10–12]. Likewise, studies examining variation in cortical anatomy have highlighted individual differences in the

medial temporal lobe as important for trait variation in these experiences in both the laboratory and daily life[13,14].

Despite this emerging evidence, we know relatively little about how these experiences are orchestrated. Contemporary views on these experiences hypothesize that they may be linked to hippocampal function[15–17] given their role in memory and prospection[18] as well as the possibility that the medial temporal lobe plays a general role in the organization of cortical function[19,20]. One possibility is that the process of self-generated thought is linked to the emergence of sharp-wave ripples (SWRs) in the hippocampus. SWRs are bursts of synchronized neuronal activity in the hippocampus that vary depending on the state of the animal[21,22]. SWRs frequently occur during non-REM sleep associated with memory consolidation[21–23]; during wakefulness[24,25], SWRs increase when animals stay in a quiescent motionless state and decrease during active

[1]Department of Neurosurgery, Graduate School of Medicine, Osaka University, Osaka 565-0871, Japan. [2]Institute for Advanced Co-Creation Studies, Osaka University, Osaka 565-0871, Japan. [3]Laboratory of Chemical Pharmacology, Graduate School of Pharmaceutical Sciences, The University of Tokyo, Tokyo 113-0033, Japan. [4]Institute for AI and Beyond, The University of Tokyo, Tokyo 113-0033, Japan. [5]National Institute of Information and Communications Technology, Center for Information and Neural Networks, Suita City, Osaka 565-0871, Japan. [6]Department of Psychology, Queen's University, Kingston, ON K7L 3N6, Canada. ✉e-mail: tyanagisawa@nsurg.med.osaka-u.ac.jp

movements[21,22], a situation that—in humans—is linked to the emergence of intrusive thoughts[3]. Moreover, SWRs during wakefulness are believed to aid in memory retrieval and guide decision-making based on past experiences[26,27]. Recent studies using human intracranial recordings have revealed that SWR rates in humans are related to mental contents that emerge in episodic recollection during cognitive tasks[23,28]. In light of these observations, coupled with evidence that lesions to the hippocampus can decrease experiences such as mind wandering[29], our study explored the hypothesis that SWR fluctuations are linked not only to circadian rhythms and physical activities[21,22] but also to accompanying mental states that are less closely linked to events in the immediate environment.

In this study, we continuously measured human hippocampal local field potentials (LFPs) for 9–15 days in 10 patients with intracranial electrodes implanted in or adjacent to the hippocampus for the presurgical evaluation of seizure onset zones and memory function (long-term electroencephalography (EEG) monitoring, Supplementary Tables 1, 2). All patients were monitored by video camera while they freely engaged in normal activities during neural activity recording (Fig. 1). In parallel with the LFP recordings, we assessed the patients' mood and thought contents using multidimensional Experience Sampling (mDES)[30,31]. Specifically, patients were asked to rate their experiences along 17 dimensions using a tablet device once every hour (see Table 1 for the list of dimensions). Additionally, we simultaneously recorded information on the patients' physical states, such as interbeat interval (IBI), electrodermal activity (EDA), accelerometer (ACC), and blood volume pulse (BVP) data, using a wearable device attached to their left arm. We analyzed the associations between these simultaneously recorded signals to better understand how human SWRs change over the circadian rhythm and to characterize their relationships with not only individuals' physical states but also the content of their ongoing thoughts.

## Results

### Diurnal fluctuations in hippocampal SWR rates during long-term EEG monitoring

SWRs were assessed using LFP recordings starting 4 days after electrode implantation. We evaluated the LFP signals between two electrodes located either within or near the hippocampus. The electrode locations were identified using pre- and postoperative computed tomography (CT) and magnetic resonance imaging (MRI) scans (Fig. 2a). The periods associated with epileptic activity or movement artifacts were excluded to detect ripple candidates (see Supplementary Fig. 10 for the histogram of the deleted periods). The LFP of hippocampal electrodes in the selected site was then converted to a bipolar signal. After removing the power-line noise, the signals were filtered between 70 and 180 Hz to select ripple candidates (Fig. 2b, see Method). Ripple candidates whose durations were shorter than 20 ms or longer than 200 ms were rejected, and peaks less than 30 ms apart were concatenated. The detected SWRs demonstrated characteristics consistent with those observed in a previous study, including similarities in waveforms, dominant frequency, and inter-ripple intervals[23,28] (Fig. 2c–e). The number of detected SWRs during each minute was Z-standardized over each day to obtain an SWR rate for each patient.

Figure 2f (top) shows the SWRs of a representative patient (Pt-04); overall, the SWR rate increased during the night, exhibiting intense rate fluctuations, and decreased upon waking (see Supplementary Tables 6, 7, and 8 for the details of the patients' behaviors). For six patients with implanted cortical and hippocampal electrodes, we computed the delta wave amplitude, an indicator of slow-wave sleep, according to the neocortical electrode data to examine the relationship between the delta amplitude and SWR rate (Fig. 2f, bottom). Over 7 days of recording among six patients, the SWR rates were significantly correlated with delta amplitudes according to a general linear mixed-effects model (Fig. 2g; $R^2 = 0.19$, $n = 6768$ time points; see

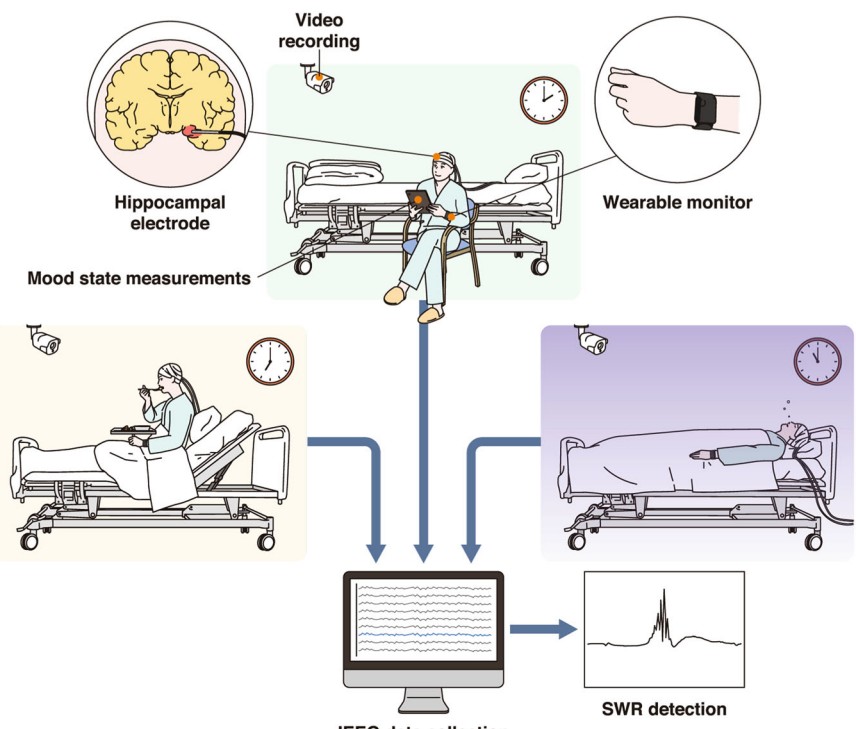

**Fig. 1 | Experimental environment.** We continuously measured human hippocampal LFPs for 9–15 days in 10 patients who had intracranial electrodes implanted in or adjacent to their hippocampus for presurgical evaluation of seizure onset zones and memory function. All patients were monitored by two video cameras in a room where they freely engaged in normal activities during LFP recording. In parallel with the LFP recordings, we assessed the patients' moods and thoughts using a tablet once every hour. Additionally, we simultaneously recorded the patients' physical state using a wearable device attached to their left arm. This figure was designed by MEDICAL EDUCATION INC. under a Creative Commons Attribution 4.0 International license. LFP local field potential.

**Table 1 | Multidimensional experience sampling questionnaire on thoughts and feelings**

| Dimension | Questions | 1 | 7 |
|---|---|---|---|
| External-task Focus | My thoughts were focused on the task I was performing. | Not at all | Completely |
| Future | My thoughts involved future events. | Not at all | Completely |
| Past | My thoughts involved past events. | Not at all | Completely |
| Self | My thoughts involved myself. | Not at all | Completely |
| Social | My thoughts involved other people. | Not at all | Completely |
| Emotion | The content of my thoughts was: | Negative | Positive |
| Visually imaginable | My thoughts were in the form of images. | Not at all | Completely |
| Words | My thoughts were in the form of words. | Not at all | Completely |
| Vivid | My thoughts were vivid as if I was there. | Not at all | Completely |
| Detail | My thoughts were detailed and specific. | Not at all | Completely |
| Habit | This thought has recurrent themes similar to those I have had before. | Not at all | Completely |
| Unwanted | My thoughts were what I did not want to think. | Not at all | Completely |
| Evolving | My thoughts tended to evolve in a series of steps. | Not at all | Completely |
| Spontaneous | My thoughts were: | Spontaneous | Deliberate |
| Happy | My feeling was: | Sad | Happy |
| Calm | My feeling was: | Restless | Calm |
| Excited | My feeling was: | Boring | Excited |

For each question, patients were asked to "rate how you feel or what you are thinking about right now" on a scale from 1 to 7 by selecting one of seven buttons representing internal states. All questions were asked using a tablet.

Supplementary Figs. 7, 9). In addition, the correlation coefficients between the SWR rates and delta amplitudes of each patient were similar during the day and night (Fig. 2h; $R = 0.38 \pm 0.25$ for daytime and $0.36 \pm 0.19$ for nighttime; $t_5 = 0.2993$; $P = 0.7768$; paired $t$ test; Bayes factor$_{10} = 0.387$). These data are consistent with a previous report demonstrating that SWR rates vary with sleep cycles[21].

For subsequent analyses, the SWR rates were $Z$-standardized for each day, averaged for each patient, and pooled among all ten patients. The SWR rate showed significant overall variation over a 24-h period (Fig. 2i; $P < 0.001$, $F_{143,83392} = 41.6$, $n = 113,760$ time points from a total of 79 days in 10 patients; one-way ANOVA; see Supplementary Figs. 18, 19 for the diurnal fluctuation in the SWR duration and the frequency of clustered or isolated SWR events). In particular, the SWR event rate was significantly higher during the night ($0.234 \pm 1.072$ counts/min (mean ± SD)) than during the daytime ($-0.153 \pm 0.918$) ($P < 0.001$, $t_{116186} = -65.58$, $n = 156960$ time points; student's $t$ test). In addition, the SWR rates decreased at 7:00, 12:00, and 18:00, coinciding with the scheduled mealtimes in the hospital, with the rates returning to baseline levels after meal completion (see Supplementary Tables 3–8 and Supplementary Fig. 12). During the day, the SWR rates were higher during the afternoon than during the morning (morning (from 8:00 to 11:59), $-0.21 \pm 0.90$; afternoon (from 13:00 to 17:59), $-0.08 \pm 0.97$; $P < 0.001$, $t_{28,136} = -11.7$, $n = 42,660$ time points, student's $t$ test). These findings indicate that SWR rates exhibit characteristic circadian rhythms in humans during long-term EEG monitoring.

We also assessed the effects of epileptic activity on the SWR. Among all recordings, we identified 177 epileptic seizures, which lasted 19 min on average. In the above analysis, we excluded signals occurring during epileptic seizures and signals occurring 30 min before to 30 min after the epileptic seizure period to reduce the effects of epileptic activity on the SWR results. However, the SWR event rates for 30 min before and 30 min after the seizure periods were not significantly different, suggesting that epileptic seizures did not significantly affect the SWR event rates (see Supplementary Fig. 16; SWR rates 30 min before seizure were $0.38 \pm 0.25$, and those after seizure were $0.36 \pm 0.19$; $t_{4755} = 0.5464$; $P = 0.5848$; paired student's $t$ test). In addition, there was no significant difference in the frequency of epileptic seizures between mealtime and non-mealtime intervals (mealtime (7:00–7:59, 12:00–12:59 and 18:00–18:59), $0.036 \pm 0.0086$; non-

mealtime (0:00–6:59, 8:00–11:59, 13:00–16:59, 19:00–23:59), $0.043 \pm 0.016$; $P = 0.33$, Wilcoxon rank sum test; Supplementary Figs. 10, 11).

**Correlation between SWR rates and biophysical parameters**

We examined whether the SWR rates were also associated with changes in the activities and autonomic states of patients. Specifically, the EDA, ACC, BVP, and IBI data of the patients were $Z$-standardized among each patient as for SWR and then averaged across patients (Fig. 3a). Among the four physiological scores, the IBI varied significantly over 24 h ($F_{22,1086} = 1.68$, $n = 1108$ time points, $P = 0.0261$ (EDA), $F_{22,1086} = 1.54$, $P = 0.0518$ (ACC), $F_{22,1125} = 1.06$, $P = 0.3918$ (BVP), $F_{22,1073} = 8.68$, $P < 0.001$ (IBI), $n = 1095$ time points, one-way ANOVA, Bonferroni corrected, see also Supplementary Fig. 23). Then, a generalized linear mixed-effects model was applied to explain the variations of the $Z$-standardized SWR rates based on the four measurements and time of day (hour). The SWR rates were weakly explained by the model using the physiological scores at 1 min after the SWR event with the highest accuracy ($R^2 = 0.024$, $n = 82,595$ time points, $P < 0.05$, permutation test, Bonferroni corrected, Fig. 3b). At the event time with the highest accuracy, the IBI (estimated coefficient (standard error) (EC (SE)) = 0.11 (0.0053), $P = 4.6 \times 10^{-95}$), ACC (EC (SE) = 0.011 (0.0053), $P = 0.0387$) and EDA (EC (SE) = −0.029 (0.00533), $P = 8.2 \times 10^{-8}$) significantly explained diurnal variations in the SWR rates (Fig. 3d).

**Correlations of SWR rates with thoughts and feelings**

To investigate the correlation between SWR rates and thought patterns, the patients answered mDES questions every hour between 8 am and 8 pm. They were asked to indicate their thought content (14 questions) and mood (3 questions), immediately preceding their responses to the mDES questions, on a Likert scale from 1 to 7. Before the data were analyzed, the responses were standardized for each participant. There was no significant diurnal variation in thought content or mood among the patients ($P > 0.05$, $n = 187$ responses; one-way ANOVA, Bonferroni-corrected) (Fig. 4a, see also Supplementary Fig. 4). We used a generalized linear mixed-effects model to investigate the $Z$-standardized SWR rates averaged over 5 min, shifting every 1 min from 15 min before to 5 min after answering the questionnaire based

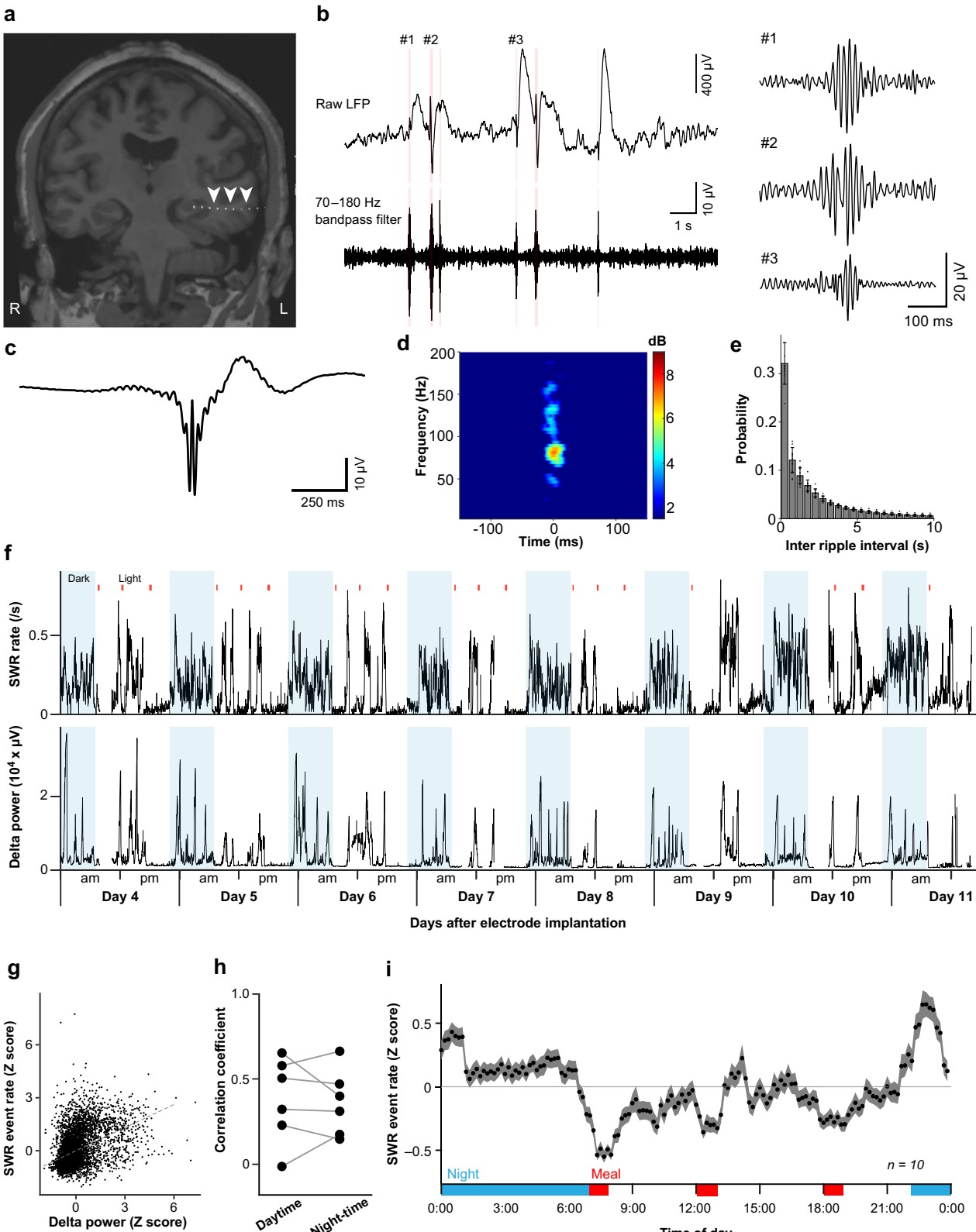

on the standardized values of the responses to each question (Fig. 4b). The mean SWR rates from −5 min to 0 min were strongly explained by the answers with the highest accuracy above the chance level ($R^2 = 0.576$, $n = 160$ responses, $P < 0.05$, permutation test, Bonferroni corrected). Specifically, we found that across the 17 dimensions analyzed, as the SWR rate increased, the patients reported less focus on an external task (estimated coefficient (standard error) (EC (SE)) = −0.236

(0.041), $P = 4.8 \times 10^{-8}$), more vivid mental contents (EC (SE) = 0.122 (0.036), $P = 0.0009$), less desired experiences (EC (SE) = 0.137 (0.037), $P = 0.0003$), less future content (EC(SE) = −0.110 (0.040), $P = 0.00673$), and more visually imaginable content (EC(SE) = 0.0960 (0.0357), $P = 0.00807$) (Fig. 4c, also see Supplementary Figs. 6 and 13). However, mood state did not significantly contribute to the fluctuation in the SWR rates. In addition, the interictal epileptic discharges were

**Fig. 2 | Diurnal variation in hippocampal SWR rates during long-term EEG monitoring. a** An image combining preoperative MRI and postimplantation CT data illustrates the placement of electrodes within the hippocampus. The white arrowheads designate the electrodes implanted in the hippocampus.
**b** Hippocampal LFPs (top) were bandpass-filtered between 70 and 180 Hz (bottom). The light red background indicates the duration detected as SWR. Three representative SWRs (asterisks) are magnified in the right inset. Mean waveform of field potential (**c**) and time frequency map (**d**) centered on ripple peak (time 0, $n = 633161$ SWR events from 10 patients). **e** Distribution of inter-ripple intervals across all patients. ($n = 10$ patients; Data are presented as the mean values ± standard deviation). **f** A time course of SWR rates (top) and neocortical delta power (bottom) for 8 consecutive days (top) in a representative patient. The light blue

background indicates the time of day when the room was darkened. The red lines represent mealtimes. **g** The normalized SWR rates of 6 patients for 10 days are positively correlated with the normalized delta power. The dotted line shows the fixed effect fitted by the generalized linear mixed-effects model. Each dot represents a single time bin of 10 min. **h** The correlation coefficients between SWR rates and delta power did not differ between daytime and nighttime in a total of six patients with implanted cortical electrodes. **i** The $Z$ score mean and 95% confidence interval of the diurnal variation in SWR rates over 24 h in all 10 patients. The red and blue bars indicate meal and sleep times, respectively. Source data are provided as a Source Data file. MRI magnetic resonance imaging, CT computed tomography, LFP local field potential, SWR sharp-wave ripple.

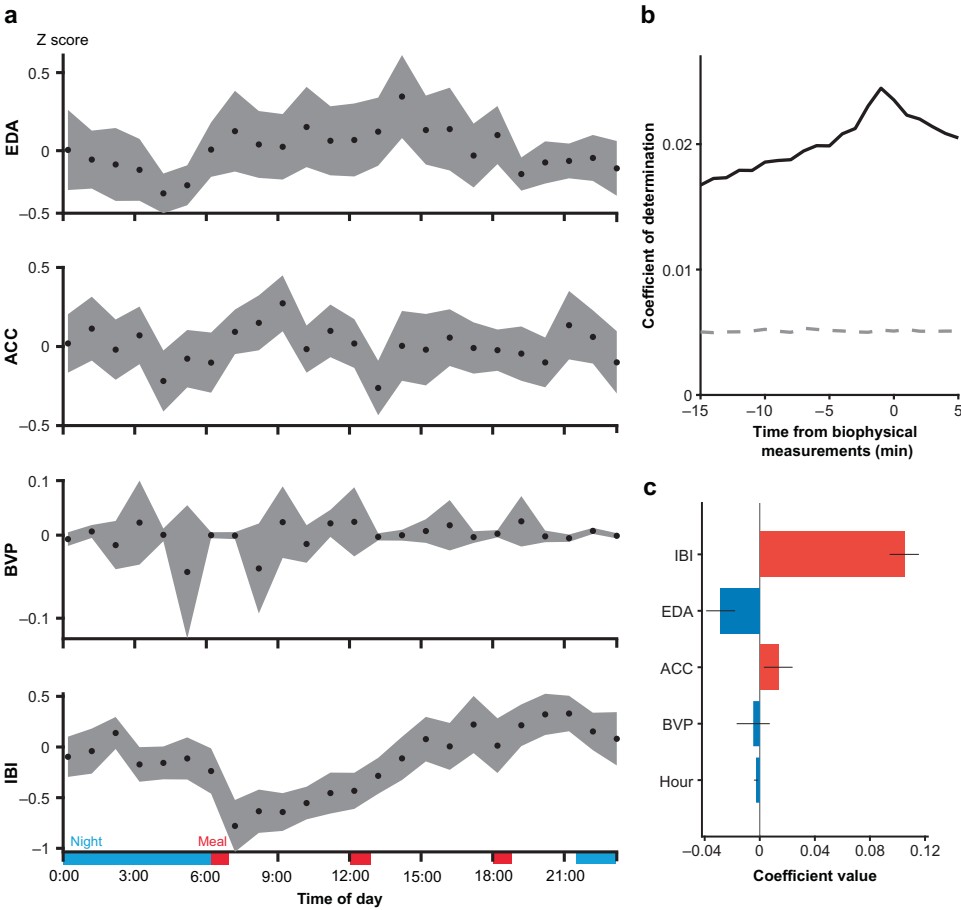

**Fig. 3 | Correlation between SWR rates and biophysical parameters. a** Diurnal variations in the mean and 95% confidence interval of the Z-scores of the EDA, ACC, BVP, and IBI values for all 9 patients. For each value, one-way ANOVA with Bonferroni-correction was applied to the Z-scores among 24 h (EDA, $F_{22,1086} = 1.68$, $n = 1108$ time points, $P = 0.0261$; ACC, $F_{22,1086} = 1.54$, $P = 0.0518$; BVP, $F_{22,1125} = 1.06$, $P = 0.3918$; IBI, $F_{22,1073} = 8.68$, $P < 0.001$; $n = 1095$ time points). The red and blue bars indicate the meal and sleep times, respectively. **b** The temporal relationship between the SWR rate and biometric data was evaluated in 1-min time windows from 10 min before to 5 min after the acquisition of the biophysical data. A general

linear mixed-effects model was applied for each time window. The gray dotted line indicates the $P = 0.05$ level after Bonferroni correction. **c** Coefficient of each predictor in the general linear mixed-effects model ($n = 82,595$ time points for 9 patients). The red and blue bars indicate negative and positive coefficients, respectively. Error bars represent the 95% confidence intervals for the coefficients. Source data are provided as a Source Data file. EDA electrodermal activity, ACC accelerometer, BVP blood volume pulse, IBI interbeat interval, SWR sharp-wave ripple.

explained with low accuracy by the thought patterns ($R^2 = 0.04$, $P > 0.05$ permutation test, $n = 160$ questions, Supplementary Fig. 17). Overall, these findings suggest that SWRs are related to ongoing thought patterns experienced by humans in daily life.

## Discussion

Hippocampal SWRs have been extensively studied in relation to memory processes in tasks; however, their relationship to naturally occurring patterns of ongoing thoughts in humans is not yet fully

understood. Here, we report evidence that human SWRs are related to the content of ongoing thoughts in a manner that is consistent with an emerging hypothesis linking the medial temporal lobe to patterns of self-generated thought[15].

Our study demonstrated a diurnal fluctuation during long-term EEG monitoring, characterized by a decrease in SWR activity around mealtimes and an increase in the afternoon compared to the morning. Previous studies in rodents demonstrated that SWRs occur during nonexploratory states, such as slow-wave sleep, rest, grooming, and

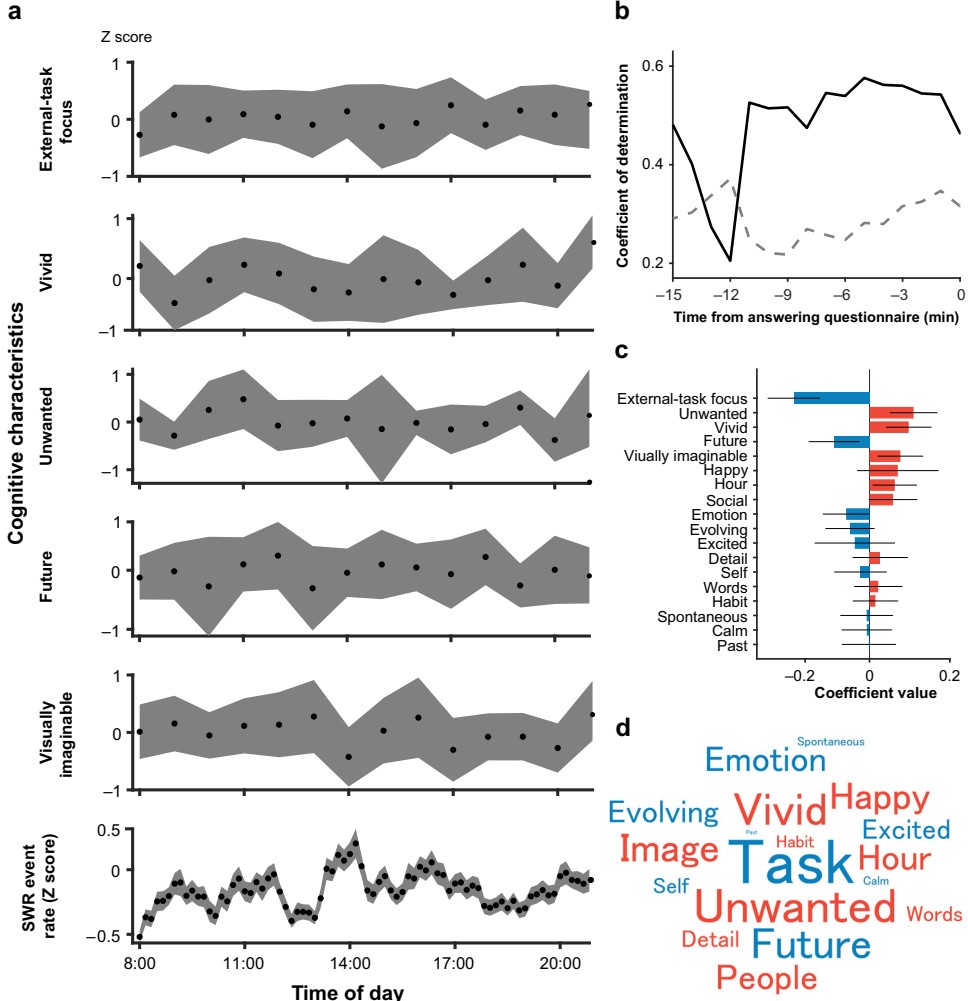

**Fig. 4 | Correlations of SWR rates with thoughts and feelings. a** Diurnal variations in the mean and 95% confidence interval of Z-score data for focused, vivid, unwanted, future, and visually imaginable thoughts in all 10 patients. For each cognitive characteristic, one-way ANOVA with Bonferroni-correction was applied to the Z-score among 13 h (external-task focus, $P = 1.00$, $F_{13,165} = 0.26$; vivid, $P = 0.61$, $F_{13,165} = 0.84$; unwanted, $P = 0.59$, $F_{13,165} = 0.87$; future, $P = 0.97$, $F_{13,165} = 0.41$; visually imaginable, $P = 0.82$, $F_{13,165} = 0.82$; $n = 187$ responses). **b** The temporal relationship between SWR rate and thought sampling data with a 5-min time window from 15 min before to 5 min after answering the questionnaire. The gray dotted line indicates the $P = 0.05$ level after Bonferroni correction. **c** Coefficient of each predictor in the mixed-effects model ($n = 160$ responses for 10 patients). The red and blue bars indicate negative and positive coefficients, respectively. Error bars represent the 95% confidence intervals for the coefficients. **d** A word cloud displaying the coefficient of each feature in the linear regression model. Red and blue indicate positive and negative coefficients, respectively, and the size of each word indicates the magnitude of the absolute value of the coefficient. Task, Images, and People indicates "External-task focus" and "visually imaginable", and "Social", respectively. Note that only the spontaneous nature of cognition varied with time of day; the other three dimensions of experience are included in Panel A because they were related to the occurrence of hippocampal SWR. Source data are provided as a Source Data file. SWR sharp-wave ripple.

eating/drinking[21]. Our results were consistent with the rodents' study in the slow-wave sleep but contradicted prior associations with eating. Although we did not investigate SWR activity during the sleep stage due to the lack of electrooculogram and electromyogram data, which are essential for identifying the stage of sleep, the clear circadian rhythm of the SWRs, which increased during the night in tandem with the cortical delta power, is consistent with the changes in SWR activity during the sleep-wake cycle observed in animal studies. Similarly, the escalation in SWRs during the early evening (13:00–15:00) may be associated with napping, mirroring the concurrent delta power patterns. On the other hand, the decreased SWR activity observed during eating differed from the results of previous studies on rodents. Although blood glucose concentrations have been reported to be inversely correlated with SWR rates[32], our data showed that this relationship alone cannot explain the decline in SWR rates at the start of mealtime (see Supplementary Fig. 12). While a previous study showed a possible link between epileptic seizures and food intake[33], our analysis

did not reveal any significant difference in the frequency of epileptic seizures or any artifacts between mealtime and non-mealtime intervals (Supplementary Figs. 10, 11). In addition, our results were consistent with the increased occurrence of SWRs during non-REM (NREM) sleep[21], which implies a relation between SWRs and the sympathetic nervous system. However, the weak relation between the SWRs and the physiological data, which we used to identify sleep and wakefulness states, suggests the effects of the sympathetic nervous system alone could not explain the variability in the SWR activity. Interestingly, in our study, SWRs were more strongly associated with the content of ongoing thoughts than with physical activity.

Animal studies are limited in terms of their ability to investigate cognition that cannot be directly linked to behavior. However, using experience sampling, we identified elevated SWRs during extended periods of time when participants reported experiences characterized by less-desired, more vivid, and visually imaginable thoughts, which were less connected to external tasks or future considerations.

Interestingly, the SWR rates in the 5 min preceding questionnaire responses were explained by the thought pattern, but the SWR rates while answering the questionnaire were less effectively explained by the same thought pattern, suggesting a potential alteration in thought patterns induced by the task of answering the questionnaire. Together, these findings support an emerging hypothesis relating the medial temporal lobe with self-generated cognition[30]. For example, prior studies have shown that hippocampal lesions are associated with reduced off-task thoughts, such as mind wandering[29]. In addition, gray matter integrity in the hippocampus/parahippocampus has been linked with the capacity for mind wandering in patients with dementia[34], while, in healthy adults, reports of ongoing thoughts with more vivid features are linked to greater gray matter volumes in the posterior parahippocampus, while volume in the anterior parahippocampus is linked to off-task states[13]. More generally, the hippocampus is functionally aligned with the default-mode network[28,35], and functional MRI studies show that self-generated contents are linked to activity in anterior regions of this system, while vivid and detailed experiences have links to posterior regions[20,36,37]. Finally, SWRs are also important in laboratory tasks that mimic naturally occurring self-generated thought, including future planning, memory recall, and imagination[23,38]—situations that depend on the self-generation of mental content and can also include a reduction in the perceptual processing of external input[39] (see Supplementary Fig. 13).

Experience sampling offers a complementary approach for mapping brain function to cognition in contrast to the utilization of specific laboratory tasks[40]. Here, we used the mDES scores to relate thought patterns to SWRs. Prior studies have shown that mDES scores allow researchers to reliably categorize thought patterns in different situations in the lab[40] and daily life[3,41]. In the laboratory, mDES scores have been shown to effectively characterize the association between brain activity and thought patterns that reflect task-relevant processes[42], off-task social episodic thoughts during tasks[10], and thoughts that emerge at rest[43]. These studies have highlighted several features of ongoing thought that are relevant to the present work. First, intrusive thought patterns (broadly similar to those seen in the present study) are most clearly observed in daily life when individuals are unoccupied by a task (doing nothing or resting[3]). It is worth noting that our mDES requests patients to report their state immediately before answering the question, otherwise, patients would always report their thoughts during the task of answering the question, making it difficult for them to report intrusive thought patterns. Second, studies that have used functional magnetic resonance imaging data to characterize thought patterns at rest have identified that such states can show increased activation within the medial temporal lobe[43]. Together with the present study, these studies suggest that SWRs may influence self-generated states that occur during periods when individuals are unoccupied by demanding external tasks.

Although mDES is useful for understanding the correlations between cognition and neural activity that are not directly related to an external task, important limitations exist to the inferences derived from this method in probing brain-cognition relationships. For example, our data demonstrates an association between SWRs and reports provided by the participants over a relatively extended period preceding the probe (~5 min). Moreover, mDES and other experience-sampling methods may be incapable of determining the duration of episodes of self-generated thought[15] as they cannot establish when the experience commenced. Consequently, our data is consistent with the possibility that SWRs may not be directly related to the experiential content of thoughts. Instead, they may be associated with the activation of a cognitive mode that facilitates the emergence of thought patterns characterized by the observed features. To gain a deeper understanding regarding the SWR role in ongoing thought patterns, future studies could benefit from combining task manipulations[42] and pharmacological interventions[17] with activity recordings within the hippocampus and descriptions of experience gained from mDES. This multi-method approach could help triangulate the mechanistic role of SWRs in both task-driven and self-generated features of cognition. Our study also has important clinical implications because it links SWRs in the hippocampus to unwanted ongoing thoughts, a type of thinking that is common in daily life[44] and that is prevalent in conditions such as obsessive-compulsive disorder[44], anxiety and depression[45], and posttraumatic stress disorder[46]. Our evidence highlighting associations between unwanted thoughts and the hippocampus using mDES may be a useful method for elaborating on contemporary views of intrusive thoughts as emerging when the hippocampus is no longer regulated by the prefrontal cortex[47]. Although our present evaluation of SWRs relies on invasive recording techniques, prior resting-state studies have shown that the medial temporal lobe is related to intrusive thoughts[43]. Moreover, SWRs and delta power are strongly correlated[48] (Supplementary Fig. 7), and delta waves can be recorded by noninvasive EEG, enabling the prediction of features related to spontaneous experiences[49,50] (see Supplementary Fig. 25). In the future, we could combine brain imaging approaches with task-based methods and experience sampling to better understand the neural basis of intrusive thinking in different clinical states.

Although our data suggest a correlation between SWRs in the hippocampus and the content of ongoing thought patterns, there are several limitations that should be borne in mind when considering these results. First, while we carefully excluded the effects of epileptic activity by excluding the period corresponding to the seizures, it remains unclear whether epileptic activity influenced the detection of SWR events and the hippocampal function or led to changes in ongoing patterns of thought. The frequency of the detected SWR events was not influenced by the timing of epileptic activities, despite an increase in power observed in the ripple range, as well as low and high-frequency ranges (Supplementary Fig. 24). However, it is difficult to exclude the possibility that the detected SWRs include some pathological high-frequency oscillations (HFOs), which is one of the epileptic activities[51]. Although the frequency and amplitude characteristics of the detected SWRs were consistent with those of the SWRs (Fig. 2d; see also Supplementary Fig. 27), it was difficult to distinguish between the SWRs and the pathological HFOs based on each waveform[52]. Previous studies have demonstrated that physiological SWRs and pathological HFOs coexist in the epileptic hippocampus, while the physiological SWRs maintain the functional characteristics even with the coexistence of the pathological HFOs[53,54]. Therefore, the observed relationship between the ongoing thought patterns and SWRs would reflect the physiological property of the SWR even with the contaminated pathological HFOs. In addition, although the frequency of interictal epileptic activity was weakly explained by the thought sampling results, the model weights differed from those of the SWR model (Supplementary Fig. 17); suggesting the relationship between SWR and ongoing thought patterns were not originated from the relationship with the epileptic activities. Moreover, the frequency of the epileptic activity and excluded periods did not explain the diurnal variation in the SWR (Supplementary Fig. 10). Finally, although the patients took anti-epileptic medication during mealtime, the SWR event rates decreased at the start of the meal and before the administration of medicines (Supplementary Fig. 12). These results suggest that the epileptic-related activities were not directly attributed to the relation between SWRs and thought content. On the other hand, although intracranial recordings are the gold standard for linking hippocampal activity to cognition[16], it will be important for future studies to use high-field imaging methods[55] to understand how neural activity within the medial temporal lobe contributes to ongoing thoughts not only in patients but also, critically, in healthy controls. Second, the

recordings were conducted in a hospital, rather than in the patients' homes or other natural environments. This situation constrains the possible activities patients can engage in and consequently may impact the accompanying patterns of thoughts[3]. It is possible, therefore, that SWR contributes to the features of ongoing thoughts more broadly than was observed in our study. In addition, the number of responses varied among patients because of features related to their condition or the occurrence of medical examinations. These issues may have facilitated the occurrence of certain thought states (in our case, self-generated thoughts in the resting state, which are more prone to intrusive features compared to other activities[3]). Thus, while our study highlights the contribution of SWRs to certain ongoing thought patterns, we cannot rule out the contribution of SWRs to thought patterns that occur in different situations and that may have different features. Additional research with a larger cohort may be needed to elucidate the relationship between SWRs and different features of human thought. Third, it is important to reiterate that our data are correlational and that the temporal link found between the SWR rate and intrusive thoughts is broad; thus, further studies that manipulate SWRs and associated cognitive functions are needed to fully understand the role the hippocampus plays in naturally occurring features of human cognition. In this regard, since SWRs in the hippocampus are influenced by neuro-modulators, it is possible that pharmacological manipulation of these systems may provide the opportunity to test causal accounts of the role that SWRs play in ongoing thought patterns in humans[17] (see Supplementary Fig. 13 for a preliminary investigation using behavioral induction).

In conclusion, our analysis of long-term intracranial EEG data from the human hippocampus has demonstrated a diurnal fluctuation in hippocampal SWRs during periods of wakefulness. Our findings suggest that these SWRs not only change over the sleep-wake cycle but also vary in association with the emergence of human thought contents that are unrelated to the immediate environment. Our study, therefore, paves the way for future studies of continuous intracranial EEG and thought sampling that will help clarify the function that SWRs play in human cognition.

## Methods

### Participants
In our hospital, 23 patients with drug-resistant epilepsy underwent intracranial electrode implantation for presurgical evaluation of seizure onset zones and memory function between January 2020 and May 2022. Of these, 15 patients (16 electrodes) had depth electrodes implanted in the hippocampus or parahippocampal gyrus. To minimize the influence of epileptic activity, recordings were excluded from the analysis if the hippocampus was pathologically diagnosed with hippocampal sclerosis. Ultimately, ten patients with eleven electrodes (self-reported sex: 6 males, 4 females; age: $33.6 \pm 14.2$ years; mean ± standard deviation [SD]; see Supplementary Table 1) who consented to participate in this study (i.e., to complete the questionnaire and wear a wearable device) were included. Electrode placement was determined solely by clinical necessity. In all patients, the clinical team determined the placement of the electrodes to best localize epileptogenic regions. To minimize the selection bias, the EEG data analyst did not participate in the determination of electrode placement. It is possible that selection bias could be introduced by not including data from patients who were uncooperative in responding to the questionnaire or wearing the wearable device. Data were collected at the Department of Neurosurgery at Osaka University Hospital. The research protocol was approved by the Institutional Review Board (approval no. 14353, UMIN000017900), and informed consent was obtained from the participants. All data were acquired in the participants' rooms during their hospital stays. The analysis included only EEG data from 4 days post implantation onward to minimize the impact of surgery. Based on visual inspection of the recorded EEG data, seizures and segments of intense epileptic activity were discarded from subsequent analyses.

### Behavioral data
Postoperatively, patients were able to walk around in the hospital room with the intracranial electrodes connected to the EEG system (Fig. 1). The connection to the EEG system was temporarily disconnected for bathing or clinical examinations. Within the room, patients were constantly recorded by video synchronized with EEG data. The initiation and termination of eating and lighting were determined based on the simultaneously recorded video; in one case, video annotation was not possible due to missing data.

**Physiological signals.** We simultaneously recorded patients' physiological states, such as EDA, three-dimensional acceleration data, BVP, and IBI, with the Empatica E4 wristband (Empatica, Milan, Italy) worn on their left arm. Physiological data were stored within the tablet that presented the questionnaire and synchronized with the EEG data via a time-to-live (TTL) signal generated at the time of response. Data from the EDA sensor are expressed in microsiemens, and the sampling rate was 4 Hz. Acceleration data were measured with a 3-axis ACC with a sampling rate of 32 Hz in units of $1/64 \times g$ (gravitational acceleration); the root mean square (RMS) of acceleration was calculated on the 3 axes as ACC. BVP data were obtained through photoplethysmography, and the sampling rate was 64 Hz[56]. IBI data were extracted from the BVP signal and expressed in seconds[57]. All physiological data were averaged in 1-min bins and then normalized for each day. One patient who wore the wearable device for under 2 days during the measurement period was not included in the analysis of behavioral data.

**Questionnaire.** Patients' mood and thought contents were measured by a tablet-based self-report questionnaire composed of 17 questions that evaluated their thought contents (14 questions) and mood (3 questions) (Table 1). The questionnaire was devised primarily to assess instances of mind wandering. The queries used to evaluate mind wandering were adapted from our previous report[17]. In each question, the patient was asked to rate how you feel or what you are thinking about right now (Table 1) on a continuum of internal states between 1 and 7. It should be noted that the patients were instructed to report their feelings and thoughts immediately before answering the questionnaire, not those while answering the questionnaire. Among the 17 questions in the questionnaire, the "External-task Focus" item was consistently presented as the first question. The remaining items were presented in a random order each time to reduce any potential order bias. All questions appeared on the display of the tablet hourly between 8:00 a.m. and 8:00 p.m. Patients were permitted to answer questions at any time. The questionnaire was designed so that it could be submitted only if all the questions were answered. The TTL signal was recorded on the EEG data at the moment the participant provided the response, and that moment was considered the response time.

When we instructed the patients to answer the 17 questions with the tablet system, we explained the meanings of the questions in detail with some examples. For example, to explain the item "External-task focus", we told the patients as following:

Sometimes when you are doing something, you may think about something other than what you are doing. For example, when you are watching TV, you may think about something other than what you are watching. If you were thinking about something completely different from what you are doing, mark this item with 1. On the other hand, if you were concentrating on what you are doing, mark 7.

The task in the question refers to what the patient was engaged in immediately before responding to the questionnaire, e.g., watching TV, not to the act of answering the questionnaire itself (Supplementary

Fig. 26). Additionally, for the item "unwanted," we instructed participants to answer 7 if the thought was something they did not want to think about and 0 if the thought was something they wanted to think about.

## Intracranial EEG

**Data processing.** We recorded intracranial EEG data from subdural electrodes and depth electrodes sampled at 10 kHz with an EEG-1200 (Nihon Kohden, Tokyo, Japan). A low-pass filter with a cutoff frequency of 3000 Hz and a high-pass filter with a cutoff frequency of 0.016 Hz were employed for signal preprocessing, and downsampling to 2 kHz was performed using an $8^{th}$-order Chebyshev Type I low-pass filter before resampling. Subdural contacts were arranged in both grid and strip configurations with intercontact spacing of 10 mm and a contact diameter of 3 mm. The intercontact spacing of the depth electrode contacts ranged from 5–15 mm, with a width of 1 mm. Electrode localization was accomplished by coregistering the postoperative CT data with the preoperative T1-weighted MRI data using SYNAPSE VINCENT (Fujifilm, Tokyo, Japan). The scanners used to obtain the MRI data were the SIGNA Architect (GE Healthcare) for four patients, and the Ingenia (Philips Healthcare, Amsterdam, Netherlands) for six patients. The CT images were obtained using the Discovery CT750 HD (GE Healthcare) for four patients and the Aquilion Precision (Toshiba Medical Systems) for six patients.

**Calculation of delta band power.** The amplitudes of delta waves were determined from electrocorticography data obtained from individuals with implanted cortical electrodes to analyze the correlation between sleep status and SWRs. Bipolar potentials were obtained by utilizing all available pairs of adjacent cortical electrodes. The power-line noise at 60, 120, 180, and 240 (±1.5) Hz was removed with a 1150-order finite impulse response (FIR) band stop filter. These potentials were filtered using a low-cut filter with a frequency of 0.5 Hz and a high-cut filter with a frequency of 4 Hz, and a Hilbert transform was applied. The Hilbert function was utilized to obtain the absolute value of the instantaneous amplitude of the analyzed signal. These computations were carried out using MATLAB 2017b.

**SWR detection.** We utilized the detection method published by ref. 23 because they reported a relationship between SWRs and visual imagery, a type of thought content (see Supplementary Fig. 20 for a comparison of our approach with other methods to detect SWRs). The LFP of hippocampal electrodes in the selected site was then converted to a bipolar signal. To remove power-line noise at 60, 120, 180, and 240 (±1.5) Hz, we used a 1150-order FIR bandpass filter. Next, the signals were filtered between 70 and 180 Hz using a zero-lag linear-phase Hamming-window FIR filter with a transition bandwidth of 5 Hz. The instantaneous amplitude was computed using the Hilbert transformation and clipped at 4 × the standard deviation (SD). Finally, the clipped signal was squared and passed through a low-pass filter before having the mean and SD calculated. Ripple candidates were selected from the range when the unclipped signal exceeded 4 × SD. Ripple candidates whose durations were shorter than 20 ms or longer than 200 ms were rejected, and peaks closer than 30 ms were concatenated (Fig. 2b). We performed spectral decomposition of SWR events at hippocampal recording sites using a time-frequency method, which was implemented in EEGLAB. For this process, we applied Fast Fourier Transforms and used Hanning window tapering across a frequency range of 4–200 Hz. We normalized the ripple-triggered spectrograms by calculating the geometric mean power at each frequency band, considering the entire epoch duration from −500 to 500 ms. Moreover, the frequency of adjacent SWR intervals was calculated for each patient and averaged across all patients. We manually reviewed and excluded data during epileptic activity and movement artifacts to confirm the SWR. In this process, epileptic

activity periods include electrographic seizures as well as pre- and post-seizure periods. This classification was performed manually by an epileptologist. The segments that were removed ranged from the normal area before seizure onset to the end of postictal slow wave activity following the seizure. Additionally, a 1-s buffer was added before and after the segments, and these periods were also excluded from the analysis.

## Standardization of data

The questionnaire and behavioral data were standardized over the entire recording period for each patient. The mean of all scores was subtracted from the individual scores, and the scores were then divided by the standard deviation of all scores. The diurnal variation in the SWR was evaluated according to the within-day standardized results. The SWRs to be explained by the questionnaire responses or behavioral data were standardized for all data of each patient.

## Scheduled patient periods during recording

We classified the 24-h period into several time periods based on the meal times and lighting of the patient's room. During the recording, all patients lived in the hospital room, so the times when meals were served and room lights were switched off and on were determined by hospital regulations. In this study, nighttime was defined as the period from 22:00 to 6:59. Daytime was defined as the period from 7:00 to 21:59. This definition was based on the hospital schedule, with the room light turned off at 22:00 and on at 7:00. Similarly, mealtime was defined as 7:00–8:00, 12:00–13:00 and 18:00-19:00 based on hospital regulations. The exact times of the start and end of each meal were assessed by video monitoring if possible (Supplementary Tables 6–8). In addition, we divided the daytime into morning and afternoon periods by excluding mealtimes. The morning was defined as the period from 8:00 to 11:59, and the afternoon was defined as the period from 13:00 to 17:59.

## General linear mixed-effects model

General linear mixed-effects models were constructed to evaluate the occurrence of SWR events based on biometric data and the answers to the self-report questionnaire. For regression analysis, several data points were removed due to lack of SWR data. First, the SWR events were explained by a general linear mixed-effects model using the Z-scored values of four types of biometric data (EDA, ACC, BVP, and IBI obtained with the wearable device). The Z-score of each value was determined over the entire period of each patient. To capture potential variability in SWR events due to individual differences among patients, we incorporated a random intercept for each patient in our model. To assess the temporal relationship between SWR events and biometric data, we evaluated the model for SWR events in different time intervals before and after SWR events to acquire biometric data. Within these time intervals, we averaged the SWR events in 1-min segments at 1-min gaps from 15 min before to 5 min after data acquisition. The SWR events in each time segment were fitted by general linear mixed-effects models. The time interval with the highest coefficient of determination was examined, and the coefficients of each biometric value were compared. Next, we employed a second mixed-effects model with a linear regression approach to evaluate the mean SWR rate over periods of 5 min from 15 min before to 5 min after the response time, as inferred according to the answers to the self-report questionnaire. To capture potential variability in SWR events due to individual differences among patients, we incorporated a random intercept for each patient in our model. The regression models were evaluated by the coefficient of determination, adjusted for degrees of freedom. The time interval with the highest coefficient of determination was examined, and the coefficients of each questionnaire score were compared. Statistical significance was analyzed with 1000

permutation tests using the collected data, and the time order of the acquired data (both the biometric data and questionnaire scores) was randomized.

## Statistical analysis

Unless otherwise specified, significant differences were assessed using one-way ANOVA, and Pearson correlation analysis (for parametric statistics) or Spearman correlation analysis (for nonparametric statistics) was used to evaluate correlations. We reported Bayes factors as outcomes of hypothesis tests to quantify the support for the true null hypothesis in the analysis of the correlation between the SWR rate and cortical delta band power. This study used observational data, and data collection and analysis were performed using MATLAB 2017b. We calculated that the target sample size for performing a regression analysis of the SWR event rate based on the 17 questionnaire items was 170, which is 10 times the number of questionnaire items. The ratio of sample size to the number of parameters in the regression analysis was determined based on previous studies[58–60]. The 170 target sample size was not per patient but was combined across patients. The data gathered for each patient were influenced by their personalized treatment plan, and the duration each participant was prepared to voluntarily contribute to the study.

## Reporting summary

Further information on research design is available in the Nature Portfolio Reporting Summary linked to this article.

## Data availability

The raw LFP data are available under restricted access as they contain information that could compromise the patients' privacy, and consent for their publication was not obtained. However, access can be obtained by reasonable request to the corresponding author, T.Y. The SWR event rates, cortical delta power, and thought sampling data generated and analyzed in the present study have been deposited in Figshare at https://doi.org/10.6084/m9.figshare.22633369.v1. The remainder of the data shown in the figures are provided in the Source Data file. Source data are provided with this paper.

## Code availability

The code written to detect SWRs is available at https://doi.org/10.6084/m9.figshare.22815746.v1.

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

## Acknowledgements

We thank Dr. Satoru Saito for his help in preparing the questionnaire. This research was conducted under the Japan Science and Technology Agency (JST) Exploratory Research for Advanced Technology (JPMJER1801, T.Y. and Y.I.). This research was also supported in part by the Core Research for Evolutional Science and Technology (JPMJCR18A5, T.Y.) and Moonshot R&D (JPMJMS2012, T.Y.), Grants-in-Aid for Scientific Research from KAKENHI (20H05705, T.Y., 22K15623, R.F. and 22K21353, Y.I.), the Institute for AI and Beyond of the University of Tokyo (Y.I) and grants from the Japan Agency for Medical Research and Development (AMED, 19dm0307008, T.Y. and AMED CREST, 22gm1510002h0002, Y.I). H.Ki. was supported by grants from AMED (19dm0207070 and 19dm0307103). J.S. was supported by a Discovery Grant from the National Engineering and Science Council of Canada, as well as an award from the New Frontiers in Research Fund.

## Author contributions

T.I. contributed to the study design, data collection, data coding, data analyses, and writing of the manuscript. T.Y. contributed to the study design, data collection, data analyses, and writing of the manuscript and editing. Y.I. contributed to manuscript editing, and study design. J.S. contributed to manuscript editing and study design. R.F. contributed to data coding and data analyses. S.O., N.T. and H.Kh. contributed to data collection. H.Ki. contributed to data collection and qualitative data analyses.

## Competing interests

The authors declare no competing interests.
