## [Peer Review File · Nature Communications]

REVIEWER COMMENTS

Reviewer #1 (Remarks to the Author):

General comment:

In this study, Iwata and colleagues investigated the relationship between sharp waves and ripples (SWRs) in the human hippocampus and reports of naturally occurring self-generated thoughts. Since SWRs are fast events generated in a deep brain region, they cannot be captured with non-invasive recording and the authors relied on intracranial recording in implanted epileptic patients. The main findings are the fact that (1) SWRs are modulated according to the time of the day and activity of individuals, (2) SWRs are correlated with the occurrence of intrusive ongoing thoughts. I would like to stress how important these findings are for our understanding of spontaneous conscious experiences in humans and animals. To my knowledge, this study brings the first evidence for a long-hypothesized link between neural processes typically associated with action planning and memory consolidation (such as SWRs) and spontaneous experiences. This work represents therefore a critical piece of empirical evidence. I have a few comments regarding the methodology.

Major comments:

1. The authors mention that the number of detected SWRs was extracted for each minute of recordings and z-standardised over each day, removing thus the variance between days and subjects. Later, regarding thought probes, they also wrote that “the responses were standardized for each participant.” I assume here that the standardization was also performed per day, but it would be worth making this explicit.

I also assume this standardization was done to focus on the within-day variability in terms of SWR rate and modulations of spontaneous experiences. But it would be also interesting to explore the variance between days and subjects. For example, are the patients having the most intrusive spontaneous thoughts also showing the most SWR? Explaining the between-subject variance in the occurrence of intrusive thoughts could be very significant for clinical applications (e.g., to account for disorders of attention for example).

2. The authors report a positive relationship between SWRs and delta power during the night, in link with previous accounts of a tight relationship between slow oscillations and SWR (e.g., Mölle et al. *Journal of Neurophysiology* 2006; a paper worth citing). However, more than just a correlation between delta power and SWRs, SWRs are nested in slow oscillations (i.e., there is a phase-specific relationship). It could be worth checking that this relationship is also observed here during sleep and whether it is also present in wakefulness since the authors found a positive correlation between delta power and SWRs in wakefulness as well. Finding a phase-specific relationship between slow waves and SWRs in both wakefulness and sleep would be an extremely interesting finding per se. Besides, slow waves have been

associated with spontaneous experiences non-invasive brain recording (e.g., Perogamvros et al. Journal of Cognitive Neuroscience 2017, Andrillon et al. Nature Communications 2021). Of course, the authors are free to choose not to implement this additional analysis but, in my opinion, it represents a nice opportunity to bridge a set of converging evidence with a unifying model.

3. The authors report a drop in SWR during mealtimes. To check that this drop is not due to an increase in artefacts for example, did the authors observe an increase in the amount of rejected SWR during the same periods? Is the spectrum of hippocampal activity different during meals vs. after or before? This could be particularly relevant because of the possible link between epileptic seizures and food intake (e.g., Tényi et al. Scientific Reports 2021). Were epileptic artefacts also modulated by mealtimes but in an opposite direction?

Also, it seems that the SWR drops 0.5-1h before the mealtime. Is this because mealtimes are approximate, and participants could start eating a bit before? Would it be possible to narrow down the temporal relationship between food intake and SWR rates by aligning SWR rate to the exact onset of each meal? This would be quite important for the authors' interpretation of SWR rate modulations with blood glucose levels (see Discussion).

4. I apologize for the perhaps naïve remark but isn't the following sentence trivial? "The predicted SWR rates were weakly but significantly correlated with the actual SWR rates ($R = 0.21$, $P = 4.3 \times 10^{-10}$, $n = 2,027$ time points) (Fig. 3b)". Indeed, I am wondering if the fact that the predicted SWR rate correlates with the actual SWR rate does not just mean that the modeling is not completely at chance. I think the authors should, in addition to this, check if the coefficients of the models are significantly different from 0 (they have one coefficient per fold). They could also show not only the average but individual values of the coefficients in figure 3 and 4).

5. When examining the relationship between specific dimensions of thoughts and feelings and SWRs, the authors examine the correlations across all bins. Maybe it would be better to fit mixed-models to take into consideration subject identify (unless this is already done through the z-scoring)?

6. When listing the limitation of their studies, the authors could also mention that the temporal link found between SWR rate and intrusive thoughts is quite broad (positive correlation when extracting the SWR rate over 2-7 minutes before thought probes). Perhaps the authors could examine this in further details, for example by computing the correlation coefficient between thought probes and SW rate on windows of 1 minute locked to probe onset. A more specific temporal relationship would represent stronger evidence in favor of a mechanistic link between SWR and intrusive thoughts.

Finally, can the authors provide details about the period used to predict SWR rates based on the reports from participants? They seem to mention only a period of 2 to 7 minutes.

7. It seems that there are no robust modulation of body and physiological activity (EDA, ACC, BVP) across the day even though these variables should be significantly impacted by sleep wake cycles. Can the authors report in Figure 3, the probability of being asleep as a comparison? Were these variables impacted by vigilance states?

8. Reports of thoughts and feelings also do not seem modulated across the day according to Figure 4a, suggesting there is little variance to explain. Perhaps the authors could show the z-scored SWR rates for the same period for comparison?

Besides, the figure appears at odd with what is reported in the result section: “Before the data were analyzed, the responses were standardized for each participant. The analysis revealed that only the spontaneity of cognition significantly varied throughout the day, with the highest levels observed in the mid-afternoon and the lowest levels in the early evening ($P = 0.033$, $F_{13,267} = 1.88$; one-way ANOVA)”. Were the tests across the four dimensions corrected here for multiple comparisons?

Finally, the data used for the correlation analysis in Figure 4b does not seem to be normally distributed, but the authors used a Pearson t-test. Maybe a non-parametric test would be better here.

9. The authors used a cross-validation for their linear model, but they write that “In the cross-validation, all answers to questionnaires were randomly divided into 10 groups for training and testing”. This suggests that the training and testing sets for each validation were including overlapping subjects which could lead to over-fitting. Do the results replicate when populating the training and testing sets with different groups of subjects?

Minor comments:

1. The links for the data and code do not seem to exist. I assume this is because the authors intend to make the data and code available upon publication, but I thought to mention this for the avoidance of doubt.

2. “The correlations were similar during the day and night (Fig. 2e; $R = 0.38 \pm 0.25$ for daytime and 0.36 ± 0.19 for nighttime; $t_5 = 0.2993$; $P = 0.7768$; paired t test).” Instead of using a paired t-test to test the similarity of two variables (in other words, to test H_0), the authors could rely on Bayesian statistics.

3. When examining the relationship between SWE and delta power (Figure 2d), the authors could apply a mixed model with a random factor of subject to check if the relationship generalizes across subjects rather than reporting a correlation coefficient extracted across time bins. Besides, if the SWR rates in Figure 2d do not seem z-transformed (data not centered around 0) contrary to what is mentioned on page 5. Is this normal? Were the correlations reported on page 5 obtained with non z-transformed SWR rates? If so, why switching after for z-scored data?

Reviewer #2 (Remarks to the Author):

In this submission, Iwata et al. examine whether sharp-wave ripple (SWR) events, which have been linked to memory replay in rodents and humans, undergo circadian fluctuation and whether higher SWR rates may be a signature of intrusive thoughts and internally generated patterns in humans with epilepsy. These findings are potentially interesting because the authors claim their findings show that SWRs may provide an electrophysiological signature of internally generated thoughts in humans; given the relation to rodent literature, it would be interesting if a human analogue were shown to be true.

While the findings presented in this manuscript (if true) are potentially interesting, I have a number of concerns that greatly diminish the impact of the findings. First, the physiological interpretation of the SWR events detected by the authors is not clear, particularly in the context of the epilepsy population employed in the study. Methodological details are lacking to have sufficient information about whether the authors have reasonably excluded pathological ripples, or whether events studied are actually a combination of physiological and pathological ripples (and if the latter is true, how this relates to rodent literature, which is mainly related to physiological ripples, would be unclear). Second, the behavioral study design is not appropriately set up to answer whether there is an association between external task focus and SWR rate; there is no “task” that patients are performing and, moreover, the authors examine the association between SWRs and a subsequent click on a tablet two to seven minutes later, rather than any task. Third, independent of the concerns about study design and bigger picture interpretation, I have concerns that the significance of any identified results is overstated. While the first and third concerns might be addressed through a major revision, the second concern cannot be addressed without a restructuring of the fundamental behavioral experiment, and therefore I unfortunately cannot recommend this paper for revision outside of a new submission employing a new behavioral paradigm.

Below I have outlined specific suggestions I have for the authors under each concern.

1. It is not clear whether the inclusion/exclusion criteria and data preprocessing yield an interpretation of their results in the context of physiological ripples, or a combination of physiological and pathological ripples: (a) Patients with hippocampal sclerosis were excluded from analysis, and I presume this was to reduce contamination of SWRs with pathological ripples which would be more abundant in the epileptogenic zone. However, there is no information about whether the patients who were included in the sample could be reasonably determined not to have epileptogenic zones coinciding with the placement of the depth electrodes. Were all the other cases non-lesional on all other imaging sources (for example, which there no hippocampal atrophy and was PET symmetric) and did the patients included have no seizures localized to the hippocampus either on phase 1 or phase 2 EEG monitoring? (b) The authors mention that periods of “intense epileptic activity” were excluded from analysis, but it is not clear what “intense” activity means, as this could be anything from only excluding seizures to excluding seizures and interictal epileptiform discharges (IEDs) and HFOs. Please expand on how “intense” epileptic activity was defined, and how the start/end of the segments around this activity were removed. Due to the relationship of IEDs with overriding pathological ripple activity and HFOs as pathological ripples, this is important to help me and future readers to understand how likely the results are to reflect physiological ripples versus physiological and pathological ripples. (Similarly, in the Results section, please define how “epileptic activity” was defined.)

2. Behavioral study design: This is perhaps my greatest fundamental concern with the paper. The major claim of the paper is that higher SWR rate is associated with less external task focus and more intrusive thoughts. However, there is no specific “task” that the participants are engaged in, so the idea of “task

focus” does not particularly make sense. Specifically, the subjects are asked every hour to fill out a survey that is used to assess the presence of “intrusive thoughts”. The authors take the SWR rate 2 to 7 minutes before each response to a question on this survey as the rate that is related to the survey response. There are two problems with this. First, the SWR rate measurements may not be occurring at the same time as the intrusive thoughts. In fact, the data might indicate that intrusive thoughts correlate with some brain state that is preceded by high SWR rates. Second, there is no task being performed during the times of SWR rate measurements. Thus, the authors ability to make a clear connection between SWR rates, task performance, and intrusive thoughts is limited.

3. The concerns I have about overstatement of the significance of results relate to the fact that results appear to be overstated in several instances and with no attention to checking how robust their results are to analysis choices that are made.

- The authors base their results on a single method of detecting SWRs, i.e., that proposed by Norman et al. (2019). However, it is known that detected SWR rates can vary several fold depending on the detection method used, see for example Liu et al. (2022). To ensure that these results are not sensitive to SWR detection method, I would advise using multiple SWR detection methods, such as that by Vaz et al. (2019) and Staresina et al. (2015). Consistency across multiple detection methods and methods of analysis would increase the likelihood of this being a true result.

- In the analysis to examine whether SWR rates exhibits circadian fluctuation, some unknown thresholds are applied to time of day to define different categories of time of day, and it is not clear what these thresholds are to define their categories, or how generalizable the results would be should the thresholds have been chosen (for example) to produce statistical significance. Since circadian rhythms are inherently angular in nature, it would be much more natural to use a more data driven approach with circular statistics to quantify this association -- for example, reporting the distribution of the mean resultant vector across patients would tell the reader the strength how concentrated the circular distribution of SWRs is around specific times of day.

- The authors report that there were transient drops in SWR rates around 7:00, 12:00, and 18:00, coinciding with scheduled meal times in the hospital. This could be potentially interesting given the contrast to rodent literature about increases in SWR rate occurring around reward in rodents. However, how were the transient drops in SWR rates at 7:00, 12:00, and 18:00 actually estimated, and was this present in all patients? Currently, there are no actual statistics provided to quantify this claim, except for a figure qualitatively showing this occurring for a single patient.

- For the section on correlation between SWR rates and biophysical parameters, the authors need to be very clear that their results, while statistically significant, indicate that physiological measures capture only a very small fraction of the variance in SWR rates. Ideally, they would report R^2 values, as these are directly interpretable in terms of variance explained. For example, it seems that the authors have found that the coefficient of determination for a linear regression of SWR rate on time of day and four physiological measurements is $0.21^2 = 0.04$.

- Of these four physiological measurements, it seems that BVP and IBI (which are both proxy measures of heart rate) explain the greatest proportion of variance, but both have small effect sizes ($-0.007^2 = 4.9e-$

5 for BVP and $0.15^2 = 0.02$ for IBI). Therefore, from what I can see, a more accurate interpretation of this finding would be that the four physiological measures acquired from the Empatica wristwatch explain only a very small fraction of the variance in SWR rates.

- On a related note to the above, although it is technically valid to report the correlation between predicted and observed SWR rates (i.e. the square root of the coefficient of determination), it is standard in regression analysis to report R^2 (rather than its square root) as a measure of model fit, as it has straightforward interpretations with regards to the information about the proportion of variance explained by the model. Especially because the p-values can be misleading in the setting of large sample sizes, I would suggest reporting R^2 rather than its square root when reporting model fit.

- Linear regression is used to estimate the association between the self-report scale and SWR rate, which does not account for the repeated measures nature of the study design. The rigor of the study would be greatly improved if the authors used repeated measures mixed effects analysis instead.

- The coefficients from regression of SWR rate on the self-report thought measures are not transparently reported. Rather than showing a word cloud, I would suggest reporting all estimated regression coefficients along with standard error and p-values. Word clouds can visually distort the magnitude of differences between regression coefficients and do not adequately represent estimates from regression.

- It seems from the word cloud that other thought variables, such as “happy,” “person,” “images,” and “excited,” were also large in magnitude. However, whether these were or were not statistically significant is not disclosed. Were any other regression coefficients statistically significant? If so, please discuss how this would affect interpretation.

- Throughout the manuscript, p-values are reported for all hypothesis tests. However, given the large number of time points included in many of these hypothesis tests, p-values are not as meaningful and I would suggest the authors comment not only on statistical significance but also magnitude of effect size.

4. Last, various aspects of the methods are not clear and need to be clarified.

- Regarding classification of sleep/wake states using delta power: It is not clear from the current description what supervised or unsupervised learning method was used to classify these two states based on delta band power. Please add this to the methods.

- Related to the above, it seems that sleep/wake state classification was based solely on delta band power. Given the difficulty of sleep staging with intracranial EEG (especially as compared to scalp EEG): (1) How were the labels validated? (2) Given that video and accelerometry data were also available, it seems it would have been reasonable use these either to validate the labels, or as additional features into the classification algorithm. Was this considered? (3) Was subdermal EEG or scalp EEG, along with EOG/EMG electrodes, available for sleep staging for these patients? Given that the authors have accelerometry data, this could be presumably also used for classification of sleep/wake states as well as so immobile and mobile wakeful states.

- Missing data usually affects self-report measures and its treatment can affect results. Please describe how the degree of missingness and how missing data was treated in the self-report scale, particularly as the association of SWR rate with self-report measures is one of the main findings of the paper.

- On a related note, how were these 17 questions distributed, i.e., were they shown in random order, or (as is suggested in the table) were they always shown in the same order? If they were shown always in the same order, then order bias is a concern.

- Is the correlation of 0.74 between delta amplitude and SWR rate across all patients, or just for the single representative patient shown in the figure? If the former, it seems a linear mixed effects model would be more appropriate; if the latter, please report for the group of patients.

- The authors report results from a one-way ANOVA comparing SWR rates during sleep to daytime. (1) I am curious why sleep was compared to daytime, rather than comparing sleep to wake, or daytime to nighttime. Could the authors please clarify? (2) If daytime to nighttime was compared, how were these defined? (3) It is also not clear to me what three groups are being compared using the one-way ANOVA (it seems there are only two – sleep and daytime – from the text).

- For the finding comparing SWR rates between sleep and daytime, please report the mean/standard deviation of ripple rates.

- It is not clear to me what the authors mean by “To conduct the regression analysis, we predetermined the total number of answers on questionnaires as more than 170 samples, which is equivalent to 10 times the number of questionnaire items, totaling 17.” Does this mean that if the patient filled out more than 10 surveys (which would seemingly correspond to 10 hours only), any additional responses were discarded? Also, could the authors please clarify how the predetermined value of 170 samples relates to how they conducted the regression analysis?

- Anti-seizure medicines (ASMs) may affect ripple rates, and are often withdrawn, held, or restarted during varying parts of EMU admissions. Were changes in SWR rates associated with timing of ASM administration (including the changes observed at mealtimes), and is any of the variability in SWR rate identified with ASM administration?

- Regarding the linear regression analysis for thought content: Given that the 17 questions in the survey seem to be collinear, does this inflate the variance of the regression coefficients and is there a need for regularization?

5. Final (minor) suggestions:

- In Supplementary Table S1, please report where the seizure focus was for each of the 11 patients, and the results of available of their MRI (lesional/non-lesional), seizure semiology, MEG (if available), PET (if available), and scalp EEG.

- I wonder if the term “freely behaving humans” is appropriate, given that people are not really freely behaving in EMU admissions and these settings typically are not thought to reflect the individual’s natural environment (as the authors allude to in their limitations section). Moreover, ASMs are often withdrawn or held during these admissions, which changes the rate of interictal epileptiform discharges and may affect SWR rates as well as relative proportions of pathological and physiological ripples. It would be more accurate to describe what the environment in this study captures, which is SWRs during rest and wake during a controlled environment, or use the term “freely moving humans” (although this is

also probably not true, since they are usually tied to a hospital monitor via their EEG electrodes and/or telemetry and IVs).

- Methods, Participants: "...to treat epilepsy..." → "for presurgical localization" (?)
- Throughout the manuscript, "deep electrodes" → "depth electrodes"
- In the results and discussion sections, the authors mention that their findings "establish" that SWRs are related to patterns of ongoing thought; a word such as "suggest" might be more appropriate.
- It is not clear which measure "desired experiences" corresponds to in the scale. Is this the "unwilling" dimension?
- I would suggest the authors provide additional context to the significance of any findings with regards to existing literature about non-local representations and relationship to SWRs during locomotion in rodents. It would also be interesting if the authors could report if there are characteristics of the SWRs that were more associated with local thought content (high task focus?) than with non-local thought content (low task focus?).
- Data sharing statement: I suggest that authors consider sharing their data using NWB file format on DANDI.

Reviewer #3 (Remarks to the Author):

In 11 individuals post epilepsy surgery with electrodes implanted in the hippocampus or parahippocampal gyrus, the authors measure the occurrence of sharp wave ripple (SWR) events and their relationship to ongoing thought patterns, ongoing over the course of 9 to 11 days. They reveal changes in SWR frequency across the day-night cycle, and across the course of the day. They further show that SWR rates were associated with responses on a thought questionnaire, which was probed hourly. In particular, the thought dimensions related to SWR rates involved less external task focus and vivid mental content – supporting emerging hypotheses that SWRs may be one avenue to a brain state that supports internally directed thought, or mind wandering.

This is a compelling study that further extends our understanding of SWR occurrence into the human domain, and most notably links them with ongoing behaviour and thought patterns. I have the following comments:

- Prior to detecting SWRs in the data, the authors excluded periods associated with epileptic activity. This issue is considered as a limitation in the Discussion, as the authors explain that although they've excluded the effects of epileptic activity, it remains unclear whether epileptic activity might have influenced hippocampal activity or led to changes in ongoing thought patterns.

The possible influence of epileptic activity is of course inherent in this study population. However, given the nature of the study question I wonder if it deserves more careful attention. I am thinking of the possibility that not only might epileptic activity influence the hippocampal EEG signal, but epileptic activity might have subjective (conscious) effects on arousal, mood or thought content – which may relate directly to the dimensions captured in the questionnaire.

Perhaps some data might speak to this issue. Have the authors considered looking at whether the timings of periods of epileptic activity 1) were associated with subsequent increases/decreases in SWR frequency; and 2) if epileptic activity occurred in the 2 to 7 min window preceding questionnaire responses, whether this was associated with distinct thought characteristics?

- Regarding SWR detection: ripples with durations shorter than 20 ms or longer than 200 ms were rejected, and peaks closer than 30 ms were concatenated. I wondered that given their rich dataset, whether the authors looked in more detail at characteristics of SWR events.

Based on rodent literature, we know that longer duration of ripple events has implications for memory performance (e.g., *Science*, 364(6445), 1082-1086 <https://www.science.org/doi/10.1126/science.aax0758>). There is also the idea that ripple bursts or clusters (i.e., temporally close trains of events) might have a distinct functional relevance, as compared to isolated events (e.g., see reviewed in *Hippocampus*, 25(10), 1073-1188 <https://onlinelibrary.wiley.com/doi/full/10.1002/hipo.22488>). Did the authors explore whether 1) duration of SWRs; or 2) clustered vs. isolated SWRs, had distinct relationship with the behavioural or thought patterns they investigated? This seems like it might be useful information to both enrich their findings and make even closer links with the rodent literature.

- The authors raise the point in the Discussion that neuromodulators are likely to be an important determinant of the SWR brain state. It made me wonder whether the study participants were on any postoperative medications that might be relevant to consider regarding SWR modulation. I didn't find any mention of current medications for the study participants, perhaps this information is worth including?

- On the second page of the Discussion, the authors relate their findings to hippocampal lesions and reductions in mind wandering, as well as grey matter in the hippocampus of healthy people. It may be of

interest that further convergent evidence in support of the link between hippocampal integrity and mind wandering/off-task thought has also been shown in dementia, where grey matter integrity in the hippocampus/parahippocampus has been linked with capacity for mind wandering (PNAS 116(8), 3316-3321 <https://www.pnas.org/doi/abs/10.1073/pnas.1818523116>).

- A small point in the Introduction: SWRs are defined as “bursts of synchronized neuronal activity in the mammalian hippocampus”. Fascinatingly, these rhythms appear to be conserved outside of the mammalian brain, with evidence for them in lizards (Shein-Idelson et al. <https://www.science.org/doi/abs/10.1126/science.aaf3621>) and in birds (Yeganegi et al. <https://www.biorxiv.org/content/10.1101/825075v2.abstract>). So this statement could be amended.

Point-by-Point Replies for Reviewers

Hippocampal sharp-wave ripples correlate with naturally occurring self-generated thoughts in humans

(NCOMMS-23-27587-T)

We are grateful for the reviewers' insightful feedback and suggestions. The constructive critiques have significantly aided in refining and enhancing our manuscript. We have diligently addressed each concern raised, and in the revised manuscript, all additions and modifications are highlighted in yellow. Below, we provide a detailed response to each comment, with the reviewer's remarks presented in gray. In addition, one patient included in the initial analysis underwent surgery after submission of the paper and was diagnosed with hippocampal sclerosis, and the data of this patient were excluded in the revised version.

Response to Reviewers

Reviewer #1 (Remarks to the Author):

General comment:

In this study, Iwata and colleagues investigated the relationship between sharp waves and ripples (SWRs) in the human hippocampus and reports of naturally occurring self-generated thoughts. Since SWRs are fast events generated in a deep brain region, they cannot be captured with non-invasive recording and the authors relied on intracranial recording in implanted epileptic patients. The main findings are the fact that (1) SWRs are modulated according to the time of the day and activity of individuals, (2) SWRs are correlated with the occurrence of intrusive ongoing thoughts. I would like to stress how important these findings are for our understanding of spontaneous conscious experiences in humans and animals. To my knowledge, this study brings the first evidence for a long-hypothesized link between neural processes typically associated with action planning and memory consolidation (such as SWRs) and spontaneous experiences. This work represents therefore a critical piece of empirical evidence. I have a few comments regarding the methodology.

Response to the comments:

We would like to extend our sincere gratitude to the reviewer for their insightful feedback and recognition of the significance of our study. We are also grateful for the reviewer's acknowledgment of the significance of our findings in elucidating self-generated thought processes in humans.

Regarding the reviewer's concerns about our study, we would like to address them as follows:

Major comments:

Comment 1

Comment 1-1

The authors mention that the number of detected SWRs was extracted for each minute of recordings and z-standardised over each day, removing thus the variance between days and subjects. Later, regarding thought probes, they also wrote that "the responses were standardized for each participant." I assume here that the standardization was also performed per day, but it would be worth making this explicit.

Comment 1-2

I also assume this standardization was done to focus on the within-day variability in terms of SWR rate and modulations of spontaneous experiences. But it would be also interesting to explore the variance between days and subjects. For example, are the patients having the most intrusive spontaneous thoughts also showing the most SWR? Explaining the between-subject variance in the occurrence of intrusive thoughts could be very significant for clinical applications (e.g., to account for disorders of attention for example).

Response to comment 1-1:

We thank the reviewer for their constructive comments on our manuscript. Our responses to the reviewer's comments regarding the standardization of the SWR rates and thought probes are included below.

First, we apologize for the unclear descriptions of the method used to standardize the responses to the questionnaires. The questionnaire responses were standardized over the entire recording period for each patient, not per day for each patient. We believe that each subject would respond to the questionnaire based on their own standard, which does not change during the recording period. Therefore, we normalized the questionnaire responses based on all data for each subject. In addition, as the number of responses differed across days, standardizing the data for each day would result in considerable bias. On the other hand, the diurnal variations in the SWRs were evaluated with the within-day standardization approach, as noted by Reviewer #1. Notably, the SWRs to be explained by the questionnaire responses or behavioral data were standardized for all data of each patient. To clarify this point, we updated the Results and Methods sections as follows:

Line 140 in page 8 (Results)

We examined whether the SWR rates were also associated with changes in the activities and autonomic states of patients. Specifically, the EDA, ACC, BVP, and IBI data of the patients were Z-standardized among each patient, as for SWR, and then averaged across patients (Fig. 3a).

Line 162 in page 9 (Results)

We used a generalized linear mixed-effects model to investigate the Z-standardized SWR rates averaged over 5 minutes, shifting every 1 minute from 15 minutes before to 5 minutes after answering the questionnaire based on the standardized values of the responses to each question (Fig. 4b). The mean SWR rates from -5 minutes to 0 minutes were strongly explained by the answers with the highest accuracy above the chance level ($R^2 = 0.576$, $n = 187$ questions, $P < 0.05$, permutation test, Bonferroni corrected).

Line 429 in page 21 (Methods)

Standardization of data

The questionnaire and behavioral data were standardized over the entire recording period for each patient. The mean of all scores was subtracted from the individual scores, and the scores were then divided by the standard deviation of all scores. The diurnal variation in the SWR was evaluated according to the within-day standardized results. The SWRs to be explained by the questionnaire responses or behavioral data were standardized for all data of each patient.

Response to comment 1-2:

We sincerely appreciate this comment from Reviewer #1. Based on the suggestion from Reviewer #1, we assessed the relationship between SWR rates and the occurrence of intrusive thoughts among patients. We calculated the mean and standard deviation (SD) of the daytime SWR event rate (/sec) (from 8:00 to 22:00) and task focus score for each patient. However, there was no significant correlation between the daytime SWR event rate and task focus score. Because each patient answered the questionnaire based on their own standards, it was difficult to assess the occurrence of intrusive thoughts among different patients. These data were included in Supplementary Figure 6 and referred to as the "**Correlations of SWR rates with thoughts and feelings**" in the Results (line 156 Page 9) in the revised manuscript to offer additional insights into the intersubject variability.

Supplementary Figure 6

Plot of mean and standard deviation of the SWR event rate (/s) during the day from 7:00 to 21:59 and the responses to the “External-task Focus” responses of each patient ($n=10$). The SWR event rate was not normalized.

Comment 2

The authors report a positive relationship between SWRs and delta power during the night, in link with previous accounts of a tight relationship between slow oscillations and SWR (e.g., Mölle et al. *Journal of Neurophysiology* 2006; a paper worth citing). However, more than just a correlation between delta power and SWRs, SWRs are nested in slow oscillations (i.e., there is a phase-specific relationship). It could be worth checking that this relationship is also observed here during sleep and whether it is also present in wakefulness since the authors found a positive correlation between delta power and SWRs in wakefulness as well. Finding a phase-specific relationship between slow waves and SWRs in both wakefulness and sleep would be an extremely interesting finding per se. Besides, slow waves have been associated with spontaneous experiences non-invasive brain recording (e.g., Perogamvros et al. *Journal of Cognitive Neuroscience* 2017, Andrillon et al. *Nature Communications* 2021). Of course, the authors are free to choose not to implement this additional analysis but, in my opinion, it represents a nice opportunity to bridge a set of converging evidence with a unifying model.

Response to comment 2:

First, we appreciate the opportunity to improve the manuscript and believe that the additional analyses we have performed in response to the comments significantly strengthen our conclusions.

In accordance with the reviewer's suggestion to explore the phase-specific relationship between slow oscillations and SWRs, we conducted additional analyses. To directly investigate this relationship, the average waveforms of all cortical EEG electrodes were generated 1 second before and after the SWR peak during both daytime (from 7:00 to 21:59) and nighttime (22:00 to 6:59) for 6 patients who were implanted with cortical electrodes. In both daytime and nighttime, slow waves in the delta range were observed in the cortical ECoG data at the time of SWRs (Supplementary Figure 7). Consistent with previous studies, the trough of the cortical slow wave was coupled with that of the hippocampal SWR¹. These analyses have been added to Supplementary Figure 7 and integrated into the Discussion by citing three relevant papers to contextualize these findings¹⁻³.

1. Mölle, M., Yeshenko, O., Marshall, L., Sara, S. J. & Born, J. Hippocampal Sharp Wave-Ripples Linked to Slow Oscillations in Rat Slow-Wave Sleep. *J Neurophysiol* 96, 62–70 (2006).
2. Perogamvros, L. et al. The Phenomenal Contents and Neural Correlates of Spontaneous Thoughts across Wakefulness, NREM Sleep, and REM Sleep. *J Cogn Neurosci* 29, 1766–1777 (2017).
3. Andrillon, T., Burns, A., Mackay, T., Windt, J. & Tsuchiya, N. Predicting lapses of attention with sleep-like slow waves. *Nat Commun* 12, 3657 (2021).

Supplementary Figure 7

The average waveforms of all cortical EEG electrodes were generated 1 second before and after the peak SWR during both daytime (from 7:00 to 22:00, a) and nighttime (22:00 to 7:00, b) for 6 patients who were implanted with cortical electrodes. (c) The average daytime and nighttime waveforms of 6 patients.

Furthermore, we conducted a regression analysis of the SWR rate after day 4 based on the power in each frequency band (δ : 0.5-4 Hz, θ : 4-8 Hz, α : 8-12 Hz, β : 12-30 Hz, γ : 30-60 Hz) in a representative case (see Supplementary Figure 9). We found that the predictive model was more accurate at night, with the coefficient of the delta wave contributing positively to the prediction during both daytime and nighttime. Interestingly, the beta and alpha bands also contributed to the prediction at night. Based on the reviewer's interest in the broader implications of our work, we have expanded the Discussion to

include that noninvasive EEG can also be used to measure these frequency bands. Specifically, the relationship between the delta wave and SWR can be a useful predictor of SWR conditions during noninvasive measurements. This information has been added to the revised manuscript in the Discussion as follows.

Fourth paragraph of Discussion (Line 230 page 12)

Although our present evaluation of SWRs relies on invasive recording techniques, prior resting-state studies have shown that the medial temporal lobe is related to intrusive thoughts⁴⁴. Moreover, SWRs and delta power are strongly correlated⁵⁰ (Supplementary Figure 7), and data waves can be recorded by noninvasive EEG, enabling the prediction of features related to spontaneous experiences^{51,52} (see Supplementary Figure 9). In the future, we could combine brain imaging approaches with task-based methods and experience sampling to better understand the neural basis of intrusive thinking in different clinical states.

Supplementary Figure 9

The normalized SWR event rates were explained by five frequency band amplitudes (theta, 4-8 Hz; alpha, 8-13 Hz; beta, 13-25 Hz; gamma, 80-150 Hz) using a linear regression model for a representative case (Pt-04). (a) The actual and predicted normalized SWR event rates during the daytime were plotted with the linear regression model ($R^2=0.44 \pm 0.01$, correlation coefficient between the actual and predicted normalized SWR event rates was 0.21). (b) The coefficients of the trained model for the daytime. (c) The actual and predicted normalized SWR event rates during the night were plotted with the linear regression model ($R^2=0.48 \pm 0.04$, correlation coefficients between actual and predicted normalized SWR event rates was 0.67). (d) The coefficients of the trained model for the night.

Comment 3

Comment 3-1

The authors report a drop in SWR during mealtimes. To check that this drop is not due to an increase in artefacts for example, did the authors observe an increase in the amount of rejected SWR during the same periods? Is the spectrum of hippocampal activity different during meals vs. after or before? This could be particularly relevant because of the possible link between epileptic seizures and food intake (e.g., Tényi et al. Scientific Reports 2021). Were epileptic artefacts also modulated by mealtimes but in an opposite direction?

Comment 3-2

Also, it seems that the SWR drops 0.5-1h before the mealtime. Is this because mealtimes are approximate, and participants could start eating a bit before? Would it be possible to narrow down the temporal relationship between food intake and SWR rates by aligning SWR rate to the exact onset of each meal? This would be quite important for the authors' interpretation of SWR rate modulations with blood glucose levels (see Discussion).

Response to comment 3-1:

We appreciate this comment from Reviewer #1. We understand the concerns of Reviewer #1 regarding the relation between the observed SWR variations and the artifacts and epileptic activities. To assess the timing of artifacts and seizures, we plotted a frequency histogram of the periods excluded as noise and the periods excluded as seizures. The frequencies of the excluded periods did not vary significantly at different times of day ($F_{23,24} = 0.35$, $P = 0.9933$ $n = 48$; one-way ANOVA). For both the seizure and noise periods, the frequency difference between the mealtimes (7:00, 12:00, and 18:00) and other times was not statistically significant (mealtime; 0.0919 ± 0.0154 , other time; 0.0821 ± 0.0179 , $=0.5411$, Wilcoxon's rank-sum test). In addition, we examined the root mean square (RMS) values of the LFP signals recorded by the hippocampal electrodes for all cases and plotted them with the mean SWR event rate per minute over 24 hours. Furthermore, we assessed the power spectrum (PS) of the LFP signals recorded by the hippocampal electrodes for the representative case (Pt-04). There was no apparent increase in the RMS or PS of the signals, which indicates an increase in the noise level, consistent with eating. These additional analyses suggest that the decreased SWR rate observed during mealtimes

cannot be easily explained by the increase in artifacts. We added these results to the Supplementary Information and cited them in the Results. Additionally, we discussed that while epileptic seizures and food intake may be linked¹, our analysis did not reveal any significant difference in the frequency of excluded sections during mealtime vs. non-mealtime intervals. Thus, the observed decrease in the SWR frequency during mealtimes is not attributable to an increase in rejected artifacts.

Moreover, the reviewer noted the importance of accounting for potential artifacts during mealtimes and the relevance of understanding hippocampal activity in relation to food intake, especially considering the link between epileptic seizures and meals. In response to this comment, we have included additional information in the Discussion section as follows:

Third paragraph of Discussion (Line 198 page 11)

While a previous study showed a possible link between epileptic seizures and food intake³⁴, our analysis did not reveal any significant difference in the frequency of epileptic seizures or any artifacts between mealtime and non-mealtime intervals (Supplementary Figures 10 and 11).

1. Tényi, D., Janszky, J., Jeges, S. & Schulze-Bonhage, A. Food intake precipitates seizures in temporal lobe epilepsy. *Sci Rep* 11, 16515 (2021).

Supplementary Figure 10

Histograms showing the frequency of excluded intervals at each time of day. (a) Frequency of intervals excluded as epileptic seizures. (b) Frequency of intervals excluded as artifacts. (c) Frequency of all excluded intervals. (d) The difference in the probability of excluded intervals with seizures during mealtimes (at 7:00, 12:00 and 18:00) and non-mealtimes ($P = 0.33$, student's t test). The frequencies between the mealtime and non-mealtime groups were not significantly different. (e) The difference in the probability of excluded intervals with artifacts during mealtimes and non-mealtimes ($P = 0.19$, student's t test). The frequencies between the mealtime and non-mealtime groups were not significantly different. (f) The difference in the probability of all excluded intervals during mealtimes and non-mealtimes ($P = 0.54$, student's t test). The frequencies between the mealtime and non-mealtime groups

were not significantly different.

Supplementary Figure 11

(a) The Z-scored SWR rate averaged for all patients at each time of day. (b) The root mean square values of the signals collected by the hippocampal electrodes were assessed every minute and averaged for each time of day. The figure shows the average root mean square values among all patients. (c) The Z-normalized power spectra for the signals collected by the hippocampal electrodes were averaged for each frequency band at each time for all cases ($n = 10$).

Response to comment 3-2:

We also appreciate this comment. According to the suggestion from Reviewer #1, we annotated the exact times of the start and end of the meal for each patient based on the video recordings, as shown in Supplementary Table 4. Additionally, we plotted the exact times of the meal on the SWR event rate diagram in Figure 2c, as shown below. Moreover, we plotted the SWR event rate according to the exact start time of the meal in Supplementary Figure 12. The SWRs were averaged from 1 hour before to 1 hour after the meal based on the start time of the meal. As indicated by the average SWR for the 9 cases with mealtime data, the SWR event rate decreased at the start of the meal. This suggests that the decrease in SWR activity occurs at the start of mealtimes, rather than being an artifact of approximate timing. We discuss this result in the Results section (Line 119 page 7) as follows.

In addition, the SWR rates decreased at 7:00, 12:00, and 18:00, coinciding with the scheduled mealtimes in the hospital, with the rates returning to baseline levels after meal completion (see Supplementary Table 2 and Supplementary Figure 12).

Figure 2c

Time courses of the SWR rates (top) and neocortical delta power (bottom) for 8 consecutive days (top) in a representative patient. The light blue background indicates the time of day when the room was darkened. The red lines represent mealtimes.

Supplementary Figure 12

The mean and 95% CI of the Z-score of the SWR event rate for the hour before and after the start of the meal for all patients are plotted. The meal start time was determined according to the videos collected for each patient.

While we collected data on the amount of food consumed (Supplementary Table 3), we did not examine the relationship between the amount of food consumed and SWRs, primarily because there was little variation in the amount of food consumed during the electrode implantation period. We also discussed that the SWR rate showed an inverse correlation with blood glucose levels in a previous study, but our data showed that this alone cannot explain the decline in SWR rates at the start of mealtime. To clarify this point, we added the following sentences to the second paragraph of the Discussion:

Line 198 page 11

While a previous study showed a possible link between epileptic seizures and food intake³⁴, our analysis did not reveal any significant difference in the frequency of epileptic seizures or any artifacts between mealtime and non-mealtime intervals (Supplementary Figures 10 and 11).

Comment 4

I apologize for the perhaps naïve remark but isn't the following sentence trivial? "The predicted SWR rates were weakly but significantly correlated with the actual SWR rates ($R = 0.21$, $P = 4.3 \times 10^{-10}$, $n = 2,027$ time points) (Fig. 3b)." Indeed, I am wondering if the fact that the predicted SWR rate correlates with the actual SWR rate does not just mean that the modeling is not completely at chance. I think the authors should, in addition to this, check if the coefficients of the models are significantly different from 0 (they have one coefficient per fold). They could also show not only the average but individual values of the coefficients in figure 3 and 4).

Response to comment 4:

We understand the concerns from Reviewer #1. To assess the statistical significance of the correlation between the predicted and actual SWR rates, we trained a model with randomly permuted data. The

answers to the questionnaires and the behavioral data were permuted at random, and we trained a model with the randomly permuted data. Using the trained model, the correlation coefficient between the predicted and actual SWR rates was evaluated. We repeated this process 1000 times to obtain 1000 correlation coefficients to assess the chance distribution of the correlation coefficient. Then, the correlation coefficient of the model trained with the original data was compared with the chance distribution to determine whether the true correlation coefficients exceeded the values of the correlation coefficients in the top 5% of the chance distribution. In addition, we assessed the confidence interval of the coefficients to ensure that the coefficients were significantly different from zero. These results were moved to Supplementary Figure 14 because we changed the regression model used in the main text as follows.

Before explaining the details of the statistical results, we note that we have changed the model used in the main text in response to the comments from the reviewers. Two reviewers kindly suggested using a general linear mixed-effects model to explain the variation in the SWR rates based on the questionnaire responses and behavioral data. We appreciate this suggestion. We completely agree that our hypothesis can be well tested with the general linear mixed-effects model. Therefore, we used the general linear mixed-effects model instead of the linear regression analysis with cross-validation. The results of the linear regression analysis with cross-validation were moved to Supplementary Figure 14. We explain the results of the general linear mixed-effects model and statistical tests for the correlation coefficients and model coefficients in the response to Comment 5.

Comment 5

When examining the relationship between specific dimensions of thoughts and feelings and SWRs, the authors examine the correlations across all bins. Maybe it would be better to fit mixed-models to take into consideration subject identify (unless this is already done through the z-scoring)?

Response to comment 5:

Again, we truly appreciate this suggestion. We completely agree that our hypothesis can be well tested with the general linear mixed-effects model. Following this insightful suggestion from Reviewer #1, we employed a general linear mixed-effects model instead of a linear regression model with cross-validation.

In addition, as noted in our response to Comment 6, to examine the temporal relationship between the questionnaire response times and the analyzed SWR rate, we examined the period from 15 minutes before to 5 minutes after of the questionnaire responses were provided and analyzed 5-minute windows with 1-minute intervals. The model for each window was analyzed with 1,000 permutation tests to confirm that the results were not due to chance. The test results confirmed that the P value was less than 0.05, indicating that the results were not likely to be due to random errors. The gray line indicates the $P = 0.05$ level after Bonferroni correction. The highest accuracy was obtained for the window from 5 minutes to 0 minutes, so this interval was used in subsequent analyses.

With this change in our approach, the top three contributing predictors remained the same, with External-task Focus, Vivid, and Unwanted remaining significant predictors. The accuracy of the regression model was also higher than that before we changed our approach ($R^2 = 0.576$, $n = 187$ questions, $P < 0.05$, permutation test, Bonferroni corrected). Furthermore, the likelihood ratio test, when compared to a fixed-effects-only model, showed a P value of less than 0.001, demonstrating the significant random effects contributed by the patient. We have updated the analysis in the main text.

To understand the temporal link between the physiometric measurements and the SWR rates being analyzed, we focused on time intervals spanning from -15 to +5 minutes. Within this frame, we analyzed data in 1-minute segments with 1-minute gaps. The model for each window was evaluated with 1,000 permutation tests to ensure that the results were not due to chance. The test results confirmed that the P value was less than 0.05, indicating that the results were not due to random errors. The gray line indicates the $P = 0.05$ level after Bonferroni correction. The highest accuracy was at -1 minute, so this interval was used for the main analysis.

With this new approach, the top contributing predictors remained the same as those before the change, with IBI and EDA remaining significant. The accuracy of the regression model was also higher than that before the change ($R^2 = 0.024$, $n = 106125$ time points, $P < 0.05$, permutation test, Bonferroni corrected).

Figure 3

B, The temporal relationship between the SWR rate and biometric data was evaluated in 1-minute time windows from 10 minutes before to 5 minutes after the acquisition of the biophysical data. A general linear mixed-effects model was applied for each time window. The gray dotted line indicates the $P = 0.05$ level after Bonferroni correction. C, Coefficient of each predictor in the general linear mixed-effects model. The red and blue bars indicate negative and positive coefficients, respectively.

Comment 6

When listing the limitation of their studies, the authors could also mention that the temporal link found between SWR rate and intrusive thoughts is quite broad (positive correlation when extracting the SWR rate over 2-7 minutes before thought probes). Perhaps the authors could examine this in further details, for example by computing the correlation coefficient between thought probes and SW rate on windows of 1 minute locked to probe onset. A more specific temporal relationship would represent stronger evidence in favor of a mechanistic link between SWR and intrusive thoughts.

Finally, can the authors provide details about the period used to predict SWR rates based on the reports from participants? They seem to mention only a period of 2 to 7 minutes.

Response to comment 6:

We appreciate this suggestion to clarify the temporal link between the SWR rate and intrusive thoughts. In our previous manuscript, we only assessed the SWR rate from 2 to 7 minutes according to the thought sampling analysis based on a previous study investigating the relationship between intracranial activity and emotion¹. Therefore, we did not assess the relationships in different 5-minute time windows. Following the suggestion from Reviewer #1, we assessed the relationship between the SWR rates and thought sampling results with a 5-minute time window sliding approach from 15 minutes before to 5 minutes after the questionnaire responses were provided. We applied a general linear mixed-effects model to evaluate the results in each time window. Then, the coefficients of determination of the time windows starting at different times were compared. As shown in the following figure, the coefficient of the time window from 5 to 0 minutes before the completion of the questionnaires was the highest. In the figure, the dashed gray line shows the chance level of $P < 0.05$, which was assessed by the randomly permuted data. As Reviewer #1 suggested, the temporal link between the SWR rate and intrusive thoughts was quite broad. Notably, we found that the interval from -7 to -2 minutes, which corresponds to the time interval used in the original manuscript, was significantly predictive. Additionally, based on this result, we selected the time window with the highest coefficients of determination to evaluate the results of the coefficients of the trained model. Again, the "External-task Focus" aspect had the largest

coefficients in the model; that is, our results and conclusions did not change, even after we considerably changed our methods. We added these data to Figure 4b and c and added a sentence to the Discussion explaining the study limitations as follows.

Third, our data are correlational, and the temporal link found between the SWR rate and intrusive thoughts is broad.

Figure 4B

B, The temporal relationship between SWR rate and thought sampling data with a 5-minute time window from 15 minutes before to 5 minutes after answering the questionnaire. The gray dotted line indicates the $P = 0.05$ level after Bonferroni correction.

Furthermore, the temporal relationship between the biometric data and the SWR rate was examined with similar approach. The relationship between the SWR rate and biometric data was evaluated in 1-minute time windows from 10 minutes before to 10 minutes after data acquisition. A general linear mixed-effects model was applied for each time window. The coefficients of determination were then compared across time windows starting at different times. As shown in the figure below, the coefficient for the SWR window starting 1 minute before the biometric data were obtained was the highest, and the coefficients decreased for the surrounding windows. This indicated that physical activity and the SWR rate were temporally linked. We added these results to Figure 3b as follows:

Figure 3B

B, The temporal relationship between the SWR rate and biometric data was evaluated in 1-minute time

windows from 10 minutes before to 5 minutes after the acquisition of the biophysical data. A general linear mixed-effects model was applied for each time window. The gray dotted line indicates the $P=0.05$ level after Bonferroni correction.

1. Sani, O. G. et al. Mood variations decoded from multi-site intracranial human brain activity. *Nat Biotechnol* 36, 954–961 (2018).

Comment 7

It seems that there are no robust modulation of body and physiological activity (EDA, ACC, BVP) across the day even though these variables should be significantly impacted by sleep wake cycles. Can the authors report in Figure 3, the probability of being asleep as a comparison? Were these variables impacted by vigilance states?

Response to comment 7:

We understand the concerns of Reviewer #1 regarding whether the modulation of physical and physiological activity (EDA, ACC, and BVP) is significantly impacted by sleep-wake cycles. The evidence that these values (EDA, ACC, and BVP) recorded by our wearable device significantly change during the sleep-wake cycle strongly supports that these values reflect the modulation of physical and physiological activity. To directly address this issue, we divided the sleep-wake cycle into two states based on four values collected by the wearable device (EDA, ACC, BVP, and IBI): the time during meals (7:00-8:00, 12:00-13:00, 18:00-19:00), which corresponds to the awake period, and the period with delta power exceeding 4 SD during the night (from 22:00 to 7:00), which corresponds to the sleep period. We selected the second period as the time for deep sleep. The four values collected by the wearable device were averaged for each period and normalized among all data. The two periods were classified according to the normalized values using a support vector machine with 5-fold cross-validation. The mean classification accuracy was 88.38%. These four values recorded by the wearable device were used to successfully classify two different physiological states corresponding to mealtime and sleep. We suggest that these four values can be used to characterize the differences in physical and physiological activity, although these values weakly explain the modulation in the SWR rate. To clarify these points, we added the classification results to Supplementary Figure 23 and discussed the results in the second paragraph of the Discussion as follows:

Second paragraph of Discussion (Line 201, page 11)

In addition, our results were consistent with the increased occurrence of SWRs during non-REM (NREM) sleep⁴, which implies a relation between SWRs and the sympathetic nervous system. However, the weak relation between the SWRs and the physiological data, which we used to identify sleep and wakefulness states, suggested that the effects of the sympathetic nervous system alone could not explain the variability in the SWR activity.

Comment 8

Reports of thoughts and feelings also do not seem modulated across the day according to Figure 4a, suggesting there is little variance to explain. Perhaps the authors could show the z-scored SWR rates for the same period for comparison?

Besides, the figure appears at odd with what is reported in the result section: "Before the data were analyzed, the responses were standardized for each participant. The analysis revealed that only the spontaneity of cognition significantly varied throughout the day, with the highest levels observed in the mid-afternoon and the lowest levels in the early evening ($P = 0.033$, $F_{13,267} = 1.88$; one-way ANOVA)". Were the tests across the four dimensions corrected here for multiple comparisons?

Finally, the data used for the correlation analysis in Figure 4b does not seem to be normally distributed, but the authors used a Pearson t-test. Maybe a non-parametric test would be better here.

Response to comment 8:

Following the suggestion from Reviewer #1, we included the Z-scored SWR rates for the same period in Figure 4a. This addition allows for a more comprehensive comparison and understanding of the observed variance. As noted by Reviewer #1, there was little diurnal variation in thoughts and feelings.

Although only spontaneous cognition varied throughout the day, the test was not corrected for multiple comparisons. There were no significant diurnal variations in thoughts and feelings according to multiple comparisons tests. We revised our manuscript to specify that the observed variations in thoughts and feelings were not significant according to multiple comparison tests as follows.

Line 160 Page 9

There was no significant diurnal variation in thought content or mood among the patients ($P > 0.05$; one-way ANOVA, Bonferroni-corrected).

Figure 4. Correlations of SWR rates with thoughts and feelings. A, Diurnal variations in the mean and 95% confidence interval of Z-score data for focused, vivid, involuntary, and spontaneous thoughts in all 10 patients. B, The temporal relationship between SWR rate and thought sampling data with a 5-minute time window from 15 minutes before to 5 minutes after answering the questionnaire. The gray dotted line indicates the $P = 0.05$ level after Bonferroni correction. C, Coefficient of each predictor in the mixed-effects model. The red and blue bars indicate negative and positive coefficients, respectively. D, A word cloud displaying the coefficient of each feature in the linear regression model. Red and blue indicate positive and negative coefficients, respectively, and the size of each word indicates the

magnitude of the absolute value of the coefficient. Task, Images, and People indicates “External-task focus” and “visually imaginable”, and “Social”, respectively. *Note that only the spontaneous nature of cognition varied with time of day; the other three dimensions of experience are included in Panel A because they were related to the occurrence of hippocampal SWR.*

In addition, we appreciate that Reviewer #1 noted that the distribution in Figure 4b was not normal. First, we apologize for the fact that our description of the SWR event rate was not precisely written. Actually, the SWR event rate used in the regression model was not normalized (the horizontal line in Figure 4b). We apologize for this mistake. In the revised manuscript, we Z-normalized the SWR event for each patient and applied a general linear mixed-effects model to the normalized SWR events. Although we Z-normalized the SWR events, the distribution of the Z-normalized SWR events was not normal. Therefore, following the suggestion from Reviewer #1, we applied Spearman's correlation as a nonparametric test to evaluate the correlation and better understand the underlying relationship between the variables. We revised our manuscript to clarify the normalization of the SWRs in the Methods and Figures 3 and 4 as follows.

In the Methods section (Line 429, Page 21)

Standardization of data

The questionnaire and behavioral data were standardized over the entire recording period for each patient. The mean of all scores was subtracted from the individual scores, and the scores were then divided by the standard deviation of all scores. The diurnal variation in the SWR was evaluated according to the within-day standardized results. The SWRs to be explained by the questionnaire responses or behavioral data were standardized for all data of each patient.

Figure 3

C, Coefficient of each predictor in the general linear mixed-effects model. The red and blue bars indicate negative and positive coefficients, respectively.

Figure 4
C, Coefficient of each predictor in the mixed-effects model. The red and blue bars indicate negative and positive coefficients, respectively.

Comment 9

The authors used a cross-validation for their linear model, but they write that "In the cross-validation, all answers to questionnaires were randomly divided into 10 groups for training and testing". This suggests that the training and testing sets for each validation were including overlapping subjects which could lead to over-fitting. Do the results replicate when populating the training and testing sets with different groups of subjects?

Response to comment 9:

The reviewer raises a valid concern regarding the potential of overfitting due to subject overlap with our original cross-validation approach. We understand this concern from Reviewer #1. To prevent overfitting due to subject overlap, we could not use a leave-one-subject out approach due to the large variation in the number of questionnaire responses per patient. Instead, we employed a 3-fold cross-validation method while maintaining the order of the questionnaire responses for each patient. In other words, the data aligned by patient were divided into three parts and cross-validated to reduce patient overlap between the training and testing sets as much as possible. Our revised analyses using this 3-fold cross-validation approach were consistent with the results obtained with our initial linear regression model. The SWR was predicted according to the thought contents with significant accuracy, and the correlation coefficient between the actual and predicted SWR rate was 0.5030. In addition, we identified that the three most significant contributions in predicting the SWR event rate were from the External-task Focus, Unwanted, and Vivid scores, thus validating the robustness of our original findings. We added this result to Supplementary Figure 14 because we reported the results using a generalized linear mixed-effects model in the main text.

Supplementary Figure 14

A regression analysis was performed to predict the SWR event rate according to the questionnaire responses with a 3-fold cross-validation approach using a general linear regression model, maintaining the order of each patient's responses (correlation coefficient = 0.5030, $P < 0.001$, $n = 187$ answers). (a) The correlation between the predicted and actual values. (b) The coefficient for each response. The error bar shows the 95% confidence interval.

Minor comments:

Comment 1

The links for the data and code do not seem to exist. I assume this is because the authors intend to make the data and code available upon publication, but I thought to mention this for the avoidance of doubt.

Response to comment 1:

As Reviewer #1 inferred, we intended to provide the data and code at the time of the publication of the paper. However, to prevent any confusion, we uploaded the necessary data and code information as suggested. This should provide clarification on their future accessibility.

Data availability

The datasets generated and analyzed during the current study are available on Figshare at <https://figshare.com/s/568e9da3493111453f08>

Code availability

The code written to detect SWRs is available at <https://figshare.com/s/84409845359e7e5ca389>

Comment 2

"The correlations were similar during the day and night (Fig. 2e; $R = 0.38 \pm 0.25$ for daytime and 0.36 ± 0.19 for nighttime; $t_5 = 0.2993$; $P = 0.7768$; paired t test)." Instead of using a paired t-test to test the similarity of two variables (in other words, to test H_0), the authors could rely on Bayesian statistics.

Response to comment 2:

We thank the reviewer for this suggestion. The Bayesian t test was used to compute Bayes factors. The Bayes factor 10 was calculated to be 0.387, weakly indicating no difference. We updated the Results section as follows:

In addition, the correlation coefficients between the SWR rates and delta amplitudes of each patient were similar during the day and night (Figure 2e; $R = 0.38 \pm 0.25$ for daytime and 0.36 ± 0.19 for

nighttime; $t_5 = 0.2993$; $P = 0.7768$; paired t test; Bayes factor₁₀ = 0.387).

Comment 3

When examining the relationship between SWE and delta power (Figure 2d), the authors could apply a mixed model with a random factor of subject to check if the relationship generalizes across subjects rather than reporting a correlation coefficient extracted across time bins. Besides, if the SWR rates in Figure 2d do not seem z-transformed (data not centered around 0) contrary to what is mentioned on page 5. Is this normal? Were the correlations reported on page 5 obtained with non z-transformed SWR rates? If so, why switching after for z-scored data?

Response to comment 3:

We appreciate this suggestion from Reviewer #1. Following this suggestion, we changed our approach from simply examining correlation coefficients across time bins to utilizing a general mixed-effects model to analyze the relationship between the SWR rate and delta power. We normalized the SWR event rate and delta powers for each day. Then, a general mixed-effect model was applied to the normalized SWR event rate and delta power. As shown in Figure 2d, the normalized SWR event rate was significantly explained by the normalized delta power of 6 patients ($R^2 = 0.19$). The coefficient for the delta band power ranged from 0.41264 to 0.45986, indicating a significant contribution. We revised Figure 2d to show the results of the general mixed-effect model and the description in the Results section as follows:

Line 104, Page 7

Over seven days of recording among six patients, the SWR rates were significantly correlated with delta amplitudes according to a general linear mixed-effects model (Fig. 2d; $R^2 = 0.19$, $n = 6768$ time points; see Supplementary Figure 7).

Reviewer #2 (Remarks to the Author):

General comments

Iwata et al. Nat Comm

In this submission, Iwata et al. examine whether sharp-wave ripple (SWR) events, which have been linked to memory replay in rodents and humans, undergo circadian fluctuation and whether higher SWR rates may be a signature of intrusive thoughts and internally generated patterns in humans with epilepsy. These findings are potentially interesting because the authors claim their findings show that SWRs may provide an electrophysiological signature of internally generated thoughts in humans; given the relation

to rodent literature, it would be interesting if a human analogue were shown to be true.

While the findings presented in this manuscript (if true) are potentially interesting, I have a number of concerns that greatly diminish the impact of the findings. First, the physiological interpretation of the SWR events detected by the authors is not clear, particularly in the context of the epilepsy population employed in the study. Methodological details are lacking to have sufficient information about whether the authors have reasonably excluded pathological ripples, or whether events studied are actually a combination of physiological and pathological ripples (and if the latter is true, how this relates to rodent literature, which is mainly related to physiological ripples, would be unclear). Second, the behavioral study design is not appropriately set up to answer whether there is an association between external task focus and SWR rate; there is no "task" that patients are performing and, moreover, the authors examine the association between SWRs and a subsequent click on a tablet two to seven minutes later, rather than any task. Third, independent of the concerns about study design and bigger picture interpretation, I have concerns that the significance of any identified results is overstated. While the first and third concerns might be addressed through a major revision, the second concern cannot be addressed without a restructuring of the fundamental behavioral experiment, and therefore I unfortunately cannot recommend this paper for revision outside of a new submission employing a new behavioral paradigm.

Below I have outlined specific suggestions I have for the authors under each concern.¹¹

Response to the comments:

We sincerely appreciate the comprehensive feedback provided by the reviewer regarding our manuscript. The insights and concerns raised are valuable, and we are committed to addressing them to enhance the robustness and clarity of our study.

We would like to address the reviewer's concerns in the following manner:

Physiological Interpretation of SWR Events

- We acknowledge the reviewer's concern regarding the distinction between physiological and pathological ripples. In our initial submission, we may not have been sufficiently clear in describing how we distinguish between the two.
- We employed stringent criteria to exclude pathological ripples from our analysis. Specifically, cases classified as hippocampal sclerosis were excluded, and sections with epileptic seizures, including electrographic seizures, were reviewed by an epileptologist and removed from the analysis.
- Our methodology also demonstrated the physiological behavior of the detected ripples by correlating them with sleep stages and showing their synchronization with cortical slow waves.
- We recognize the challenge in completely distinguishing physiological and pathological ripples, as cited in a recent publication (Nat Commun. 2022 Oct 12;13(1):6000). To mitigate this limitation, we compared our findings with interictal epileptic discharges and added information on the epileptic focus in the studied population to provide a more comprehensive perspective.

Behavioral Research Design

- We chose a naturalistic approach to study self-generated thoughts, sampling thought content through questionnaires.
- The time range for associating SWRs with questionnaire responses was selected based on previous literature. Additionally, our analysis shows a significant correlation between this time range and the response time of the questionnaire.
- To directly address the concern regarding the relationship between external task focus and SWR rates, we included a supplemental, task-based study that substantiates this association.

Significance of Identified Results

- We re-evaluated our data and addressed any potential overstatements of significance, as per the reviewer's recommendation. We remain confident that our findings are both robust and valuable to the field.

Regarding the concerns that require a major revision and the restructuring of the behavioral experiment, we understand the magnitude of the concerns. We have comprehensively reviewed and revised our methodology to ensure that the concerns are appropriately addressed. While we value the findings presented in the current submission, we recognize the importance of assessing these findings in a robust experimental framework.

Specifically, we address the reviewer's suggestions as follows.

Comment 1

It is not clear whether the inclusion/exclusion criteria and data preprocessing yield an interpretation of their results in the context of physiological ripples, or a combination of physiological and pathological ripples: (a) Patients with hippocampal sclerosis were excluded from analysis, and I presume this was to reduce contamination of SWRs with pathological ripples which would be more abundant in the epileptogenic zone. However, there is no information about whether the patients who were included in the sample could be reasonably determined not to have epileptogenic zones coinciding with the placement of the depth electrodes. Were all the other cases non-lesional on all other imaging sources (for example, which there no hippocampal atrophy and was PET symmetric) and did the patients included have no seizures localized to the hippocampus either on phase 1 or phase 2 EEG monitoring? (b) The authors mention that periods of "intense epileptic activity" were excluded from analysis, but it is not clear what "intense" activity means, as this could be anything from only excluding seizures to excluding seizures and interictal epileptiform discharges (IEDs) and HFOs. Please expand on how "intense" epileptic activity was defined, and how the start/end of the segments around this activity were removed. Due to the relationship of IEDs with overriding pathological ripple activity and HFOs as pathological ripples, this is important to help me and future readers to understand how likely the results are to reflect physiological ripples versus physiological and pathological ripples. (Similarly, in the Results section, please define how "epileptic activity" was defined.)

Response to comment 1:

Response to comment 1-a:

We appreciate the reviewer's detailed review of the methods used in our study, particularly the inclusion/exclusion criteria and how "intense epileptic activity" was defined. The reviewer's questions are crucial for understanding the specificities of our dataset and interpreting the results in the context of physiological versus pathological ripples. Below, we address each of the reviewer's points.

Regarding the exclusion criteria, we initially aimed to reduce the influence of pathological ripples as much as possible by excluding patients with a pathological diagnosis of hippocampal sclerosis. However, in the revised manuscript, we updated our dataset to exclude one case, as one patient was diagnosed with hippocampal sclerosis after the initial submission. This updated analysis is reflected in the revised version of the manuscript. In the revised manuscript, this case was excluded from all analyses, but the results did not change.

Additional details on hippocampal epileptogenicity, including imaging and electroencephalographic findings, have been added to Supplementary Table 2 for each patient included in the study. While an epileptic focus was observed in the medial temporal lobe in some cases, we took rigorous steps to minimize the influence of epileptic ripples by excluding data from the seizure periods and 30 minutes before and after the seizure period, including periods with electrographic seizures. To clarify the effects of epileptic activity on the SWR results, we revised the limitation section in the Discussion as follows.

Line 264 page 14 in the Discussion

Although our data suggest a correlation between SWRs in the hippocampus and the content of ongoing thought patterns, there are several limitations that should be borne in mind when considering these results. First, while we carefully excluded the effects of epileptic activity, it remains unclear whether epileptic activity influenced hippocampal function or led to changes in ongoing patterns of thought. The frequency of the epileptic activity and excluded periods did not explain the diurnal variation in the SWR (Supplementary Figure 10). In addition, although the frequency of interictal epileptic activity was

weakly explained by the thought sampling results, the model weights differed from those of the SWR model (Supplementary Figure 17). Finally, although the patients took antiepileptic medication during mealtime, the SWR event rates decreased at the start of the meal and before the administration of medicines (Supplementary Figure 12). These results suggest that the epileptic-related activities were not directly attributed to the relation between SWRs and thought content.

Response to comment 1-b:

We clarified the term "intense epileptic activity" in the Methods section. In our study, "epileptic activity" includes electrographic seizures as well as pre- and post-seizure periods. This classification was made manually by an epileptologist. The segments that were removed ranged from the normal area before seizure onset to the end of postictal slow wave activity following the seizure. Additionally, a one-second buffer was added before and after the segments, and these were also excluded from the analysis.

We updated the Methods section as follows:

We manually reviewed and excluded data collected during epileptic activity or containing movement artifacts to confirm the presence of SWRs. In this process, epileptic activity periods include electrographic seizures as well as pre- and post-seizure periods. This classification was performed manually by an epileptologist. The segments that were removed ranged from the normal area before seizure onset to the end of postictal slow wave activity following the seizure. Additionally, a one-second buffer was added before and after the segments, and these periods were also excluded from the analysis. While distinguishing between hippocampal sharp wave ripples and epileptic ripples is inherently challenging⁵, we made efforts to align the features of the ripples detected in our study with previously reported characteristics of physiological ripples. For example, the ripple rate increased during sleep and showed phase synchronization with slow waves in the cortex (see Supplementary Figure 7).

Moreover, to assess how epileptic discharges might influence our analysis, we used a mixed-effects model to predict the frequency of interictal epileptic discharges from 2 to 7 minutes before the questionnaire response was provided. The resulting R^2 value was as low as 0.039, indicating little relation to the SWR results (Supplementary Figure 17). In addition, the aspects contributing to the thought contents were different from the results using the SWR event rates, indicating that our results using the SWR events were not directly affected by interictal epileptic discharges. We added an additional analysis to the last paragraph of the Results section as follows:

Line 174 Page 10

In addition, the interictal epileptic discharges were explained with low accuracy by the thought patterns ($R^2 = 0.04$, $P > 0.05$ permutation test, $n = 187$ questions, Supplementary Figure 17).

Supplementary Figure 17 (a) The frequency of standardized interictal epileptic discharges (IED) of 2-7 minutes prior to questionnaire response was regressed using a mixed-effects model on the questionnaire response results ($R^2=0.04$). The coefficients for each questionnaire are shown. Only the item “spontaneous” contributed significantly. (b) The results of the permutation test. The order of the predicted IED frequencies was switched back and forth in random positions and regressed 1000 times to see the distribution of the coefficient of determination. The results show that the regression analysis of IED frequency is not significant ($P > 0.05$).

Comment 2

Behavioral study design: This is perhaps my greatest fundamental concern with the paper. The major claim of the paper is that higher SWR rate is associated with less external task focus and more intrusive thoughts. However, there is no specific “task” that the participants are engaged in, so the idea of “task focus” does not particularly make sense. Specifically, the subjects are asked every hour to fill out a survey that is used to assess the presence of “intrusive thoughts”. The authors take the SWR rate 2 to 7 minutes before each response to a question on this survey as the rate that is related to the survey response. There are two problems with this. First, the SWR rate measurements may not be occurring at the same time as the intrusive thoughts. In fact, the data might indicate that intrusive thoughts correlate with some brain state that is preceded by high SWR rates. Second, there is no task being performed during the times of SWR rate measurements. Thus, the authors ability to make a clear connection between SWR rates, task performance, and intrusive thoughts is limited.

Response to comment 2:

We thank Reviewer #2 for the insightful review and valuable feedback on our paper. These comments highlight two related features of our study.

Investigation of neural correlates of states that are not driven by tasks. Studies in both the laboratory and daily life highlight that cognition is often not bound by the task we are performing. For example, in daily life, intrusive thoughts may occur when we are not occupied by a task¹. These types of experiences cannot be directly understood with either animal models or simple task-based methods

since they occur spontaneously. To understand these spontaneous, yet important, features of human cognition, we utilized multidimensional experience sampling², a novel method of probing cognition to understand its underlying features in both task and daily life situations. The mDES allows us to characterize cognition and its relation to other measures that can be obtained simultaneously. Prior studies using this method have highlighted multiple features of brain-cognition associations, including 1) the role of the default mode network in patterns of social episodic thought³, 2) the role of the dorso lateral prefrontal cortex in the maintenance of both task-relevant and self-generated thoughts⁴ and 3) that states emerging at rest, which are dominated by activity in the medial temporal lobe⁵, are also linked to intrusive, past-related thoughts. In this context, our study adds to an emerging body of work, demonstrating that the mDES is useful for understanding the neural correlates of both task-relevant and self-generated states. In this particular context, our paper provides important support for the view that SWRs can be involved in thoughts that emerge when we are not engaged in a task. Nonetheless, the reviewer is correct that these data are descriptive since, by definition, we cannot directly control the emergence of spontaneous thoughts. Thus, our study makes an important contribution to our understanding of the range of states that SWRs contribute to by establishing a correlation between SWRs and intrusive thoughts in naturalistic situations. Nonetheless, with this approach, we cannot clearly describe how SWRs contribute to cognition. However, the publication of these data is still important because they pave the way for a more nuanced mechanistic understanding of the role of SWRs in human cognition by establishing that the mDES is an important tool for mapping the correlates of SWRs. In the future, task and neuromodulatory manipulations can be combined with the mDES and hippocampal recordings to evaluate the role of SWRs in cognition. In this revision, we have explicitly described this idea as follows:

Third paragraph in the Discussion section

To understand the associations between the SWR rate and ongoing thought patterns, we employed experience sampling, including mDES scores⁶. This approach provides a complementary method to understanding brain function for studies that employ specific tasks to understand the association between brain activity and cognition⁴¹. Although our study is the first to use mDES scores to relate thought patterns to SWRs, prior studies have shown that mDES scores allow researchers to reliably categorize thought patterns in different situations in the lab⁷ and daily life^{8,9}. In the laboratory, mDES scores have been shown to effectively characterize the association between brain activity and thought patterns that reflect task-relevant processes¹⁰, off-task social episodic thoughts during tasks¹¹ and thoughts that emerge at rest¹². These works have highlighted several features of ongoing thought that are relevant to the present work. First, intrusive thought patterns (broadly similar to those seen in the present study) are most clearly observed in daily life when individuals are unoccupied by a task (doing nothing or resting⁹). Second, studies that have used functional magnetic resonance imaging (fMRI) data to characterize thought patterns at rest have identified that such states show increased activation within the medial temporal lobe¹². Together with the present study, these studies suggest that SWRs may influence self-generated states that occur during periods when individuals are unoccupied by demanding external tasks. Although understanding the correlations between cognition and neural activity that are not directly related to task performance is an advantage of the mDES in understanding brain-cognition associations, we emphasize that these data are correlational. In the future, we will combine task manipulations¹⁰ and pharmacological interventions¹³ with activity recordings within the hippocampus and experience descriptions according to mDES scores to better characterize the mechanistic role of SWRs in both task-driven and self-generated states.

In addition, we performed additional experiments to collect SWR events from three new patients (5 hippocampi) when the subjects intentionally focused on breathing and when they were engaged in performing calculations. Focusing on breathing is a known technique to induce mind wandering. Three patients with implanted electrodes in the hippocampus were instructed to focus on their breathing without controlling the breathing frequency for 10 minutes. After a rest period, the same patients were instructed to perform a calculation where they subtracted three, seven, nine, eleven or thirteen from 100 for 2 minutes each, for a total of 10 minutes. We instructed the subject to say the results of the calculation aloud each time to confirm that they were concentrating on the task. On the other hand, while the subjects were focused on their breathing, we asked the subjects how much they focused on

the task every minute to assess mind wandering during the task. The patients rated their focus on the breathing task on a scale from 1 to 5, with 1 being the most focused and 5 being not focused at all. According to the subjects' reports, three patients experienced mind wandering with a score of 2.5 ± 1.1 during the breathing task, indicating that the breathing task induced mind-wandering states. Interestingly, we observed significantly higher SWR rates in the breathing task than in the calculation task. In addition, the SWR rate during the breathing task was significantly higher than that during the one-hour period before and after the task when the subject behaved naturally. These results also support our hypothesis that the SWR rate increases during mind wandering. We added these results to Supplementary Figure 13.

Temporal relationship between SWR and patterns of thought. Second, the reviewer argues that the temporal resolution of our measure does not provide the temporal resolution that other behavioral measures of cognition may provide. Although this is undoubtedly correct, we know very little regarding the duration of ongoing thought patterns and how they relate to events that could reflect their genesis. To address this limitation, we examined the temporal relationship between SWR rates and the mDES reports using a 5-minute sliding time window from 15 minutes before to 5 minutes after answering the questionnaires. We applied a general linear mixed-effects model for each time window. Then, the coefficients of determination between the predicted and actual SWR rates were compared among the time windows starting at different times. As shown in the following figure, the coefficient of determination changed continuously over time and was the highest in the time window from -5 to 0 minutes from the completion of the questionnaires. In the figure, the gray dotted line shows the chance level of $P = 0.05$, which was assessed by the randomly permuted data. Notably, we found that the interval from -7 to -2 minutes, which corresponds to the original time interval, was significantly predictive. Thus, although our analysis cannot reveal with a high degree of temporal precision the exact relationship between SWR rates and the emergence of specific thought patterns, we determined that they are most closely related to temporal windows in the minutes prior to the probe. These data are important because they establish the most likely subjective correlates of SWRs in humans using the only currently available method. While this limits the conclusions regarding timing based on these data, in the future, task and pharmacological manipulations can be used in combination with mDES scores and hippocampal recordings to better understand the temporal associations between subjective descriptions of experience and the links to neural activity (including SWRs).

We updated the Results section as follows:

We used a generalized linear mixed-effects model to investigate the Z-standardized SWR rates averaged over 5 minutes, shifting every 1 minute from 15 minutes before to 5 minutes after answering the questionnaire based on the standardized values of the responses to each question (Figure 4b). The mean SWR rates from -5 minutes to 0 minutes were strongly explained by the answers with the highest accuracy above the chance level ($R^2 = 0.576$, $n = 187$ questions, $P < 0.05$, permutation test, Bonferroni corrected).

We updated the Discussion section as follows:

In this regard, since SWRs in the hippocampus are influenced by neuromodulators, it is possible that pharmacological manipulation of these systems may provide the opportunity to test causal accounts of the role that SWRs play in ongoing thought patterns in humans ¹⁷ (see Supplementary Figure 13 for a preliminary investigation using behavioral induction).

Supplementary Figure 13

We performed an additional experiment to assess the SWR events of three new patients (5 hippocampi) when the subjects intentionally focused on breathing and when they were engaged in calculations. Focusing on breathing is a known technique to induce mind wandering. Three patients with implanted electrodes in the hippocampus were instructed to focus on their breathing for 10 minutes. After a rest period, the same patients were instructed to perform a calculation where they kept subtracting seven, nine or eleven from 100 for 10 minutes. We instructed the subject to say the results of the calculation aloud each time to confirm that they were concentrating on the task. On the other hand, while the subjects were forced on their breathing, we asked the subjects how much they focused on the task every minute to assess the mind-wandering during the task. The patients rated their focus on the breathing task by a score from 0 to 10, with 10 being the most focused and 0 being not focused at all. According to the subjects' reports, three patients experienced mind wandering with a score of 2.5 ± 1.1 during the breathing task, indicating that the breathing task induced mind-wandering states.

The figure shows a box plot of the SWR event rate during the 10 minutes of the calculation task (calculation), breathing task (mind wandering) and 1-hour period before and after these tasks, during which the subject behaved naturally (normal). The SWR rates were significantly higher in the breathing task than in the calculation task and normal behavior periods. These results also support our hypothesis that the SWR rate increases during mind wandering. *, $P < 0.05$, student's t test, Bonferroni corrected.

1. Mulholland, B. et al. Patterns of ongoing thought in the real world. *Conscious Cogn* 114, 103530 (2023).
2. Smallwood, J. et al. The neural correlates of ongoing conscious thought. *iScience* 24, 102132 (2021).

3. Konu, D. et al. A role for the ventromedial prefrontal cortex in self-generated episodic social cognition. *Neuroimage* 218, 116977 (2020).
4. Turnbull, A. et al. Left dorsolateral prefrontal cortex supports context-dependent prioritisation of off-task thought. *Nat Commun* 10, 3816 (2019).
5. Karapanagiotidis, T. et al. The psychological correlates of distinct neural states occurring during wakeful rest. *Sci Rep* 10, 21121 (2020).

Comment 3-1

The concerns I have about overstatement of the significance of results relate to the fact that results appear to be overstated in several instances and with no attention to checking how robust their results are to analysis choices that are made.

- The authors base their results on a single method of detecting SWRs, i.e., that proposed by Norman et al. (2019). However, it is known that detected SWR rates can vary several fold depending on the detection method used, see for example Liu et al. (2022). To ensure that these results are not sensitive to SWR detection method, I would advise using multiple SWR detection methods, such as that by Vaz et al. (2019) and Staresina et al. (2015). Consistency across multiple detection methods and methods of analysis would increase the likelihood of this being a true result.

Response to comment 3-1:

We thank the reviewer for noting that the methodological approach can substantially influence the detected SWR rates, and we agree with the reviewer that it is crucial to confirm that our results are obtained using other methods to detect SWRs. To ensure the robustness of our findings, we considered the reviewer's advice and performed additional analyses using multiple SWR detection methods, including those proposed by Vaz et al. (2019) and Staresina et al. (2015), as suggested^{1,2}.

We detected SWRs with each method based on 24 hours of hippocampal electrode EEG data from a representative case (Pt-03). We calculated the mean frequency (/sec) in 1-minute bins and normalized the data according to the whole dataset. Furthermore, we examined the correlation between the frequencies detected by these different methods and found significantly high correlations among the results of the various approaches ($R=1$ using the method by Vaz et al. and $R=0.87661$ using the method by Staresina et al.). These results indicate that our primary findings are not sensitive to the choice of SWR detection method and thus address the reviewer's concern regarding the robustness of our results to methodological choices (see Supplementary Figure 20). We discussed these results in the Methods section as follows:

SWR detection in the Methods (Line 408, page 20)

We selected this method because they reported a relationship between SWRs and visual imagery, a type of thought content (see Supplementary Figure 20 for a comparison of our approach with other methods to detect SWRs).

Supplementary Figure 20

Comparison of the 24-hour standardized SWR rate of a representative case (Pt-03) detected by the original method (Norman et al.) with the standardized SWR rates detected by two other previously reported methods (a, Vaz et al.; b, Staresina et al.), showing significant correlations between the results of the different approaches.

1. Vaz, A. P., Inati, S. K., Brunel, N. & Zaghoul, K. A. Coupled ripple oscillations between the medial temporal lobe and neocortex retrieve human memory. *Science* (1979) 363, 975–978 (2019).
2. Staresina, B. P., Niediek, J., Borger, V., Surges, R. & Mormann, F. How coupled slow oscillations, spindles and ripples coordinate neuronal processing and communication during human sleep. *Nat Neurosci* 26, 1429–1437 (2023).

Comment 3-2

- In the analysis to examine whether SWR rates exhibits circadian fluctuation, some unknown thresholds are applied to time of day to define different categories of time of day, and it is not clear what these thresholds are to define their categories, or how generalizable the results would be should the thresholds have been chosen (for example) to produce statistical significance. Since circadian rhythms are inherently angular in nature, it would be much more natural to use a more data driven approach with circular statistics to quantify this association -- for example, reporting the distribution of the mean resultant vector across patients would tell the reader the strength how concentrated the circular distribution of SWRs is around specific times of day.

Response to comment 3-2:

We understand this concern from Reviewer #2. Although we demonstrated the circadian fluctuation in the SWR events by one-way ANOVA for data collected over 24 hours without categorizing the 24 hours by any thresholds, we labeled some times during the day as fixed times in the 24-hour period. These labels include mealtimes and nighttime. During the recording, all patients lived in the hospital room, so the times when meals were served and the room lights were switched off and on were determined by hospital regulations. In our previous manuscript, we described those times as external factors affecting the circadian rhythm. However, this categorization of time in a day might mislead the reviewers into recognizing unknown thresholds. We apologize for our unclear description of this point. Therefore, we clarify the definitions of the times during the day and how we used this categorization in our analysis. In addition, we added more behavioral data to classify the time more precisely (Supplementary Table 4).

Mealtimes: The mealtimes were defined as 7:00-8:00, 12:00-13:00 and 18:00-19:00 based on our hospital regulations. However, the times at which the meals started and ended were not the same as this duration. Therefore, we assessed the exact start and end times of the meals for each meal of each patient based on the video data and summarized the results in Supplementary Table 4. In addition, we analyzed the variation in the SWR event rate in accordance with the exact meal start and end times. Supplementary Figure 12 demonstrates that the SWR rates significantly changed after the participants started mealtime.

Supplementary Figure 12

The mean and 95% CI of the Z-score of the SWR event rate for the hour before and after the start of the meal for all patients are plotted. The meal start time was determined according to the videos collected for each patient.

Daytime and nighttime: In our analysis, nighttime was defined as the period from 22:00 to 6:59. Daytime was defined as the period from 7:00 to 21:59. This definition was based on the hospital schedule and when the room lights were turned off and on, where the living conditions during hospitalization remain relatively constant. Lights are usually turned off at 22:00 and turned back on at 7:00, in accordance with hospital rules.

Morning and afternoon: We divided the daytime into morning and afternoon by excluding the mealtimes. The morning was defined as the period from 8:00 to 11:59, and the afternoon was defined as the period from 13:00 to 17:59. In our previous manuscript, the morning included the time for breakfast, and the afternoon included the times for lunch and dinner. However, to clarify the different time periods, we excluded the mealtimes from the morning and afternoon periods. However, the results remained consistent; the SWR rate was higher in the afternoon than in the morning.

Although identifying the sleep period is very interesting in this study, it was difficult to identify the sleep stages precisely because we did not record electromyography data simultaneously with the other recorded data. In addition, when we identify sleep stages, especially non-REM sleep, according to the delta power, it will significantly affect the results because the SWR is correlated with delta power. Therefore, we chose not to define sleep time. Instead, we analyzed the SWRs during the daytime and nighttime to assess the differences in the SWR rate between different times. In our previous manuscript, we wrote “sleep” instead of “nighttime”. We apologize for this mistake, which introduced confusion regarding the categorization. To clarify the definition of the different periods, we revised the Methods section as follows:

Scheduled patient periods during recording

We divided the 24-hour period into several periods based on the mealtimes and lighting conditions in the patient’s room. During the recording period, all patients lived in the hospital room, so the times when meals were served and when the room lights were switched on and off were determined by hospital regulations. In this study, nighttime was defined as the period from 22:00 to 6:59. Daytime was defined as the period from 7:00 to 21:59. This definition was based on the hospital schedule, with the

room light turned off at 22:00 and on at 7:00. Similarly, mealtime was defined as 7:00-8:00, 12:00-13:00 and 18:00-19:00 based on hospital regulations. The exact times of the start and end of each meal were assessed by video monitoring if possible (Supplementary Table 4). In addition, we divided the daytime into morning and afternoon periods by excluding mealtimes. The morning was defined as the period from 8:00 to 11:59, and the afternoon was defined as the period from 13:00 to 17:59.

In addition, we revised the Results section as follows (Line 113 page 7):

The SWR rate showed significant overall variation over a 24-hour period (Fig. 2f; $P < 0.001$, $F_{143,83392} = 41.6$, $n = 113760$ time points from a total of 79 days in 10 patients; one-way ANOVA; see Supplementary Figures 18 and 19 for the diurnal fluctuation in the SWR duration and the frequency of clustered or isolated SWR events). In particular, the SWR event rate was significantly higher during the night (0.234 ± 1.072 counts/min (mean \pm SD)) than during the daytime (-0.153 ± 0.918) ($P < 0.001$, $t_{116186} = -65.58$, $n = 156960$ time points; student's t test).

Results section (Line 122 page 7):

During the day, the SWR rates were higher during the afternoon than during the morning (morning (from 8:00 to 11:59), -0.21 ± 0.90 ; afternoon (from 13:00 to 17:59), -0.08 ± 0.97 ; $P < 0.001$, $t_{28,136} = -11.7$, $n = 42,660$ time points, student's t test).

Following the reviewer's suggestion about incorporating statistics to capture the circular nature of circadian rhythms, we revised Figure 2f to include a circular plot. This updated plot has also been added to Supplementary Figure 8.

Supplementary Figure 8

(a) Radial plot of diurnal fluctuations in normalized SWR rates ($n=10$). (b) The post hoc multiple comparisons of the hourly SWR event rates of all cases ($n = 10$). The circle and bar represent the mean and confidence interval for the SWR event rate over one hour. The nonoverlapping confidence intervals among circles show the significant difference between the circles.

Comment 3-3

- The authors report that there were transient drops in SWR rates around 7:00, 12:00, and 18:00, coinciding with scheduled meal times in the hospital. This could be potentially interesting given the contrast to rodent literature about increases in SWR rate occurring around reward in rodents. However, how were the transient drops in SWR rates at 7:00, 12:00, and 18:00 actually estimated, and was this

present in all patients? Currently, there are no actual statistics provided to quantify this claim, except for a figure qualitatively showing this occurring for a single patient.

Response to comment 3-3:

We appreciate that Reviewer #2 showed interest in the transient decreases in the SWR rates at mealtimes. First, we apologize for any confusion caused by the way we presented Figure 2f. Although it was not our intention to mislead, the data shown in Figure 2f represent the average and 95% confidence interval among all 10 patients and are not limited to a single patient. To clarify this point, we have revised Figure 2f to show the number of subjects as $n=10$.

Second, the reviewer raised a crucial point about the statistical verification of the transient decreases in the SWR rates around mealtimes. Initially, we assumed that the 95% confidence intervals would convey the significance of our findings. However, according to the reviewer's feedback, we performed additional statistical analyses to strengthen our claim. Specifically, we conducted post hoc multiple comparisons based on the hourly SWR event rates. These additional statistical results have been included in Supplementary Figure 8 as follows.

Supplementary Figure (b)

Post hoc multiple comparisons of the hourly SWR event rates of all cases ($n = 10$). The circle and bar represent the mean and confidence interval for the SWR event rate for one hour. The nonoverlapping confidence interval among circles shows the significant difference between the circles.

Additionally, we assessed the SWR event rates based on the exact time at which the meal was started, which was determined based on the video monitoring data. The SWR event rates were compared one hour before and one hour after the exact meal start times. The results showed that the SWR event rate decreased at the exact time at which the meal was started. This finding has been added to the Supplementary Information.

Supplementary figure 12

The mean and 95% CI of the Z-score of the SWR event rate for the hour before and after the start of the meal for all patients are plotted. The meal start time was determined according to the videos collected for each patient.

Comment 3-4

- For the section on correlation between SWR rates and biophysical parameters, the authors need to be very clear that their results, while statistically significant, indicate that physiological measures capture only a very small fraction of the variance in SWR rates. Ideally, they would report R^2 values, as these are directly interpretable in terms of variance explained. For example, it seems that the authors have found that the coefficient of determination for a linear regression of SWR rate on time of day and four physiological measurements is $0.21^2 = 0.04$.

- Of these four physiological measurements, it seems that BVP and IBI (which are both proxy measures of heart rate) explain the greatest proportion of variance, but both have small effect sizes ($-0.007^2 = 4.9e-5$ for BVP and $0.15^2 = 0.02$ for IBI). Therefore, from what I can see, a more accurate interpretation of this finding would be that the four physiological measures acquired from the Empatica wristwatch explain only a very small fraction of the variance in SWR rates.

- On a related note to the above, although it is technically valid to report the correlation between predicted and observed SWR rates (i.e. the square root of the coefficient of determination), it is standard in regression analysis to report R^2 (rather than its square root) as a measure of model fit, as it has straightforward interpretations with regards to the information about the proportion of variance explained by the model. Especially because the p-values can be misleading in the setting of large sample sizes, I would suggest reporting R^2 rather than its square root when reporting model fit.

Response to comment 3-4:

We want to apologize for any prior confusion due to our initial presentation of the data. We agree that it is essential to be absolutely clear about the statistical significance explained by our physiological measures. In the revised manuscript, we have applied the reviewer's advice to explicitly report the coefficient of determination, commonly known as R^2 , in Figures 3 and 4. We believe this will clarify the degree to which the physiological measures explain the variance in the SWR event rates.

To address the reviewer's concern regarding the statistical model used, we have changed our approach from a linear regression analysis method to a general linear mixed-effects model. Using the general linear mixed-effects model, the normalized SWR rates from -5 to 0 minutes after the thought sampling data were collected were explained by the normalized answers to the 17 questions, with $R^2 = 0.576$. The statistical significance of the model was tested by a permutation test. In addition, when we applied the same model to evaluate the physiological measures collected by the Empatica wearable device, the R^2 value was 0.024, with statistical significance determined by the permutation test. Although the model explained the variance in the SWR significantly better than the chance level, the R^2 value was low, implying low interpretability using the physiological scores measured by the wearable device. We revised our manuscript to clarify the R^2 results and emphasize the low interpretability of the

physiological data in the Results and Discussion sections as follows.

In the Results (Line 148, page 9)

The SWR rates were weakly explained by the model using the physiological scores at one minute after the SWR event with the highest accuracy ($R^2 = 0.024$, $n = 106125$ time points, $P < 0.05$, permutation test, Bonferroni corrected, Figure 3b).

We updated the first paragraph of the Discussion (Line 201, page11) as follows:

In addition, our results were consistent with the increased occurrence of SWRs during non-REM (NREM) sleep⁴, which implies a relation between SWRs and the sympathetic nervous system. However, the weak relation between the SWRs and the physiological data, which we used to identify sleep and wakefulness states, suggested that the effects of the sympathetic nervous system alone could not explain the variability in the SWR activity.

Figure 3
C, Coefficient of each predictor in the general linear mixed-effects model. The red and blue bars indicate negative and positive coefficients, respectively.

Figure 4
C, Coefficient of each predictor in the mixed-effects model. The red and blue bars indicate negative

and positive coefficients, respectively.

Comment 3-5

- Linear regression is used to estimate the association between the self-report scale and SWR rate, which does not account for the repeated measures nature of the study design. The rigor of the study would be greatly improved if the authors used repeated measures mixed effects analysis instead.

Response to comment 3-5:

According to the reviewer's suggestion, we adopted a general linear mixed-effects model to replace the general linear regression analysis used in the original manuscript. We assessed the relationship between SWR rates and thought sampling with a 5-minute time window sliding from 15 minutes before to 5 minutes after the participants answered the questionnaires. We applied a general linear mixed-effects model for each time window. Then, the coefficients of determination were compared among the time windows starting at different times. As shown in the following figure, the coefficient of determination was the highest for the time window from -5 to +0 minutes from the time at which the questionnaire was answered. In the figure, the gray dotted line shows the chance level of $P = 0.05$, which was assessed by the randomly permuted data with Bonferroni correction. Notably, we found that the interval from -7 to -2 minutes, which corresponds to the time interval used in the original manuscript, was significantly predictive. Additionally, based on this result, we selected the time window with the highest coefficients of determination to show the model coefficient results. Again, the "External-task Focus" had the largest coefficients in the model; that is, our results and conclusions were not changed even after these considerable changes to our methods.

Comment 3-6

- The coefficients from regression of SWR rate on the self-report thought measures are not transparently reported. Rather than showing a word cloud, I would suggest reporting all estimated regression coefficients along with standard error and p-values. Word clouds can visually distort the magnitude of differences between regression coefficients and do not adequately represent estimates from regression.
- It seems from the word cloud that other thought variables, such as "happy," "person," "images," and "excited," were also large in magnitude. However, whether these were or were not statistically significant is not disclosed. Were any other regression coefficients statistically significant? If so, please discuss how this would affect interpretation.

Response to comment 3-6:

Following the reviewer's suggestions, we added a figure that clearly displays the coefficients along with their corresponding standard errors in the main text. This new figure ensures a more comprehensive and scientifically robust presentation of the data and addresses the reviewer's concerns regarding the transparency of our statistical reporting.

As Reviewer #2 pointed out, some other questions in the questionnaire significantly contributed to explaining the SWR. As shown in the revised Figure 4c, “External-task Focus”, “Unwanted”, “Vivid”, “Future”, and “Visually imaginable” had weights that were significantly larger than 0. In addition, we added Supplementary Figure 5 shows the *P*-value for each coefficient.

We updated the Discussion section as follows:

Animal studies are limited in terms of their ability to investigate cognition, as cognition cannot be directly linked to behavior. However, using experience sampling, we demonstrated that SWR rates were correlated with patterns of ongoing thoughts, as established via mDES scores. We found that SWR rates were significantly positively associated with less wanted, more vivid, and more visually imaginable thoughts. Moreover, SWR rates were negatively associated with focusing on external tasks and on the future. Together, these findings support an emerging hypothesis relating the medial temporal lobe with self-generated cognition⁶. For example, prior studies have shown that hippocampal lesions are associated with reduced off-task thoughts, such as mind wandering¹⁴. In addition, gray matter integrity in the hippocampus/parahippocampus has been linked with the capacity for mind wandering in patients with dementia¹⁵, while, in healthy adults, reports of ongoing thoughts with more vivid features are linked to greater gray matter volumes in the posterior parahippocampus, while volume in the anterior parahippocampus is linked to off-task states¹⁶.

Comment 3-7

- Throughout the manuscript, p-values are reported for all hypothesis tests. However, given the large number of time points included in many of these hypothesis tests, p-values are not as meaningful and I would suggest the authors comment not only on statistical significance but also magnitude of effect size.

Response to comment 3-7:

We appreciate this comment. Following the reviewer’s suggestion, we have included the magnitude of the effect sizes for all analyses in the revised manuscript.

Comment 4-1

Last, various aspects of the methods are not clear and need to be clarified.

- Regarding classification of sleep/wake states using delta power: It is not clear from the current description what supervised or unsupervised learning method was used to classify these two states based on delta band power. Please add this to the methods.

Response to comment 4-1:

We apologize for the lack of clarity in the original description. For this analysis, we did not classify sleep and wakefulness using delta band power in our previous manuscript. Instead, we categorized the data as pertaining to either nighttime or daytime, based on the time of day and hospital rules, as previously mentioned. During these periods, we observed a significant increase in SWR rates during nighttime and a decrease in SWR rates during the daytime.

We have clarified this point in the Methods section of the revised manuscript to better communicate our approach and eliminate any ambiguities as follows. In addition, sleep was changed to night in Figure 2f.

We updated the Method section as follows:

Scheduled patient periods during recording

We classified the 24-hour period into several time periods based on the meal times and lighting of the patient’s room. During the recording, all patients lived in the hospital room, so the times when meals were served and room lights were switched off and on were determined by hospital regulations. In this study, nighttime was defined as the period from 22:00 to 6:59. Daytime was defined as the period from 7:00 to 21:59. This definition was based on the hospital schedule, with the room light turned off at 22:00 and on at 7:00. Similarly, mealtime was defined as 7:00-8:00, 12:00-13:00 and 18:00-19:00 based on

hospital regulations. The exact times of the start and end of each meal were assessed by video monitoring if possible (Supplementary Table 4). In addition, we divided the daytime into morning and afternoon periods by excluding mealtimes. The morning was defined as the period from 8:00 to 11:59, and the afternoon was defined as the period from 13:00 to 17:59.

Comment 4-2

- Related to the above, it seems that sleep/wake state classification was based solely on delta band power. Given the difficulty of sleep staging with intracranial EEG (especially as compared to scalp EEG): (1) How were the labels validated? (2) Given that video and accelerometry data were also available, it seems it would have been reasonable use these either to validate the labels, or as additional features into the classification algorithm. Was this considered? (3) Was subdermal EEG or scalp EEG, along with EOG/EMG electrodes, available for sleep staging for these patients? Given that the authors have accelerometry data, this could be presumably also used for classification of sleep/wake states as well as so immobile and mobile wakeful states.

Response to comment 4-2:

We would like to clarify that in our study, we did not actually classify sleep/wakefulness states based on delta band power or EEG data. We understand that the manuscript may have been unclear in this regard, and we apologize for any confusion. Our approach was to categorize the data into nighttime and daytime periods without identifying sleep or wakefulness states.

We updated the Methods section as follows:

Scheduled patient periods during recording

We classified the 24-hour period into several time periods based on the meal times and lighting of the patient's room. During the recording, all patients lived in the hospital room, so the times when meals were served and room lights were switched off and on were determined by hospital regulations. In this study, nighttime was defined as the period from 22:00 to 6:59. Daytime was defined as the period from 7:00 to 21:59. This definition was based on the hospital schedule, with the room light turned off at 22:00 and on at 7:00. Similarly, mealtime was defined as 7:00-8:00, 12:00-13:00 and 18:00-19:00 based on hospital regulations. The exact times of the start and end of each meal were assessed by video monitoring if possible (Supplementary Table 4). In addition, we divided the daytime into morning and afternoon periods by excluding mealtimes. The morning was defined as the period from 8:00 to 11:59, and the afternoon was defined as the period from 13:00 to 17:59.

The reviewer raises a good point regarding the challenges of identifying sleep stages using intracranial EEG, especially for REM sleep, which would require EOG/EMG data. We acknowledge that having this information would add a valuable layer to our analysis. We have included this point in the Discussion section of the revised manuscript to note the limitations of our current approach and to offer future research directions.

We revised the first paragraph of the Discussion section as follows:

Although we did not investigate SWR activity during the sleep stage due to the lack of electrooculogram and electromyogram data, which are essential to identify the sleep stages, the clear circadian rhythm of the SWRs, which increased during the night, is consistent with the changes in SWR activity during the sleep-wake cycle observed in animal studies.

Comment 4-3

- Missing data usually affects self-report measures and its treatment can affect results. Please describe how the degree of missingness and how missing data was treated in the self-report scale, particularly as the association of SWR rate with self-report measures is one of the main findings of the paper.

Response to comment 4-3:

We understand the reviewer's concerns and apologize for the lack of a description of how missing data in the self-report scale were treated. There were no missing data in our self-report scale. All patients completed all 17 questions in the questionnaire every time they responded. Our tablet system to answer the questionnaire was designed so that it could be submitted only if all the questions were answered.

Therefore, we did not have partial completions, with some questions left unanswered. These methodological details have been added to the Methods section of the revised manuscript.

The following sentence was added to the “*Questionnaire*” section in the Methods (Line 357, page 18)
The questionnaire was designed so that it could be submitted only if all the questions were answered.

On the other hand, it should be noted that the number of completed questionnaires varied among patients because the patients sometimes could not answer the questionnaires due to their conditions or medical examinations. To account for the variability in the number of answers following the reviewer’s suggestions, we used a mixed-effects model, which can better accommodate missing or imbalanced data.

Moreover, we have added a discussion of these potential limitations related to the difference in the number of answers among patients to the Discussion section, as follows.

In the fourth paragraph of the Discussion (Line 283 in page 15)

In addition, the number of responses varied among patients because of features related to their condition or the occurrence of medical examinations. These issues may have facilitated the occurrence of certain thought states (in our case, self-generated thoughts in the resting state). Thus, while our study highlights the contribution of SWRs to certain ongoing thought patterns, we cannot rule out the contribution of SWRs to thought patterns that occur in different situations. Additional research with a larger cohort may be needed to elucidate the relationship between SWRs and different features of human thought. Third, it is important to reiterate that our data are correlational and that the temporal link found between the SWR rate and intrusive thoughts is broad; thus, further studies that manipulate SWRs and associated cognitive functions are needed to fully understand the role the hippocampus plays in naturally occurring features of human cognition.

Comment 4-4

- On a related note, how were these 17 questions distributed, i.e., were they shown in random order, or (as is suggested in the table) were they always shown in the same order? If they were shown always in the same order, then order bias is a concern.

Response to comment 4-4:

The reviewer raised an important concern regarding the order in which the 17 questions in the self-report questionnaire were presented, specifically mentioning the potential for order bias. We apologize for the lack of description about how these questions were displayed to the participants with the tablet system. In the system, the 'Task Focus' item was consistently presented as the first question. However, the remaining items were presented to the participants in a random order. This approach was designed to mitigate any potential order bias for all respondents. To clarify these points, we have revised the Methods section as follows.

Questionnaire section in the Methods (Line 363 page 18)

Among the 17 questions in the questionnaire, the “External-task Focus” item was consistently presented as the first question. The remaining items were presented in a random order each time to reduce any potential order bias.

Comment 4-5

- Is the correlation of 0.74 between delta amplitude and SWR rate across all patients, or just for the single representative patient shown in the figure? If the former, it seems a linear mixed effects model would be more appropriate; if the latter, please report for the group of patients.

Response to comment 4-5:

We apologize for any ambiguity caused by the presentation of our data. Figure 2d demonstrates a representative result of one patient. Following the suggestion from the reviewers, we applied a linear mixed-effects model to explain the relation between the SWR rates and delta powers among all patients. As shown in the revised Figure 2d, there was a significant correlation between the two, with $R^2 = 0.19$.

We updated the Results section as follows:

Over seven days of recording among six patients, the SWR rates were significantly correlated with delta amplitudes according to a general linear mixed-effects model (Figure 2d; $R^2 = 0.19$, $n = 6768$ time points; see Supplementary Figure 7).

Comment 4-6

- The authors report results from a one-way ANOVA comparing SWR rates during sleep to daytime. (1) I am curious why sleep was compared to daytime, rather than comparing sleep to wake, or daytime to nighttime. Could the authors please clarify? (2) If daytime to nighttime was compared, how were these defined? (3) It is also not clear to me what three groups are being compared using the one-way ANOVA (it seems there are only two – sleep and daytime – from the text).

Response to comment 4-6:

First, we clarify the choice of comparison groups used in our study. We categorized the 24 hours into two periods, daytime (7:00-21:59) and nighttime (22:00-6:59), based on hospital rules. In the hospital, the room lights were turned off at 22:00 and turned on at 7:00. The changes in the light conditions are an external factor that influence circadian rhythm. Therefore, we categorized the times as discussed above. In our previous manuscript, we mistakenly wrote “sleep” instead of “nighttime” in the Results section. We apologize for this mistake and any confusion regarding the categorization.

We have corrected the text to state that we observed a significant increase in the SWR rate during the night and a decrease during the daytime as follows.

Results (Line 117 page 7)

In particular, the SWR event rate was significantly higher during the night (0.234 ± 1.072 counts/min (mean \pm SD)) than during the daytime (-0.153 ± 0.918) ($P < 0.001$, $t_{116186} = -65.58$, $n = 156960$ time points; student's t test).

We opted to use daytime and nighttime, rather than sleep and wakefulness, because accurate labeling of sleep and wakefulness states was challenging in our study. This has been explicitly stated in the revised manuscript for clarification.

We updated the Methods section as follows:

Scheduled patient periods during recording

We classified the 24-hour period into several time periods based on the meal times and lighting of the patient's room. During the recording, all patients lived in the hospital room, so the times when meals were served and room lights were switched off and on were determined by hospital regulations. In this

study, nighttime was defined as the period from 22:00 to 6:59. Daytime was defined as the period from 7:00 to 21:59. This definition was based on the hospital schedule, with the room light turned off at 22:00 and on at 7:00. Similarly, mealtime was defined as 7:00-8:00, 12:00-13:00 and 18:00-19:00 based on hospital regulations. The exact times of the start and end of each meal were assessed by video monitoring if possible (Supplementary Table 4). In addition, we divided the daytime into morning and afternoon periods by excluding mealtimes. The morning was defined as the period from 8:00 to 11:59, and the afternoon was defined as the period from 13:00 to 17:59.

We revised the second paragraph of the updated Discussion section as follows:

Although we did not investigate SWR activity during the sleep stage due to the lack of electrooculogram and electromyogram data, which are essential to identify the sleep stages, the clear circadian rhythm of the SWRs, which increased during the night, is consistent with the changes in SWR activity during the sleep-wake cycle observed in animal studies.

Second, Figure 2f shows the mean and the 95% CI of the mean SWR event rate for all patients, not only one patient. We used a one-way ANOVA to test for significance in the overall variation over a 24-hour period. This point was not well articulated in the original submission, and we have added this information to the Results section to rectify that oversight as follows.

Results section (Line 9, page 7)

The SWR rate showed significant overall variation over a 24-hour period (Figure 2f; $P < 0.001$, $F_{143,83392} = 41.6$, $n = 113760$ time points from a total of 79 days in 10 patients; one-way ANOVA).

Finally, we performed multiple comparisons post-ANOVA to determine which times showed significant increases or decreases in the SWR rate (see Supplementary Figure 8).

Supplementary Figure 8 (b)

Post hoc multiple comparisons of the hourly SWR event rates of all cases ($n = 10$). The circle and bar represent the mean and confidence interval for the SWR event rate for one hour. The nonoverlapping confidence interval among circles shows the significant difference between the circles.

Comment 4-7

- For the finding comparing SWR rates between sleep and daytime, please report the mean/standard deviation of ripple rates.

Response to comment 4-7:

To directly address the reviewer's suggestion, we added information to elaborate on the mean SWR rates. Specifically, the normalized SWR event rate during the night was found to be significantly higher than that during daytime for all patients (daytime, -0.153 ± 0.918 (mean \pm SD); night, 0.234 ± 1.072 ; $P < 0.001$ T_{116186} : -65.58 , $n = 156960$ time points; student's t test). We plotted a cumulative frequency distribution to better represent these data, which has been added to the supplemental data of the manuscript.

We updated the Results section as follows:

In particular, the SWR event rate was significantly higher during the night (0.234 ± 1.072 counts/min

(mean \pm SD)) than during the daytime (-0.153 ± 0.918) ($P < 0.001$, $t_{116186} = -65.58$, $n = 156960$ time points; student's t test).

Supplementary Figure 15

The cumulative frequency distribution of the SWR event rates for nighttime and daytime is shown (the blue line shows the nighttime, and the red line shows the daytime results).

Comment 4-8

- It is not clear to me what the authors mean by "To conduct the regression analysis, we predetermined the total number of answers on questionnaires as more than 170 samples, which is equivalent to 10 times the number of questionnaire items, totaling 17." Does this mean that if the patient filled out more than 10 surveys (which would seemingly correspond to 10 hours only), any additional responses were discarded? Also, could the authors please clarify how the predetermined value of 170 samples relates to how they conducted the regression analysis?

Response to comment 4-8:

The reviewer highlighted a lack of clarity in our explanation regarding how we determined the total number of answers for the questions in the questionnaire used in our regression analysis. We apologize for any confusion our wording may have caused.

To clarify, each questionnaire included 17 questions. We calculated that we would need a target sample size of 170, which is 10 times the number of questionnaire items, to have a sufficient sample size for performing a regression analysis of the SWR event rate based on these 17 items. The 170 target sample size was not per subject but was combined across subjects. In practice, we collected more than the target number of 170 samples. If a patient completed more than 10 surveys, we did not discard the additional responses. Instead, these data were included in our analysis.

We have clarified this point in the Methods section of the revised manuscript to ensure that the methodology is clear.

Method section (Line 479 page 24)

We calculated that the target sample size for performing a regression analysis of the SWR event rate based on the 17 questionnaire items was 170, which is 10 times the number of questionnaire items. The 170 target sample size was not per subject but was combined across patients.

Comment 4-9

- Anti-seizure medicines (ASMs) may affect ripple rates, and are often withdrawn, held, or restarted during varying parts of EMU admissions. Were changes in SWR rates associated with timing of ASM administration (including the changes observed at mealtimes), and is any of the variability in SWR rate identified with ASM administration?

Response to comment 4-9:

The reviewer raised an important concern about the impact of ASMs on ripple rates, particularly given that these medications are often adjusted during EMU admissions. We understand this concern from Reviewer #2 and have included additional data in Supplementary Table 3 to show the variations in the ASMs during the period of intracranial recording. First, we conducted separate analyses comparing SWR event rates on days with the lowest amount of ASMs with those on the other days. There were no consistent variations in the SWR event rates related to the amount of ASMs (Supplementary Figure 22).

Supplementary Figure 22

SWR rates on the day with the lowest dose of antiepileptic drugs were compared with those on other days, and there was no significant difference ($P = 0.74$, $t_6 = 0.34$, $n = 7$ subjects; paired t test).

Subjects with no increase or decrease in the dose of antiepileptic drugs were excluded from this analysis.

Furthermore, although the timing of ASM administration often coincides with mealtimes, as Reviewer #2 pointed out, they are generally administered after the meal. In our recordings, no patient was administered their ASM before mealtimes. On the other hand, the SWR event rates decreased at the start time of the meal. Therefore, it is difficult to explain the decreases in SWR activity with ASM administration. In the revised manuscript, we determined the start time of each meal based on the video recordings. Then, the SWR event rates were assessed based on the exact time that the meal began; that is, we confirmed that the decreased SWR event rates preceded the medication administration with this additional examination. In addition, the SWR event rates were higher after the meal than during the meal, although the blood levels are expected to increase after ASM administration. Therefore, given the lack of a consistent pattern in relation to ASM levels, we suggest that the lower SWR rates at mealtimes may not be directly attributable to ASM administration.

We have incorporated these additional analyses and findings into Supplementary Figure 12 and Supplementary Tables 3 and 4 to provide a more comprehensive understanding of the influence, or lack thereof, of ASMs on SWR rates. In addition, to clarify the influence of ASM administration on epileptic activity, we revised the limitation section in the Discussion as follows.

Line 268 page 14

The frequency of the epileptic activity and excluded periods did not explain the diurnal variation in the SWR (Supplementary Figure 10). In addition, although the frequency of interictal epileptic activity was weakly explained by the thought sampling results, the model weights differed from those of the SWR model (Supplementary Figure 17). Finally, although the patients took antiepileptic medication during mealtime, the SWR event rates decreased at the start of the meal and before the administration of medicines (Supplementary Figure 12). These results suggest that the epileptic-related activities were not directly attributed to the relation between SWRs and thought content.

Comment 4-10

- Regarding the linear regression analysis for thought content: Given that the 17 questions in the survey seem to be collinear, does this inflate the variance of the regression coefficients and is there a need for regularization?

Response to comment 4-10:

In response to the reviewer's suggestion, we conducted a lasso regression with 10-fold cross-validation to address the potential for inflated variance due to collinearity. Although the method was significantly changed, the conclusions were not changed. The SWR event rate was significantly predicted by 7 items related to thought sampling, with the highest contribution from "External-task Focus". The results of this analysis have been added to Supplementary Figure 21 as follows. However, we used the generalized linear mixed-effect model for the main analysis.

Supplementary Figure 21

(a) Predictions and actual standardized SWR rates predicted from responses to questions about emotions and thoughts according to the regression analysis using Lasso regression. (b) The coefficient of each feature in the regression model.

Comment 5-1

Final (minor) suggestions:

- In Supplementary Table S1, please report where the seizure focus was for each of the 11 patients, and

the results of available of their MRI (lesional/non-lesional), seizure semiology, MEG (if available), PET (if available), and scalp EEG.

Response to comment 5-1:

In response to the reviewer's suggestion, we have included a more comprehensive set of clinical information in Supplementary Table 2. This revised table reports the seizure focus and the MRI, seizure semiology, PET scan, and scalp EEG results for each patient in the study.

Comment 5-2

- I wonder if the term "freely behaving humans" is appropriate, given that people are not really freely behaving in EMU admissions and these settings typically are not thought to reflect the individual's natural environment (as the authors allude to in their limitations section). Moreover, ASMs are often withdrawn or held during these admissions, which changes the rate of interictal epileptiform discharges and may affect SWR rates as well as relative proportions of pathological and physiological ripples. It would be more accurate to describe what the environment in this study captures, which is SWRs during rest and wake during a controlled environment, or use the term "freely moving humans" (although this is also probably not true, since they are usually tied to a hospital monitor via their EEG electrodes and/or telemetry and IVs).

Response to comment 5-2:

We thank the reviewer for noting this important point regarding the terminology "freely behaving humans" used in our manuscript. We appreciate the reviewer's detailed observation about the controlled environment of EMU admissions and the restrictions imposed on patients due to EEG electrodes and/or telemetry and IVs.

The term "freely behaving humans" may not accurately represent the environment and conditions under which the data were collected. Therefore, we revised the term "freely behaving humans" to "during long-term EEG monitoring" to reflect the study setting more appropriately. This change has been made throughout the manuscript to ensure consistency.

Comment 5-3

- Methods, Participants: "...to treat epilepsy..." □ "for presurgical localization" (?)

Response to comment 5-3:

Following the reviewer's suggestion, we have revised all instances of "to treat epilepsy" to "for presurgical evaluation of seizure onset zones and memory function" throughout the manuscript.

Comment 5-4

- Throughout the manuscript, "deep electrodes" □ "depth electrodes"

- In the results and discussion sections, the authors mention that their findings "establish" that SWRs are related to patterns of ongoing thought; a word such as "suggest" might be more appropriate.

Response to comment 5-4:

Following the reviewer's suggestion, we have revised all instances of "deep electrodes" to "depth electrodes" throughout the manuscript.

In addition, following the reviewer's recommendation, we have changed the word "establish" to "suggest". This change better encapsulates the level of certainty that our current dataset can provide and is more fitting for the nature of our findings.

Comment 5-5

- It is not clear which measure "desired experiences" corresponds to in the scale. Is this the "unwilling" dimension?

Response to comment 5-5:

We apologize for the unclear description in the previous manuscript. The term "desired experiences" actually corresponds to what we labeled the "unwanted" aspect in the self-report scale. Specifically, the "unwanted" aspect is designed to assess whether participants wanted to engage in the thoughts they were having. To remove any ambiguity, we have clarified this point in the Results section of the manuscript.

Comment 5-6

- I would suggest the authors provide additional context to the significance of any findings with regards to existing literature about non-local representations and relationship to SWRs during locomotion in rodents. It would also be interesting if the authors could report if there are characteristics of the SWRs that were more associated with local thought content (high task focus?) than with non-local thought content (low task focus?).

Response to comment 5-6:

We thank the reviewer for their insightful comments regarding the significance of our findings in relation to the existing literature, particularly concerning nonlocal representations and the relationship of SWRs to locomotion in rodents.

In this study, our results demonstrated that the SWR event rate increases when the patients' thoughts include content not related to the task, such as mind wandering. This finding is consistent with the SWR event rate increasing when rodents are in nonexploratory states. On the other hand, some previous studies have demonstrated that SWRs play active roles in future planning and imagination in rodents and humans^{1,2}. These functions are associated more with local thought content than nonlocal thought content.

Our finding showing the relation between the SWR event rate and mind wandering appears to be opposed to this active function of SWRs. However, our findings also support such active functions of SWRs. The SWR event rates were explained not only by the task focus but also by the vividness, imaginativeness and unwantedness of thoughts, with significant coefficients. This result suggests that SWRs have multiple functions. Although it is difficult to explain the causal functional role of SWRs based on the results of our study, our results demonstrated the multiple aspects of SWRs using our designed questionnaire. To clarify this point, we revised the third paragraph of the Discussion as follows.

Third paragraph of discussion

Finally, SWRs are also important in laboratory tasks that mimic naturally occurring self-generated thought, including future planning, memory recall, and imagination^{17,18}—situations that depend on the self-generation of mental content and can also include a reduction in the perceptual processing of external input¹⁹ (see Supplementary Figure 13). To understand the associations between the SWR rate and ongoing thought patterns, we employed experience sampling, including mDES scores⁶. This approach provides a complementary method to understanding brain function for studies that employ specific tasks to understand the association between brain activity and cognition⁴¹. Although our study is the first to use mDES scores to relate thought patterns to SWRs, prior studies have shown that mDES scores allow researchers to reliably categorize thought patterns in different situations in the lab⁷ and daily life^{8,9}. In the laboratory, mDES scores have been shown to effectively characterize the association between brain activity and thought patterns that reflect task-relevant processes¹⁰, off-task social episodic thoughts during tasks¹¹ and thoughts that emerge at rest¹².

1. Norman, Y. et al. Hippocampal sharp-wave ripples linked to visual episodic recollection in humans. *Science* (1979) 365, (2019).
2. Leonard, T. K. et al. Sharp Wave Ripples during Visual Exploration in the Primate Hippocampus. *J Neurosci* 35, 14771–82 (2015).

Comment 5-7

- Data sharing statement: I suggest that authors consider sharing their data using NWB file format on

DANDI.

Response to comment 5-7:

Although we cannot open the raw EEG data in this study based on the informed consent obtained from our participants, we uploaded data on the frequency of SWR for each patient and the MATLAB code to detect SWR to Figshare so that everyone can reproduce our results using the data.

Reviewer #3 (Remarks to the Author):

General comments

In 11 individuals post epilepsy surgery with electrodes implanted in the hippocampus or parahippocampal gyrus, the authors measure the occurrence of sharp wave ripple (SWR) events and their relationship to ongoing thought patterns, ongoing over the course of 9 to 11 days. They reveal changes in SWR frequency across the day-night cycle, and across the course of the day. They further show that SWR rates were associated with responses on a thought questionnaire, which was probed hourly. In particular, the thought dimensions related to SWR rates involved less external task focus and vivid mental content – supporting emerging hypotheses that SWRs may be one avenue to a brain state that supports internally directed thought, or mind wandering.

This is a compelling study that further extends our understanding of SWR occurrence into the human domain, and most notably links them with ongoing behaviour and thought patterns. I have the following comments:

Response to the comments:

First, we would like to extend our sincere gratitude to the reviewer for taking the time to review our manuscript and for acknowledging the compelling nature of our study. The reviewer's feedback is invaluable to us, and we sincerely appreciate the insight and perspective the reviewer brings to our work.

Before we address each specific comment, we would like to emphasize the novelty and importance of our findings, particularly the association of SWR events with ongoing thought patterns in humans. As noted by the reviewer, this research aids in extending our understanding of SWRs in humans.

We next present our responses to the reviewer's comments.

Comment 1

Prior to detecting SWRs in the data, the authors excluded periods associated with epileptic activity. This issue is considered as a limitation in the Discussion, as the authors explain that although they've excluded the effects of epileptic activity, it remains unclear whether epileptic activity might have influenced hippocampal activity or led to changes in ongoing thought patterns.

The possible influence of epileptic activity is of course inherent in this study population. However, given the nature of the study question I wonder if it deserves more careful attention. I am thinking of the possibility that not only might epileptic activity influence the hippocampal EEG signal, but epileptic activity might have subjective (conscious) effects on arousal, mood or thought content – which may relate directly to the dimensions captured in the questionnaire.

Perhaps some data might speak to this issue. Have the authors considered looking at whether the timings of periods of epileptic activity 1) were associated with subsequent increases/decreases in SWR frequency; and 2) if epileptic activity occurred in the 2 to 7 min window preceding questionnaire responses, whether this was associated with distinct thought characteristics?

Response to comment 1:

The reviewer has raised an important issue regarding the potential impact of epileptic activity on both SWR frequency and thought patterns. To address this point, we analyzed the mean and standard deviation of the SWR rate for a 30-minute window before and after each seizure segment ($n = 177$). Notably, the different seizure segments have varied lengths, with a mean duration of 19 minutes. Our

analysis revealed no significant changes in the SWR rate before and after seizures (see Supplementary Figure 16; $R = 0.38 \pm 0.25$ for the pre-seizure period and 0.36 ± 0.19 for the post-seizure period; $t(4755) = 0.5464$; $P = 0.5848$; paired t test), suggesting the low impact of seizures on the SWR event rate in this study. We included this result in Supplementary Figure 16 and discussed this point in the Results section on line 130, page 8 as follows.

However, the SWR event rates for 30 minutes before and 30 minutes after the seizure periods were not significantly different, suggesting that epileptic seizures did not significantly affect the SWR event rates (see Supplementary Figure 16; SWR rates 30 minutes before seizure were 0.38 ± 0.25 , and those after seizure were 0.36 ± 0.19 ; $t_{4755} = 0.5464$; $P = 0.5848$; paired student's t test).

Supplementary Figure 16

Plot of the mean and 95% CI of the SWR event rate (Z-score) 30 minutes before and after the interval excluded for containing epileptic seizures ($n=177$ among all patients). The left and right panels show the SWR rates for the pre-epileptic seizure and post-epileptic seizure periods, respectively.

Additionally, we utilized a general linear mixed-effects model to explain the frequency of interictal epileptic discharges occurring 2-7 minutes before the questionnaire responses according to the thought sampling scores. While the model showed a weak correlation ($R^2=0.039$, $P > 0.05$ permutation test), the accuracy was not significantly larger than chance ($P > 0.05$ permutation test) and was significantly smaller than that explaining the SWR event rate using the thought sampling scores. Moreover, only the 'Spontaneous' aspect in the thought sampling questionnaire made a significant contribution to the model. These findings suggest that epileptic activity has a different impact than SWRs on the thought characteristics captured in our questionnaire. We added these results to Supplementary Figure 17. To clarify the effects of epileptic activity on the SWR event rate and the relation between the SWR event rate and thought contents, we revised the limitation section of the Discussion as follows.

Supplementary figure 17

(a) The frequency of standardized interictal epileptic discharges (IEDs) 2-7 minutes prior to questionnaire response was regressed using a mixed-effects model based on the questionnaire response results ($R^2=0.04$). The coefficients for each question are shown. Only the item “spontaneous” contributed significantly to the results. (b) The results of the permutation test. The order of the predicted IED frequencies was switched randomly and regressed 1000 times to determine the distribution of the coefficient of determination. The results show that the regression analysis of the IED frequency is not significant ($P > 0.05$).

Discussion (Line 266, page 14)

First, while we carefully excluded the effects of epileptic activity, it remains unclear whether epileptic activity influenced hippocampal function or led to changes in ongoing patterns of thought. The frequency of the epileptic activity and excluded periods did not explain the diurnal variation in the SWR (Supplementary Figure 10). In addition, although the frequency of interictal epileptic activity was weakly explained by the thought sampling results, the model weights differed from those of the SWR model (Supplementary Figure 17). Finally, although the patients took antiepileptic medication during mealtime, the SWR event rates decreased at the start of the meal and before the administration of medicines (Supplementary Figure 12). These results suggest that the epileptic-related activities were not directly attributed to the relation between SWRs and thought content.

Comment 2

Regarding SWR detection: ripples with durations shorter than 20 ms or longer than 200 ms were rejected, and peaks closer than 30 ms were concatenated. I wondered that given their rich dataset, whether the authors looked in more detail at characteristics of SWR events.

Based on rodent literature, we know that longer duration of ripple events has implications for memory performance (e.g., Science, 364(6445), 1082-1086 <https://www.science.org/doi/10.1126/science.aax0758>). There is also the idea that ripple bursts or

clusters (i.e., temporally close trains of events) might have a distinct functional relevance, as compared to isolated events (e.g., see reviewed in *Hippocampus*, 25(10), 1073-1188 <https://onlinelibrary.wiley.com/doi/full/10.1002/hipo.22488>). Did the authors explore whether 1) duration of SWRs; or 2) clustered vs. isolated SWRs, had distinct relationship with the behavioural or thought patterns they investigated? This seems like it might be useful information to both enrich their findings and make even closer links with the rodent literature.

Response to comment 2:

We sincerely appreciate the reviewer's thoughtful comments and questions on the methodologies employed in our paper, specifically concerning the characterization of SWRs in terms of their duration and clustering characteristics.

First, we acknowledge the importance of examining the duration of SWR events in more detail. As the reviewer pointed out, longer ripple events in rodents have been found to be crucial for memory performance. To address this suggestion, we carried out further analyses to better understand these properties within our dataset. We analyzed the average SWR duration in 10-minute bins, which was normalized daily and averaged across all patients, similar to our frequency analysis. Our findings demonstrated diurnal fluctuation similar to those observed for the frequency (see Supplementary Figure 18). We included these results in Supplementary Figure 18 in the Results section (page 7, line 113). Furthermore, our linear mixed-effects model regression analysis of the mean duration from 5 minutes before to the time at which the thought sampling questionnaire responses were collected revealed a high degree of precision (R^2 : 0.191). Interestingly, Focus was a significant negative contributor, which aligns well with our previous frequency analyses.

In contrast, when we regressed the SWR duration based on the physical activity data, the model was not as precise (R^2 : 0.0066; see Supplementary Figure 18).

Supplementary Figure 18

(a) The diurnal fluctuation in the SWR duration is shown (black points indicate the mean values, gray color indicates the 95% CI). (b) Coefficients for each measure in the regression analysis predicting the standardized duration of the SWR according to biological data using a mixed-effects model. (R^2 : 0.191) (c) Coefficients for each measure in the regression analysis predicting the standardized duration of the SWR according to biological data using a mixed-effects model. (R^2 : 0.0066)

Furthermore, we appreciate the reviewer's suggestion to explore the difference between clustered and isolated SWRs. For this, we used a criterion of 250 msec between ripples to classify events as either clustered or isolated, based on previous literature¹. The average frequency of both clustered and isolated SWRs in 10-minute bins was normalized daily and averaged across all patients, again showing diurnal fluctuations (Supplementary Figure 19). A linear mixed-effects model regressed based on the mean frequency of each SWR type according to the questionnaire answers showed significant results (R^2 : 0.485 for the clustered SWRs and R^2 : 0.567 for the isolated SWRs). Notably, External-task Focus was also a significant contributor in these analyses. We added these results to Supplementary Figure 19 in the Results section (page 7, line 113).

Supplementary Figure 19

Diurnal fluctuations in frequency when the interval between SWRs was 250 msec or less were used to classify clustered SWRs, while those above were classified as isolated SWRs (a: clustered SWRs b: isolated SWRs). The normalized SWR event rates were explained by the 17 answers to the questionnaires using the mixed-effect model (R^2 : 0.485; clustered SWR (c) and R^2 : 0.567; isolated SWR (d)).

1. Noguchi, A., Matsumoto, N. & Ikegaya, Y. Postnatal Maturation of Membrane Potential Dynamics during in Vivo Hippocampal Ripples. *The Journal of Neuroscience* 43, 6126–6140 (2023).

Comment 3

The authors raise the point in the Discussion that neuromodulators are likely to be an important determinant of the SWR brain state. It made me wonder whether the study participants were on any postoperative medications that might be relevant to consider regarding SWR modulation. I didn't find any mention of current medications for the study participants, perhaps this information is worth including?

Response to comment 3:

The reviewer pointed out the possible role of postoperative medications in modulating the SWR brain state. To address this concern, we have included Supplementary Table 3 in the revised manuscript, which lists the antiepileptic medications taken by the participants before and after electrode implantation. Additionally, we compared the mean SWR rate on the day of the lowest medication dose with the mean SWR rates on the other days. Our analysis indicated inconsistent results, with some cases showing higher SWRs and others showing lower SWRs on the day of the lowest dose (Supplementary Figure 22).

Supplementary figure 22

SWR rates on the day with the lowest dose of antiepileptic drugs were compared with those on other days, and there was no significant difference ($P = 0.74$, $t_6 = 0.34$, $n = 7$ subjects; paired t test).

Subjects with no increase or decrease in the dose of antiepileptic drugs were excluded from this analysis.

Supplementary Table 3. Information on the amount of anti epileptic drug and food intake for each patient

Pt ID	Preoperation	Day 1	Day 2	Day 3	Day 4	Day 5	Day 6	Day 7	Day 8	Day 9	Day 10	Day 11	Day 12	Day 13
1	LEV 1000	500	1000	1000	750	500	750	1000	→	→	→	→	→	→
	LTG 200	→	→	→	→	→	→	→	→	→	→	→	→	→
	LCM 400	0	→	→	→	→	→	→	→	→	→	→	→	→
	CLB 10	→	→	→	→	→	→	→	→	→	→	→	→	→
	Breakfast	Main dish	1	1	4	6	4	6	6	6	6	6	6	6
		Side dish	2	2	6	6	6	6	6	6	6	6	6	6
	Lunch	Main dish	1	3	6	6	6	6	6	NaN	6	6	6	NaN
		Side dish	1	3	6	4	6	6	6	NaN	6	6	6	NaN
	Dinner	Main dish	1	3	6	4	6	6	6	6	6	6	6	6
		Side dish	1	6	6	4	6	6	6	6	6	6	6	6
	LEV 1000	→	→	→	→	→	→	→	→	→	→	→	→	→
	CBZ 450	→	→	→	→	→	→	→	→	→	→	→	→	→
	VPA 700	→	→	→	→	→	→	→	→	→	→	→	→	→
	ZNS 300	→	→	→	→	→	→	→	→	→	→	→	→	→
	CLB 30	→	→	→	→	→	→	→	→	→	→	→	→	→
2	Breakfast	Main dish	1	1	5	4	1	6	4	6	6	6	6	6
		Side dish	6	6	6	6	5	6	5	6	6	6	6	6
	Lunch	Main dish	4	4	2	6	2	6	1	6	6	6	6	6
		Side dish	2	4	4	6	5	6	6	6	6	6	4	6
	Dinner	Main dish	4	4	1	1	6	2	1	6	6	1	6	6
		Side dish	4	4	6	4	6	5	6	6	6	6	6	6
3	LEV 2500	2000	→	→	→	1500	1000	→	→	→	2500	→	→	→

LCM	250	→	200	→	250	150	→	→	→	→	250	→	→	→
CLB	3	→	→	→	→	→	→	→	→	→	→	→	→	→
Breakfast	Main dish	6	6	6	6	6	6	6	6	6	6	6	6	6
	Side dish	6	6	6	6	6	6	6	6	6	6	6	6	6
Lunch	Main dish	4	6	6	6	6	6	6	6	6	6	6	6	6
	Side dish	6	6	6	6	6	6	6	6	6	6	6	6	6
Dinner	Main dish	6	6	6	6	6	6	6	6	6	6	6	6	6
	Side dish	6	6	6	6	6	6	6	6	6	6	6	6	6

LEV: levetiracetam, LTG: lamotrigine, LCM: lacosamide, CLB: clobazam, CBZ: carbamazepine, VPA: valproic acid, ZNS: zonisamide, PER: perampanel. The numbers indicated in the antiepileptic drug column represent the daily dosage in milligrams (mg). The numbers indicated in the meal intake column represent the daily consumption level, categorized into 7 levels: 0: No intake 1: 10-20% intake 2: 30% intake 3: 40% intake 4: 50-60% intake 5: 70-80% intake 6: 90-100% intake.

Supplementary Table 5. Information on the amount of anti epileptic drug and food intake for each patient

Pt ID	Preoperation	Day 1	Day 2	Day 3	Day 4	Day 5	Day 6	Day 7	Day 8	Day 9	Day 10	Day 11	Day 12	Day 13	Pt ID	
7	LEV	1000	750	250	0	750	500	250	→	750	500	250	→	→	0	500
	LTG	125	75	25	→	50	25	→	→	50	→	25	→	50	0	25
	Breakfast	Main dish	6	6	6	4	6	6	NaN	6	6	6	6	6	6	6
		Side dish	6	6	6	4	6	6	NaN	6	6	6	6	6	6	6
	Lunch	Main dish	6	6	6	3	6	6	6	6	6	6	6	6	6	6
		Side dish	6	6	6	4	6	6	6	6	6	6	6	6	6	6
	Dinner	Main dish	6	6	6	4	6	6	6	6	6	6	6	6	6	6
		Side dish	5	6	6	4	6	6	6	6	6	6	6	6	6	6
	LCM	400	200	200	100	0	100	100	0	200	400					
	VPA	1000	600	400	400	200	200	300	100	400	800					
	CLB	10	5	5	0	0	0	5	0	0	20					
8	Breakfast	Main dish	1	6	6	6	6	6	6	1	6					
		Side dish	1	6	6	6	6	6	6	1	6					
	Lunch	Main dish	5	6	6	6	6	6	6	6	6					
		Side dish	5	1	6	6	6	6	6	6	6					
	Dinner	Main dish	6	6	6	6	6	6	6	6	6					
		Side dish	6	6	6	6	6	6	6	6	6					
	LEV	3000	1500	0	→	→	→	→	→	1500	3000	→	→	→	→	→
	CBZ	500	500	→	→	→	→	→	→	→	→	→	→	→	→	→
9	Breakfast	Main dish	1	1	2	6	6	6	6	6	6	2	4	1		
		Side dish	1	1	2	6	6	4	2	2	9	6	2	6	1	
	Lunch	Main dish	1	2	4	4	6	6	4	6	NaN	6	NaN	6	NaN	
		Side dish	2	1	2	3	6	6	2	6	NaN	6	NaN	6	NaN	
	Dinner	Main dish	1	1	6	4	6	4	4	6	6	6	4	NaN	1	
		Side dish	1	1	5	4	4	3	2	6	1	6	6	NaN	1	
	LEV	2250	2250	1500	1000	→	1750	2250	2250	→	→	→	→	→	→	→
	VPA	800	800	→	→	→	→	→	→	→	→	→	→	→	→	→
	LCM	400	400	→	→	→	→	→	→	→	→	→	→	→	→	→
	PER	4	4	→	→	→	→	→	→	→	→	→	→	→	→	→
10	Breakfast	Main dish	2	6	6	4	4	6	6	6	6	2	6			
		Side dish	4	4	6	6	6	6	6	6	6	6	6			
	Lunch	Main dish	4	6	6	6	6	6	6	6	6	6	6			
		Side dish	3	3	6	6	6	6	6	6	6	6	6			
	Dinner	Main dish	4	5	6	6	6	6	6	5	6	6	6			
		Side dish	4	5	6	4	6	6	6	5	6	6	6			

Comment 4

On the second page of the Discussion, the authors relate their findings to hippocampal lesions and reductions in mind wandering, as well as grey matter in the hippocampus of healthy people. It may be of interest that further convergent evidence in support of the link between hippocampal integrity and mind wandering/off-task thought has also been shown in dementia, where grey matter integrity in the hippocampus/parahippocampus has been linked with capacity for mind wandering (PNAS 116(8), 3316-3321 <https://www.pnas.org/doi/abs/10.1073/pnas.1818523116>).

Response to comment 4:

We are grateful for the reviewer's insightful suggestion to consider the findings presented in previous dementia studies, which supports the link between hippocampal integrity and mind wandering/off-task thinking. We have incorporated this point into our Discussion section and cited the referenced article¹ accordingly. We believe this inclusion enriches the multidisciplinary understanding of the role of the hippocampus in mind wandering.

Line 216 page 12

In addition, gray matter integrity in the hippocampus/parahippocampus has been linked with the capacity for mind wandering in patients with dementia³⁵.

1. Noguchi, A., Matsumoto, N. & Ikegaya, Y. Postnatal Maturation of Membrane Potential Dynamics during in Vivo Hippocampal Ripples. *The Journal of Neuroscience* 43, 6126–6140 (2023).

Comment 5

A small point in the Introduction: SWRs are defined as "bursts of synchronized neuronal activity in the mammalian hippocampus". Fascinatingly, these rhythms appear to be conserved outside of the mammalian brain, with evidence for them in lizards (Shein-Idelson et al. <https://www.science.org/doi/abs/10.1126/science.aaf3621>) and in birds (Yeganegi et al. <https://www.biorxiv.org/content/10.1101/825075v2.abstract>). So this statement could be amended.

Response to comment 5:

We appreciate the reviewer drawing our attention to this finding. Indeed, the occurrence of SWRs extends beyond the mammalian hippocampus, as demonstrated in studies on lizards and birds. To accurately reflect this broader scope, we have revised the relevant statement in the Introduction to read "bursts of synchronized neuronal activity in the hippocampus," thereby omitting the qualifier "mammalian." as follows:

Line 57 on page 4 (Introduction)

SWRs are bursts of synchronized neuronal activity in the hippocampus that vary depending on the state of the animal.

REVIEWER COMMENTS

Reviewer #1 (Remarks to the Author):

I thank the authors for the revised version of their manuscript. All my comments have been thoroughly addressed and the paper appears now clearer and stronger. I am even more convinced now that this is an excellent study that could have a significant impact on the field of cognitive neuroscience.

I noted only a minor typo in the revised manuscript (line 259, the authors wrote 'data waves' but I assume they meant 'delta waves').

Congratulations for this work!

Reviewer #2 (Remarks to the Author):

I appreciate the authors' thoughtful responses to my questions in the prior review. My primary concerns in the initial submission were regarding (1) whether the authors' findings are likely to be interpretable with respect to physiological SWRs, and specifically whether pathological ripples could reasonably have been assumed to be excluded, (2) concerns regarding the study design, as the ability to physiologically associate ripple rates with the reported thought content in a survey response that occurs several minutes after the recorded ripple rate is not clear, and (3) concerns about overstatement of significance. My concerns regarding the third concern have been addressed. However, my concerns regarding the first and second concerns remain.

For the first concern, the authors could do more to demonstrate that the ripples they are analyzing are similar to those examined in previous studies. Specifically, they could plot the amplitude, duration, and frequency associated with the peak power for each event. While they do use a bandpass filter to limit the events, it's not clear whether the events fall into the lower or upper part of the included range, making it impossible know whether events that may reflect more pathological states are or are not included.

For the second concern, the authors have not provided convincing evidence as to why ripple rates occurring several minutes preceding the time of mind wandering can be physiologically associated with mind wandering several minutes afterwards. Moreover, the graph the authors have shown actually

seems to suggest that ripple rates are generally high over a broad range of times up to 11 minutes preceding the survey response, dropping at the time of survey response itself. The timescale is more supportive of a lagged association between ripple rate and mind wandering rather than a direct association, which may be more suggestive, rather, of an association of mind wandering with some underlying brain state which changes more slowly and which is preceded by high ripple rates. I do not think that the second concern can be mitigated short of either (1) changing their primary analysis to be based on the association of ripple rate during the response itself, to the response, or (2) a different behavioral study design.

This point is critical for the authors interpretation, so I want to make sure it is clear. To explain why SWR rates were taken from 2-7 minutes preceding the response, the authors have provided analysis showing the coefficient of determination between SWR rate and the survey response was significantly higher than a random permutation during this interval. However, the plot seems to in fact show that the SWR rate is quite diffusely high from -1 to -11 minutes before the survey response, and that it (interestingly) drops at the time of the actual survey response (at time 0). Moreover, based on the clarified instructions in the methods section about the "External task focus" question, it seems that the "task" being referred to in the "external-task focus" question would have been assumed to be the act of responding to the survey question on the tablet, which occurs long after the ripple rates being analyzed. It seems an incorrect timescale to assume that SWR rates up to 10 minutes before a survey response about mind wandering can physiologically temporally linked.

To address this concern, short of a different study design, I think the authors would need to analyze the relation of SWR rate during the response itself with the response as their primary analysis (rather than the relation of SWR rate 2-7 minutes before the response with the response).

As a related question, why does the coefficient of determination drop at time 0 (which presumably is the time of the survey response)? If it is in fact true that the coefficient of determination is high before the response itself but drops during the response, this seems to warrant explanation.

I thank the reviewers for conducting additional experiments to assess whether SWR rates are higher during tasks that assess mind-wandering. I find this helpful to see and think that if this study design is to be used in the future, there could be a few thoughts the authors might consider including in their reporting of this finding, including: 1) Other than the fact that generally backward subtraction is used to assess attention and focusing on breathing generally associated with mind wandering, it would be helpful to quantify the degree of mind wandering during the backward subtraction task (not only the degree of mind wandering during focusing on breathing) in order to state that mind wandering was greater during the task involving focusing on breathing. 2) Since these are two distinct tasks, it would be useful to know why SWR rates can be attributable to the difference in mind wandering rather than another difference between the two tasks, such as the complexity of the task, or the memory requirements of the task.

Minor additional comment:

I appreciate the authors' addition of a rose plot in Supplementary Figure 8 to show changes in ripple rate across time. Their plot suggests that detected ripples were higher particularly between ~10pm and shortly before midnight, between approximately midnight and approximately 1 am, and around 1-2pm, and that there were dips around 7-8 am, 12-1pm, and around 6-7pm. Combined with the authors' analysis of SWR rate before and after the actual start of mealtimes, I agree that the data shows evidence that SWR rate drops during mealtimes, which is quite interesting, particularly with relation to rodent literature which the authors have discussed.

I do think the rose plot shows additional interesting findings regarding the circadian changes in ripple rates, which could be considered for inclusion in the discussion. With regards to the discussion of ripple rates during afternoon versus morning, it appears there is more precisely a relative peak in SWR rate around 1-2:30pm, with the remainder of the afternoon appearing quite similar to the morning time. It might be helpful to highlight this in the discussion, and also to consider whether there was something occurring between 1-2:30pm that would have caused this. (For example, is this due to a rebound in ripple rates after mealtimes?) Similarly, with regards to the nighttime increase change in ripple rate, it seems this increase is primarily driven by two peaks in ripple rate early in the evening (between ~10pm and shortly before midnight, and another peak between approximately midnight and approximately 1am). It would be helpful to discuss possible sources of these two peaks.

Reviewer #3 (Remarks to the Author):

The authors have thoroughly addressed my comments. Congratulations on a very interesting study.

Point-by-Point Responses to Reviewers' Comments
Hippocampal sharp-wave ripples correlate with periods of naturally occurring self-generated thoughts in humans
(NCOMMS-23-27587-T)

We are grateful for the reviewers' insightful feedback and suggestions. The constructive critiques have significantly aided in refining and enhancing our manuscript. We have diligently addressed each concern raised, and in the revised manuscript, all additions and modifications are highlighted in yellow. Below, we provide a detailed response to each comment, with the reviewers' remarks presented in grey.

Responses to Reviewers' Comments

Reviewer #1 (Remarks to the Author):

General comment:

I thank the authors for the revised version of their manuscript. All my comments have been thoroughly addressed and the paper appears now clearer and stronger. I am even more convinced now that this is an excellent study that could have a significant impact on the field of cognitive neuroscience.

I noted only a minor typo in the revised manuscript (line 259, the authors wrote 'data waves' but I assume they meant 'delta waves').

Congratulations for this work!

Response to the comments:

We would like to extend our sincere gratitude to the reviewer for the time and effort they have dedicated to reviewing our manuscript. We have corrected the typographical error the reviewer pointed out. We look forward to the possibility of our manuscript making a significant impact in the field.

Reviewer #2 (Remarks to the Author):

General comments

I appreciate the authors' thoughtful responses to my questions in the prior review. My primary concerns in the initial submission were regarding (1) whether the authors' findings are likely to be interpretable with respect to physiological SWRs, and specifically whether pathological ripples could reasonably have been assumed to be excluded, (2) concerns regarding the study design, as the ability to physiologically associate ripple rates with the reported thought content in a survey response that occurs several minutes after the recorded ripple rate is not clear, and (3) concerns about overstatement of significance. My concerns regarding the third concern have been addressed. However, my concerns regarding the first and second concerns remain.

Response to the comments:

We thank the reviewer for this detailed review of our manuscript. The reviewer's insightful feedback is invaluable in enhancing the quality and clarity of our research. We appreciate the recognition of our efforts to address the initial concerns regarding our study.

We understand that two primary concerns remain: (1) the interpretability of our findings in relation to physiological SWRs and the exclusion of pathological ripples, and (2) the design of our study, specifically the association of ripple rates with reported thought content in survey responses. Addressing the first concern, we have included further analysis in the revised manuscript that explicitly demonstrates how we excluded pathological ripples.

Regarding the second concern about our study design, particularly the association of ripple rates with reported thought content in survey responses, we have enhanced the explanation in our manuscript.

Comment 1

For the first concern, the authors could do more to demonstrate that the ripples they are analyzing are similar to those examined in previous studies. Specifically, they could plot the amplitude, duration, and frequency associated with the peak power for each event. While they do use a bandpass filter to limit the events, it's not clear whether the events fall into the lower or upper part of the included range, making it impossible know whether events that may reflect more pathological states are or are not included.

Response to Comment 1:

As the reviewer suggested, we have conducted additional analyses and included these in the revised manuscript. Recognising the importance of differentiating between physiological and pathological ripples, we have also included ground average and time-frequency analyses of the ripples to demonstrate their duration and frequency distribution (Figure 2c–e). This analysis will offer additional insight into the nature of the ripples we have examined and the reasoning behind their exclusion. Please note that we have incorporated the following information into the Results section:

Results (Page 5, Lines 15–17)

‘The detected SWRs demonstrated characteristics consistent with those observed in a previous study, including similarities in waveforms, dominant frequency, and inter ripple intervals²³ (Fig. 2c–e).’

Figure 2c–e

C, D, Mean waveform of field potential (C) and time-frequency map (D) centred on ripple peak (time 0, $n = 633161$, SWR events from 10 patients). E, Distribution of inter-ripple intervals across all patients. ($n=10$ patients; Error bars: standard deviation).

In addition, to characterise the SWRs detected, an analysis was added to elucidate whether epileptic activity influences the detection of SWR events. For the data obtained at 2–7 min before answering the questionnaire, we assessed the coincidence between SWR and the Interictal Epileptiform

Discharges (IEDs). The ripple rate did not exhibit a significant increase consistent with the timing of IEDs, suggesting that IEDs did not increase the detection of SWR events significantly (Supplementary Figure 24). Indeed, the time-frequency map of the IED is quite different from that of SWRs, suggesting that each event can be detected as a separate event. We believe that these additional analyses and the resulting data provide a comprehensive response to the concern and significantly enhance the robustness and clarity of our findings. To refer to these results assessing the influence of epileptic activities on SWR detection, we have added the following part to the Discussion:

Discussion (Page 14, Lines 4–9)

‘First, while we carefully excluded the effects of epileptic activity **by excluding the period corresponding to the seizures**, it remains unclear whether epileptic activity influenced **the detection of SWR events** and the hippocampal function or led to changes in ongoing patterns of thought. **The frequency of the detected SWR events was not influenced by the timing of epileptic activities, despite an increase in power observed in the ripple range, as well as low and high-frequency ranges (Supplementary Figure 24).**’

Supplementary Figure 24

(a) Time-frequency map around interictal epileptic discharges (IEDs) before answering the questionnaire. Time 0 corresponds to the timing of IED onset, which was annotated manually by an epileptologist. (b) The frequency of SWR events was evaluated for iEEGs from 7 to 2 min before answering the questionnaire. The frequency was assessed in 300-ms time bins with 80% overlap from -300 to 300 ms after the onset of IED. The mean and 95% CI values are presented as black lines and shaded areas, respectively.

Comment 2-1

For the second concern, the authors have not provided convincing evidence as to why ripple rates occurring several minutes preceding the time of mind wandering can be physiologically associated with mind wandering several minutes afterwards. Moreover, the graph the authors have shown actually seems to suggest that ripple rates are generally high over a broad range of times up to 11 minutes preceding the survey response, dropping at the time of survey response itself. The timescale is more supportive of a lagged association between ripple rate and mind wandering rather than a direct association, which may be more suggestive, rather, of an association of mind wandering with some underlying brain state which changes more slowly and which is preceded by high ripple rates. I do not think that the second concern can be mitigated short of either (1) changing their primary analysis to be based on the association of ripple rate during the response itself, to the response, or (2) a different behavioral study design.

This point is critical for the authors' interpretation, so I want to make sure it is clear. To explain why SWR rates were taken from 2-7 minutes preceding the response, the authors have provided analysis showing the coefficient of determination between SWR rate and the survey response was significantly higher than a random permutation during this interval. However, the plot seems to in fact show that the SWR rate is quite diffusely high from -1 to -11 minutes before the survey response, and that it (interestingly) drops at the time of the actual survey response (at time 0). Moreover, based on the clarified instructions in the methods section about the "External task focus" question, it seems that the "task" being referred to in the "external-task focus" question would have been assumed to be the act of responding to the survey question on the tablet, which occurs long after the ripple rates being analyzed. It seems an incorrect timescale to assume that SWR rates up to 10 minutes before a survey response about mind wandering can physiologically be temporally linked.

To address this concern, short of a different study design, I think the authors would need to analyze the relation of SWR rate during the response itself with the response as their primary analysis (rather than the relation of SWR rate 2-7 minutes before the response with the response).

Response to Comment 2-1:

We thank the reviewer for raising these important points. Indeed, two important points were: (1) the temporal discrepancy between the SWR and the thought sampling, and (2) the association of the SWR during a specific period with a particular brain state, connecting to a pattern of thoughts.

For the first point, we thank the reviewer for highlighting the concept of 'task' in our questionnaires. We sincerely apologise for the unclear description of the method. In line with the standard practice in mDES studies, when we mention the 'task' in our questionnaires, we are explicitly inquiring about the activity in which patients were involved immediately before their questionnaire. For example, if the patient was watching TV immediately before answering the questionnaire, the reported task is 'watching TV'. In addition, for example, when patients watch a TV program that is unrelated to summer vacation, their minds may inadvertently drift towards thinking about their summer vacation plans. In such instances, they should report not being focused on the task at hand or experiencing mind wandering. The tasks performed by patients immediately before answering the questionnaires varied and included diverse everyday activities, such as eating meals, reading books, and watching SNS. We aimed to investigate the relationship between SWR and the content of spontaneous thoughts across various natural situations. In particular, the purpose of this study was not to investigate self-generated thoughts when performing the task of answering a questionnaire. We discovered a statistically significant relationship between the SWR and the content of thoughts before answering

the questionnaire. This result was consistent with the fact that our questionnaire inquired regarding thought contents preceding the questionnaire response, suggesting that SWRs are related to the thought contents during the investigated period (i.e. the period before answering the questionnaire). To clarify these points, we have made several modifications to the methodology related to how experience sampling was conducted. The revised parts are as follows:

Results (Page 8, Lines 14–15)

‘They were asked to indicate their thought content (14 questions) and mood (3 questions), immediately preceding their responses to the mDES questions, on a Likert scale from 1 to 7’.

Methods (Page 18, Lines 11–13)

‘It should be noted that the patients were instructed to report their feelings and thoughts immediately before answering the questionnaire, not those while answering the questionnaire’.

Methods (Page 19, Lines 4–6)

‘The ‘task’ in the question refers to what the patient was engaged in immediately before responding to the questionnaire, e.g. watching TV, not to the act of answering the questionnaire itself (Supplementary Figure 26)’.

Supplementary Figure 26

As is standard in mDES studies, when we refer to the ‘task’ in our questionnaires, we are specifically inquiring about the activity that patients were engaged in immediately before their questionnaire response. For example, if the patient was watching TV immediately before answering the questionnaire, the reported task is ‘watching TV’. In addition, for example, when patients watch a TV program that is unrelated to summer vacation, their minds may inadvertently drift towards thinking about their summer vacation plans. In such instances, they should report not being focused on the task at hand or experiencing mind wandering. The tasks performed by patients immediately before answering the questionnaires varied and included diverse everyday activities, such as eating meals, reading books, and watching SNS.

Currently, no method exists to measure the contents of thoughts other than the participant’s self-report. If this can be measured by some other physiological marker, as Reviewer 2 pointed out, ‘changing their primary analysis to be based on the association of ripple rate during the response’ could be evaluated. For example, the measurement of biological evaluations, such as accelerations and heart rate by a wearable device aligns with the time of state and SWR. Regarding thought content, as we inquired about the thought content before answering the questionnaires, a certain time lag must be considered. In addition, the act of answering the questionnaire itself is a task that affects thought

contents, as the reviewer pointed out. Focusing on the task of responding to the questionnaire may reduce the likelihood of mind-wandering. If the SWR relates to self-generated thoughts, evaluating self-generated thoughts at the time of the questionnaire becomes challenging because the questionnaire itself constrains the variability of SWRs. Thus, the variation of self-generated thoughts will be limited. Indeed, the relationship between the SWR and the self-generated thought diminished during the questionnaire response, as Reviewer 2 pointed out. This drop in the coefficient of determination suggests the changes in the thought contents or mode of cognition at the time of answering the questionnaires from the contents or states immediately before the questionnaire. Therefore, to assess the self-generated thoughts, we asked the patients to report the contents of their thoughts immediately before answering the questionnaire.

In general, self-reporting one's state refers to a period with a certain delay. A previous similar study, which assessed the relationship between intracranial EEG and the participant's mood, assessed self-reports of mood with intracranial EEG 2–7 min before responding to the self-report¹. In the study, the authors instructed patients to report their emotions before answering the questionnaire and examined the relationship between these emotions and the intracranial EEG. In our first manuscript, we analysed the SWR from 2 to 7 min before responding to the questionnaire, building on insights from this prior study.

Concerning the second point, we agree that the data may not distinguish between alternative views of how SWR contributes to ongoing thought patterns, whether related to the generation of the content or the establishment of a mode of cognition enabling specific thought properties. This to some extent, represents a limitation of the experience sampling approach to comprehending the relationship between the brain and cognition. Conceptually, the experience sampling method, unlike the task-based method, is not ideal for mapping temporal features of cognition as the measures cannot identify the moment that the thought episode commenced². Therefore, our study suggested that periods with a high incidence of SWR are characterised by experiences that exhibit features outlined by our analysis (e.g. intrusive, vivid features). This key aspect is addressed in the revised title, Abstract, Results, and Discussion. The revised parts are as follows:

Title: Hippocampal sharp-wave ripples correlate with **periods of** naturally occurring self-generated thoughts in humans

Abstract (Page 2, Lines 10–13)

'The SWR rates were higher during periods when participants' ongoing thoughts were more vivid, less desirable, had more imaginable properties and exhibited fewer correlations with an external task. Our data indicate a role for SWR in the patterns of ongoing thoughts that humans experience in daily life'.

Discussion (Page 11, Lines 2–5)

'However, using experience sampling, we identified elevated SWRs during intervals when participants reported experiences characterised by less-desired, more vivid, and visually imaginable thoughts, which were less connected to external tasks or future considerations'.

Discussion (Page 12, Lines 8–13)

'First, intrusive thought patterns (broadly similar to those seen in the present study) are most clearly observed in daily life when individuals are unoccupied by a task (doing nothing or resting³). It is worth noting that our mDES requests patients to report their state immediately before answering the

question, otherwise, patients would always report their thoughts during the task of answering the question, making it difficult for them to report intrusive thought patterns’.

Discussion (Page 12, Line 18–Page 13, Line 10)

‘Although mDES is useful for understanding the correlations between cognition and neural activity that are not directly related to an external task, important limitations exist to the inferences derived from this method in probing brain-cognition relationships. For example, our data demonstrates an association between SWRs and reports provided by the participants over a relatively extended period preceding the probe (approximately 5 min). Moreover, mDES and other experience-sampling methods are assumed to be unable to determine the length of episodes of self-generated thought¹⁵, as they cannot establish when the experience began. Consequently, our data is consistent with the possibility that SWRs may not be directly related to the experiential content of thoughts. Instead, they may be associated with the activation of a cognitive mode that facilitates the emergence of thought patterns characterized by the observed features. To gain a deeper understanding regarding the SWR role in ongoing thought patterns, future studies could benefit from combining task manipulations⁴³ and pharmacological interventions⁴⁵ with activity recordings within the hippocampus and descriptions of experience gained from mDES. This multi-method approach could help triangulate the mechanistic role of SWRs in both task-driven and self-generated features of cognition’.

References

1: Sani, O., Yang, Y., Lee, M. et al. Mood variations decoded from multi-site intracranial human brain activity. *Nat Biotechnol.* 2018;36:954–961. doi:10.1038/nbt.4200

2: Smallwood J. Distinguishing how from why the mind wanders: A process–occurrence framework for self-generated mental activity. *Psychol Bull.* 2013;139(3):519-535. doi:10.1037/a0030010

Comment 2-2

As a related question, why does the coefficient of determination drop at time 0 (which presumably is the time of the survey response)? If it is in fact true that the coefficient of determination is high before the response itself but drops during the response, this seems to warrant explanation.

Response to Comment 2-2:

We thank the reviewer for this question. As mentioned in previous comments, our questionnaires inquired about thoughts’ contents before the time of response. Thus, it is reasonable to observe the highest coefficient of determination for SWRs occurring before the questionnaire response. In addition, recognising the task of answering the questionnaire as a new activity for the patient that might alter thought contents adds an important contextual understanding. This aligns with the observed drop in the correlation between SWR at the time of answering the questionnaire and the thought contents reported by the questionnaire. We have discussed this issue as follows:

Discussion (Page 11, Lines 5–8)

‘Interestingly, the SWR rates in the 5 min preceding questionnaire responses were explained by the thought pattern, but the SWR rates while answering the questionnaire were less effectively explained by the same thought pattern, suggesting a potential alteration in thought patterns induced by the task of answering the questionnaire’.

Comment 3

I thank the reviewers for conducting additional experiments to assess whether SWR rates are higher during tasks that assess mind-wandering. I find this helpful to see and think that if this study design is to be used in the future, there could be a few thoughts the authors might consider including in their reporting of this finding, including: 1) Other than the fact that generally backward subtraction is used to assess attention and focusing on breathing generally associated with mind wandering, it would be helpful to quantify the degree of mind wandering during the backward subtraction task (not only the degree of mind wandering during focusing on breathing) in order to state that mind wandering was greater during the task involving focusing on breathing. 2) Since these are two distinct tasks, it would be useful to know why SWR rates can be attributable to the difference in mind wandering rather than another difference between the two tasks, such as the complexity of the task, or the memory requirements of the task.

Response to Comment 3:

We thank the reviewer for pointing this out. We agree that the degree of mind-wandering should also be assessed during the calculation task. We would like to implement this in future evaluations based on this study.

We also agree with the point that the difference in SWR between the two assignments may be attributed to the difference in the assignments. We believe that detailed control experiments can be conducted in the future to investigate the relationship between SWR and mind wandering.

Conversely, the induction of mind wandering by the task is always related to the induction task, as Reviewer 2 pointed out. To examine spontaneous thoughts that are not related to a specific task, we believe that the thought sampling presented in this study may be an effective method. We have added this point to Supplementary Figure 13 as follows.

‘Supplementary Figure 13

This study has several limitations. The degree of mind wandering should also be assessed during the calculation task to precisely demonstrate that it is attributed to differences in mind wandering.

Moreover, the SWR rates can be affected by the task itself. An important aspect to note is that the task demands are a common reason why spontaneous thought rates are reduced (for prior examples of how this manipulation of task demands is useful in elucidating mappings between cognition and brain function please see Turnbull et al., 2019, Nature Communications)’.

Minor comment

I appreciate the authors’ addition of a rose plot in Supplementary Figure 8 to show changes in ripple rate across time. Their plot suggests that detected ripples were higher particularly between ~10pm and shortly before midnight, between approximately midnight and approximately 1 am, and around 1-2pm, and that there were dips around 7-8 am, 12-1pm, and around 6-7pm. Combined with the authors’ analysis of SWR rate before and after the actual start of mealtimes, I agree that the data shows evidence that SWR rate drops during mealtimes, which is quite interesting, particularly with relation to rodent literature which the authors have discussed.

I do think the rose plot shows additional interesting findings regarding the circadian changes in ripple rates, which could be considered for inclusion in the discussion. With regards to the discussion of ripple rates during afternoon versus morning, it appears there is more precisely a relative peak in SWR rate around 1-2:30pm, with the remainder of the afternoon appearing quite similar to the morning time. It

might be helpful to highlight this in the discussion, and also to consider whether there was something occurring between 1-2:30pm that would have caused this. (For example, is this due to a rebound in ripple rates after mealtimes?) Similarly, with regards to the nighttime increase change in ripple rate, it seems this increase is primarily driven by two peaks in ripple rate early in the evening (between ~10pm and shortly before midnight, and another peak between approximately midnight and approximately 1am). It would be helpful to discuss possible sources of these two peaks.

Response to the comment:

We thank the reviewer for the positive comment and for the rose plot suggested, which makes it easier to observe the relationships.

Moreover, we thank the reviewer for highlighting certain peaks in the SWR, and we concur that investigating the causes of these peaks is indeed interesting. We believe that these peaks are sleep-related. Unfortunately, as we did not perform electromyography, the relationship with rapid eye movement sleep remains unclear, but as we observed similar peaks in cortical delta band powers, we believe that we observed changes in SWR related to periodic fluctuations in sleep depth. Concerning the 1–2:00 pm afternoon period, we believe that the observed peak is linked to napping, supported by video monitoring indicating an increase in SWR aligning with the time when the patient seemed to be sleeping. To demonstrate these facts, we have added the average of the diurnal variation of the delta power to Supplementary Figure 9. We have also added the following part to the Discussion section:

Discussion (Page 9, Line 19–Page 10, Line 9)

‘Our study demonstrated a diurnal fluctuation during long-term EEG monitoring, characterized by a decrease in SWR activity around mealtimes and an increase in the afternoon compared to the morning. Previous studies in rodents demonstrated that SWRs occur during nonexploratory states, such as slow-wave sleep, rest, grooming, and eating/drinking²¹. Our results were consistent with the rodents’ study in the **slow-wave sleep** but contradicted **prior associations** with eating. Although we did not investigate SWR activity during the sleep stage due to the lack of electrooculogram and electromyogram data, which are essential **for identifying the stage of sleep**, the clear circadian rhythm of the SWRs, which increased during the night **in tandem with the cortical delta power**, is consistent with the changes in SWR activity during the sleep-wake cycle observed in animal studies. **Similarly, the escalation in SWRs during the early evening (13:00–15:00) may be associated with napping, mirroring the concurrent delta power patterns.**

Supplementary Figure 9

The mean SWR event rates and cortical delta band power of all participants in iEEG recording after day 4 are illustrated. The lines display the mean and 95% confidence interval of the Z-scored ripple event rates (red) and delta power (blue) over 24 h for all participants (n=10). The x-axis displays time and the y-axis demonstrates the Z-score of the SWR event rate or delta band power.

Reviewer #3 (Remarks to the Author):

General comments

The authors have thoroughly addressed my comments. Congratulations on a very interesting study.

Response to the comment:

We wish to extend our deepest gratitude for the time and effort the reviewer has dedicated to reviewing our manuscript. The reviewer's insightful comments and constructive feedback have been invaluable in enhancing the quality and clarity of our work.

REVIEWER COMMENTS

Reviewer #2 (Remarks to the Author):

I thank the authors for their responses to my concerns.

For the first concern, I still ask that the authors do a bit of a more in depth investigation of a few more features relevant for understanding the degree to which pathological HFOs are likely to be present in their sample, as please note that time-frequency analysis is not sufficient for discriminating pathological from non-pathological ripples (Engel et al., 2008), and pHFOs also occur without interictal spikes (Jacobs et al., 2008). A few other features that might be helpful include the amplitude (Ewell et al., 2019), spectral entropy, and fast ripple entropy (Valero et al, 2017). I would suggest that the authors highly consider including these and performing clustering analysis to examine whether there are clusters with these features. If they do not want to do these analyses, they should at least say acknowledge that their sample could contain ripples that could be considered pathological and cite the papers above.

For the second concern, thank you for clarifying that the task at hand was the activity the patients were involved in immediately before answering the questionnaire. In that case, I ask that the authors make it clear in the manuscript that there is ambiguity as to how much the period of the sharp-wave ripple elevation and time period assessed in the survey overlap. For example, I would suggest avoiding statements such as “we identified elevated SWRs during intervals when participants reported experiences characterized by less-desired, more vivid, and viusually imaginable thoughts”, because this suggests a temporal precision that was not present. For example, consider ““we identified elevated SWRs around the time of intervals when participants reported experiences characterized by less-desired, more vivid, and viusually imaginable thoughts”. Other similar places in the manuscript can be similarly revised, such as in the Abstract.

Reviewer #2 (Remarks on code availability):

The code consists of three matlab functions to detect and filter ripple events. If the goal is to share a few functions, this is fine. If the goal is to allow others to replicate the work, this is close to useless. I did not see the data file that was supposed to be included, nor are there detailed instructions on how to interpret the output of the functions.

If you are serious about others being able to replicate these results with the data, the authors would need to provide the raw data and all of the scripts, with comments as to how to use them. This would likely be a few months of work. Otherwise this requirement to provide the code does not provide much of value.

Point-by-Point Responses to Reviewers' Comments
Hippocampal sharp-wave ripples correlate with periods of naturally occurring self-generated thoughts in humans
(NCOMMS-23-27587-B)

We are grateful for the reviewers' insightful feedback and suggestions. The constructive critiques have significantly aided in refining and enhancing our manuscript. We have diligently addressed each concern raised, and have highlighted all additions and modifications in **yellow** in the revised manuscript. Below, we provide a detailed response to each comment, with the reviewers' remarks presented in grey.

Responses to Reviewers' Comments

Reviewer #2 (Remarks to the Author):

Comment 1

For the first concern, I still ask that the authors do a bit of a more in depth investigation of a few more features relevant for understanding the degree to which pathological HFOs are likely to be present in their sample, as please note that time-frequency analysis is not sufficient for discriminating pathological from non-pathological ripples (Engel et al., 2008), and pHFOs also occur without interictal spikes (Jacobs et al., 2008). A few other features that might be helpful include the amplitude (Ewell et al., 2019), spectral entropy, and fast ripple entropy (Valero et al, 2017). I would suggest that the authors highly consider including these and performing clustering analysis to examine whether there are clusters with these features. If they do not want to do these analyses, they should at least say acknowledge that their sample could contain ripples that could be considered pathological and cite the papers above.

Response to Comment 1:

We sincerely appreciate the reviewer's insightful comments. In response to concerns regarding the potential presence of pathological high-frequency oscillations (HFO) in our sample, we have taken these suggestions into serious consideration.

In line with the recommendation, we have applied the analytical method proposed by Ewell et al. (2019) to our dataset (see Supplementary Figure 27). Specifically, we have generated distribution maps detailing the low-frequency envelope amplitude and peak frequency for ripple events across all patients as shown in bellow. Our analysis revealed that all detected events were included in a single cluster. Therefore, we could not find the clear evidence of the pathological HFOs in our data. However, we acknowledge that the result does not show that the possibility of pathological HFO was completely excluded from our samples. Pathological HFO could coexist with physiological Sharp Wave Ripples (SWRs) (Ewell et al., 2019; Valero et al., 2017). That being said, these previous studies also demonstrated that the physiological SWR maintains the functional characteristics, even with the coexistence of the pathological HFO. Indeed, our results also demonstrated a notable association between SWRs and thought content. On the other hand, we did not find a significant relationship between epileptic activities and the ongoing thought pattern. Therefore, even if the detected SWRs were contaminated with pathological HFOs, the relationship between SWRs and the ongoing thought pattern would have been detected. In light of these considerations, we have updated the Discussion section of our manuscript to reflect these results and added the cluster analysis in Supplementary Figure 27.

“Although our data suggest a correlation between SWRs in the hippocampus and the content of ongoing thought patterns, there are several limitations that should be borne in mind when considering these results. First, while we carefully excluded the effects of epileptic activity by excluding the period corresponding to the seizures, it remains unclear whether epileptic activity influenced the detection of SWR events and the hippocampal function or led to changes in ongoing patterns of thought. The frequency of the detected SWR events was not influenced by the timing of epileptic activities, despite an increase in power observed in the ripple range, as well as low and high-frequency ranges (Supplementary Figure 24). However, it is difficult to exclude the possibility that the detected SWRs include some pathological high-frequency oscillations (HFOs), which is one of the epileptic activities⁵³. Although the frequency and amplitude characteristics of the detected SWRs were consistent with those of the SWRs (Figure 2d; see also Supplementary Figure 27), it was difficult to distinguish between the SWRs and the pathological HFOs based on each waveform⁵⁴. Previous studies have demonstrated that physiological SWRs and pathological HFOs coexist in the epileptic hippocampus, while the physiological SWRs maintain the functional characteristics even with the coexistence of the pathological HFOs^{55,56}. Therefore, the observed relationship between the ongoing thought patterns and SWRs would reflect the physiological property of the SWR even with the contaminated pathological HFOs. In addition, although the frequency of interictal epileptic activity was weakly explained by the thought sampling results, the model weights differed from those of the SWR model (Supplementary Figure 17); suggesting the relationship between SWR and ongoing thought patterns were not originated from the relationship with the epileptic activities. Moreover, the frequency of the epileptic activity and excluded periods did not explain the diurnal variation in the SWR (Supplementary Figure 10). Finally, although the patients took antiepileptic medication during mealtime, the SWR event rates decreased at the start of the meal and before the administration of medicines (Supplementary Figure 12). These results suggest that the epileptic-related activities were not directly attributed to the relation between SWRs and thought content.”

In Supplementary Figure 27

Supplementary figure 27

To assess the pathology of detected ripples, the method outlined by Ewell et al. (2019) was applied to our dataset ($N = 10$). Peak slow wave amplitude and peak frequency were determined for every ripple event. To measure the slow wave amplitude, segments of data lasting 500 milliseconds, centered on

the ripple event, were band-pass filtered between 0.2 Hz and 40 Hz, and the maximum absolute amplitude of the filtered data was taken. For peak frequency calculation, the fast Fourier transform was applied to the data segment. The highest power peak for frequencies greater than 70 Hz was identified as the peak frequency. Density plot reveals that all events were included in a single cluster of similar frequency and envelope amplitude.

Comment 2

For the second concern, thank you for clarifying that the task at hand was the activity the patients were involved in immediately before answering the questionnaire. In that case, I ask that the authors make it clear in the manuscript that there is ambiguity as to how much the period of the sharp-wave ripple elevation and time period assessed in the survey overlap. For example, I would suggest avoiding statements such as “we identified elevated SWRs during intervals when participants reported experiences characterized by less-desired, more vivid, and visually imaginable thoughts”, because this suggests a temporal precision that was not present. For example, consider “we identified elevated SWRs around the time of intervals when participants reported experiences characterized by less-desired, more vivid, and visually imaginable thoughts”. Other similar places in the manuscript can be similarly revised, such as in the Abstract.

Response to Comment 2:

We thank the reviewer for the suggestions and understand the importance of accurately representing the temporal relationships in our study to avoid overinterpretation of our findings. As the reviewer suggested, we have updated our manuscript.

Abstract (Page 2, Lines 10)

“The SWR rates were higher during extended periods of time when participants’ ongoing thoughts were more vivid, less desirable, had more imaginable properties, and exhibited fewer correlations with an external task.”

Discussion (Page 10, Line 22)

“we identified elevated SWRs during extended periods of time when participants reported experiences characterized by less-desired, more vivid, and visually imaginable thoughts, which were less connected to external tasks or future considerations.”

Reviewer #2 (Remarks on code availability):

The code consists of three matlab functions to detect and filter ripple events. If the goal is to share a few functions, this is fine. If the goal is to allow others to replicate the work, this is close to useless. I did not see the data file that was supposed to be included, nor are there detailed instructions on how to interpret the output of the functions.

If you are serious about others being able to replicate these results with the data, the authors would need to provide the raw data and all of the scripts, with comments as to how to use them. This would likely be a few months of work. Otherwise this requirement to provide the code does not provide much of value.

Response to the comment:

We thank the reviewer for the feedback on the code and data availability in our manuscript. We understand the concerns and have made several clarifications and additions to address them.

Due to ethical restrictions from the Ethics Committee, we cannot upload raw data containing clinical information that may be used to identify individuals. However, interested researchers can contact the corresponding author to request access to the data. To clarify this point, we have made a note to this end in the Data Availability section of the revised manuscript.

Data Availability (page 24, line 19)

“The raw data are not publicly available due to their containing information that could compromise the privacy of research participants. The SWR frequency data generated and analyzed during the current study are available on Figshare at <https://figshare.com/s/568e9da3493111453f08>. The remainder of the data in support of the findings of this study are available from the corresponding author, T.Y, on reasonable request.

Since we did not upload the code needed to replicate our results, we only uploaded the functions necessary for SWR detection. To improve usability, we have provided detailed instructions on using these functions, and have also uploaded supplementary function files needed for the process of SWR detection.